# Decomposing Complex Visual Comprehension into Atomic Visual Skills for Vision Language Models

**Hyunsik R. Chae**                                          HSCH@MATH.UCLA.EDU
*Department of Mathematics*
*UCLA*
*Los Angeles, CA 90095, USA*

**Seungwoo Yoon**                                          DOTORI000@SNU.AC.KR
*Department of Computer Science*
*Seoul National University*
*Seoul 08826, Republic of Korea*

**Jaden Park**                                          JADENPARK@CS.WISC.EDU
*Department of Computer Science*
*University of Wisconsin–Madison*
*Madison, WI 53706-1380, USA*

**Chloe Yewon Chun**                                          CHLOE1000@SNU.AC.KR
*Department of Mathematical Science*
*Seoul National University*
*Seoul 08826, Republic of Korea*

**Yongin Cho**                                          YONGINCHOA@SNU.AC.KR
*Department of Chemistry*
*Seoul National University*
*Seoul 08826, Republic of Korea*

**Mu Cai**                                          IM.MUCAI@GMAIL.COM
*Google DeepMind*
*London, N1C 4AG, UK*

**Yong Jae Lee**                                          YONGJAELEE@CS.WISC.EDU
*Department of Computer Science*
*University of Wisconsin–Madison*
*Madison, WI 53706-1380, USA*

**Ernest K. Ryu**                                          ERYU@MATH.UCLA.EDU
*Department of Mathematics*
*UCLA*
*Los Angeles, CA 90095, USA*

## Abstract

Recent Vision-Language Models (VLMs) have demonstrated impressive multimodal comprehension and reasoning capabilities, yet they often struggle with trivially simple visual tasks. In this work, we focus on the domain of basic 2D Euclidean geometry and systematically categorize the fundamental, indivisible visual perception skills, which we refer to as atomic visual skills. We then introduce the Atomic Visual Skills Dataset (AVSD) for

The review page for this paper is available at: `https://openreview.net/forum?id=d18yO97PCp`.

evaluating VLMs on the atomic visual skills. Using AVSD, we benchmark state-of-the-art VLMs and find that they struggle with these tasks, despite being trivial for adult humans. Our findings highlight the need for purpose-built datasets to train and evaluate VLMs on atomic, rather than composite, visual perception tasks.

**Keywords:** vision language models, plane geometry, benchmark, mathematical reasoning, visual perception

## 1 Introduction

Recent Vision Language Models (VLMs), also referred to more generally as Multimodal Large Language Models (MLLM), integrate vision components into language models and demonstrate an impressive breadth of multimodal comprehension and reasoning capabilities (Bordes et al. (2024)). At the same time, however, VLMs often struggle with trivially easy visual tasks as shown in Figure 1, a puzzling phenomenon that seems almost contradictory to their remarkable performance (Fu et al. (2024); Rahmanzadehgervi et al. (2024)).

In this work, we introduce the Atomic Visual Skills Dataset (AVSD) to evaluate models on fundamental, indivisible visual perception skills. We refer to these skills as *atomic visual skills*, and we systematically categorize 36 atomic visual skills that encompass diagrams arising in the domain of 2D Euclidean geometry at the level of high school or lower. We then evaluate the state-of-the-art VLMs on AVSD and demonstrate that current VLMs are incapable of such atomic visual skills.

This inability of VLMs to accurately perceive such basic geometric features is concerning, as these capabilities are likely crucial for multimodal perception and reasoning tasks that require *precise* understandings of the visual input. Recent studies have highlighted VLMs' struggles with tables (Zheng et al. (2024)), scientific plots (Roberts et al. (2024a)), and other structured visual data Roberts et al. (2024b), and our results suggest that these challenges may stem from the VLMs limitation in basic visual perception. Therefore, our findings underscore the need for specialized datasets to train and evaluate VLMs on atomic, rather than composite, visual perception tasks.

AVSD[1] consists of three sub-datasets: AVSD-h, a handcrafted dataset designed for in-depth evaluation; AVSD-s, a procedurally generated synthetic dataset styled to resemble geometry problems found in mathematics textbooks and exams; and AVSD-c, a synthetic dataset with style and texture augmentations via ControlNet (Zhang et al. (2023a)), aimed at evaluating VLMs' robustness to variations in styles. In total, the dataset provides an average of 366 problems per skill across the 36 atomic visual skills. These problems are designed to be trivial for adult humans, but all current state-of-the-art VLMs–both open-source and commercial–struggle with them. This holds true even for domain-specific models, such as Math-LLaVA (Shi et al. (2024)) and G-LLaVA (Gao et al. (2025)), as well as for models employing chain-of-thought reasoning or test-time scaling, including OpenAI o1 and o3 (OpenAI (Dec. 2024,A)), and Gemini 2.5 Flash and Pro (Deepmind (Apr. 2025a,A)).

---

1. Dataset and code available at `https://github.com/hs-chae/AVSD25.git`

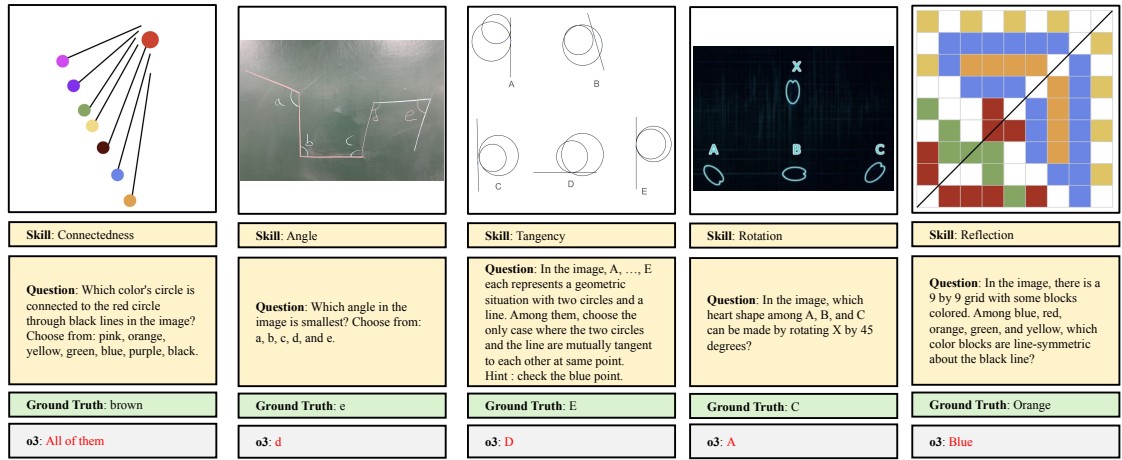

Figure 1: Examples of AVSD problems and responses by o3 model. Other state-of-the-art models exhibit similar failures. These examples demonstrate a deficiency in the VLMs' understanding of basic geometric concepts.

## 2 Related works

**VLM benchmarks and language shortcuts.** Existing VLM benchmarks evaluate models on their ability to solve diverse vision-language problems from general real-world tasks (Antol et al. (2015); Goyal et al. (2017); Gurari et al. (2018)), tasks that require specific skills such as high-school geometry (Lu et al. (2024); Kazemi et al. (2023); Chen et al. (2022); Cao and Xiao (2022)), analyzing charts and tables (Masry et al. (2022); Zheng et al. (2024); Methani et al. (2020)), and other scientific visual data (Kafle et al. (2018); Kembhavi et al. (2016)). Recent VLM research has also moved beyond conventional recognition and question answering toward more demanding visual tasks, including agentic 3D scene understanding (Liu et al. (2025)), image restoration for autonomous driving (Lin et al. (2025a)), and interactive photo retouching (Lin et al. (2025b)). These directions underscore the need for precise and reliable visual grounding across diverse domains. However, most VLM benchmarks do not contain a mechanism for verifying whether a correct solution is based on correctly comprehending the visual information, allowing models to sometimes rely on linguistic biases to find a solution (Bordes et al. (2024)). Lin et al. (2024) revealed that by simply avoiding implausible or less fluent sentences, blind language models can distinguish the correct description of an image from wrong ones on CREPE (Ma et al. (2023)), VL-Checklist (Zhao et al. (2022)), and ARO (Yuksekgonul et al. (2023)). MathVerse (Zhang et al. (2024b)) observed that, when solving geometry problems,

Some recent work has started to seek unbiased ways to measure visual capabilities. Winoground (Thrush et al. (2022)) prevents choosing image captions based on the plausibility of the sentence structure, by providing two images with same objects or concepts but with different relationships. Blink (Fu et al. (2024)) and CV-Bench (Tong et al. (2024a)) present novel vision-oriented tasks with minimized effects of linguistic biases. MMStar (Chen et al. (2024)) demonstrates that existing multi-modal benchmarks include problems where

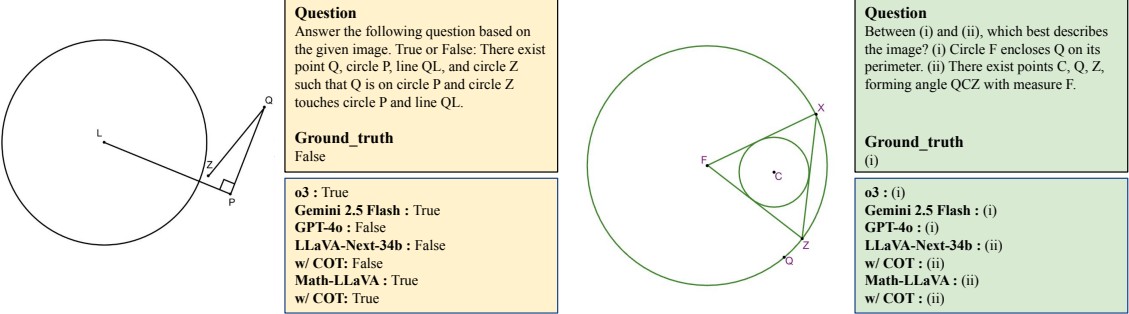

Figure 2: Examples of $\nu$-geometry. These tasks test composite geometric perception but do not require any mathematical reasoning. They demonstrate that the state-of-the-art VLMs struggle with geometric perception, even before they get to geometric reasoning.

the visual content is unnecessary, as well as problems where the visual content is necessary but can still be answered by a language-only model. They therefore propose a benchmark of 1,500 problems designed to avoid both issues.

**Compositional reasoning.** There has been intensive recent research on the compositional capabilities of Language Models (Arora and Goyal (2023); Xu et al. (2024); He et al. (2024); Song et al. (2025); Ramesh et al. (2024); Zhao et al. (2024); Lake and Baroni (2018); nón et al. (2021); Press et al. (2023)). VLMs have additionally shown compositional capabilities in visual tasks (Lewis et al. (2024); Okawa et al. (2023); Ma et al. (2023); Zhao et al. (2022); Yuksekgonul et al. (2023)). However, such studies left the visual portion with less attention, thus vulnerable to linguistic shortcuts such as removing grammatically wrong sentences or choosing more realistic sentences as answers. To mitigate this issue, SugarCrepe (Hsieh et al. (2023)) generated sentences with ChatGPT to provide incorrect captions of given images, with different compositional structures while as realistic as the ground truths.

**Geometry problem solving.** Solving geometry problems is a useful but yet difficult task for VLMs. To evaluate the geometry problem solving capability of VLMs, there have been a lot of visual question answering datasets regarding the geometry. For example, MathVista (Lu et al. (2024)), MathVerse (Zhang et al. (2024b)), and MathVision (Wang et al. (2024a)) evaluated intensive VLMs from open-source to commercial models. Such benchmarks eventually emphasize that current VLMs are not good at geometry problem solving.

There has also been research trying to specialize VLMs in geometry or math. G-LLaVA (Gao et al. (2025)) and Math-LLaVA (Shi et al. (2024)) collected images from existing geometry dataset and instruction-finetuned the VLM. MAVIS further utilizes their own data engine and preference alignment using DPO.

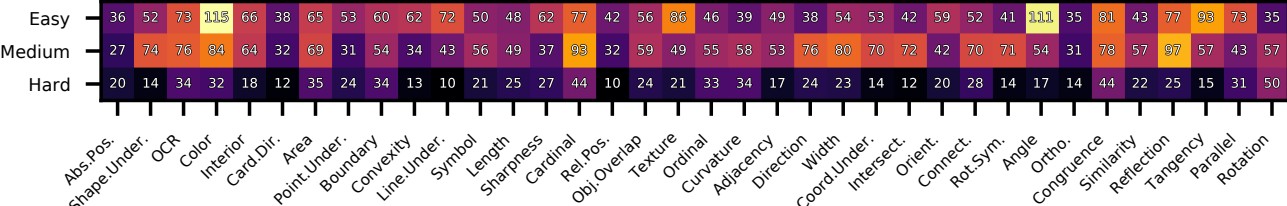

Figure 3: List of 36 atomic visual skills and the number of easy, medium, and hard problems for each skill from AVSD-h. The difficulty is judged by the authors. We provide a total of 5,163 new handcrafted problems.

Though not perfectly aligned with geometric problem solving, there are some prior works composing datasets regarding tables (Zheng et al. (2024)) and scientific figures (Roberts et al. (2024a,b)) either for training or evaluation, to which mathematical diagram understanding is related.

**Research on atomic skills of LLMs.** To understand the capabilities of LLMs, there has been prior work on studying LLMs in simple idealized experiments. This includes research on in-context learning (Lin and Lee (2024); Min et al. (2022)), arithmetic (addition and multiplication) (nón et al. (2021); Hanna et al. (2023); Lee et al. (2024)), fact search and reverse fact search (Allen-Zhu and Li (2025); Berglund et al. (2024); Golovneva et al. (2024)), and programming (Austin et al. (2021); Rozière et al. (2023); Guo et al. (2024)).

Euclid (Zhang et al. (2024a)) decomposes the mathematical problems into seven tasks motivated by Euclid's Postulates. Although all problems in Euclidean geometry can be explained only by five postulates in ideal cases, Euclid cannot evaluate the concepts that derive from those postulates. Moreover, it lacks the data of modern but simple knowledge such as smoothness or tangency because of its design. Therefore, Euclid can cover only a small scope of plane geometry.

However, there have been far fewer studies of this kind for vision language models. Paiss et al. (Paiss et al. (2023)) focused on counting objects in image and suggested CountBench. Shen et al. (Shen et al. (2024)) suggests a skill-based approach to evaluating VLMs, but their list of skills is not atomic. CV-Bench (Tong et al. (2024a)) evaluates 4 vision-centric skills: spatial relationship, object count, depth order, and relative distance. MMVP (Tong et al. (2024b)) challenges VLMs to understand 9 visual patterns. Rahmanzadehgervi et al. (2024) observed failures of VLMs with 7 simple tasks focusing on fundamental geometric features, some of which share similar approaches with AVSD.

**Edge conditional image generation.** With the advancements of diffusion models (Dhariwal and Nichol (2021); Ho et al. (2020); Song et al. (2021)), recent image generative models are available to generate realistic images. Especially, Latent Diffusion Models (Rombach et al. (2022); Labs (2024)) have shown promising text-to-image generation results in diverse subjects. On top of that, ControlNet (Zhang et al. (2023a)) enables spatial conditioning on pre-trained diffusion models. For instance, the spatial condition of ControlNet includes Canny Edge (Canny (1986)), which will be further used to augment our synthetically generated dataset.

## 3 Failure on composite geometric perception

Before considering atomic visual skills, we first verify the inability of VLMs on *composite* geometric perception tasks, which require the integration of multiple atomic visual skills. For this, we introduce the $\nu$-geometry dataset, which is synthetically constructed using the AlphaGeometry framework (Trinh et al. (2024)). By selecting specific construction rules, we can visualize diagrams and procedurally generate corresponding captions. Examples are shown in Figure 2, and further details of the construction are provided in Appendix A.

Although we do not consider $\nu$-geometry to be a major contribution, it serves as a novel dataset that effectively assesses VLMs' ability to perceive composite geometric *perception* while excluding confounding factors related to geometric *reasoning*. Our evaluations, summarized in Table 1, reveal that many state-of-the-art VLMs perform poorly in composite perception, considering that random guessing should have 0.5 accuracy, underscoring the need to decompose perception tasks into simpler atomic components.

| Model | LN-13b | LN-34b | Math-L | G-L | 4o | o1 | o3 | Flash | Pro |
|---|---|---|---|---|---|---|---|---|---|
| Accuracy | 0.57 | 0.60 | 0.44 | 0.18 | 0.78 | 0.82 | 0.83 | 0.80 | 0.80 |
| Cot-Acc | 0.62 | 0.69 | 0.46 | 0.16 | 0.85 | - | - | - | - |

Table 1: VLM's performances on composite geometric perception tasks of $\nu$-geometry. In the model names, 'LN' stands for LLaVA-NeXT and 'L' stands for LLaVA.

## 4 Atomic Visual Skills Dataset (AVSD)

Many visual perception tasks can be decomposed into fundamental, fine-grained diagnostic visual perception skills, which we refer to as *atomic visual skills*. We use "atomic" operationally, not as a formal claim of rigorous indivisibility. For adult humans, these skills are trivially simple and require little to no thinking to perform. Therefore, we use the term **perception**, contrasting with the term **reasoning**, to emphasize our belief that these skills do not require much reasoning or thinking to perform, for both humans and VLMs. This belief is partially supported by our findings of Section 5.1 that chain-of-thought prompting does not help with AVSD.

**Identifying atomic visual skills.** We systematically categorize 36 atomic visual skills based on the following criteria: (i) each skill is intuitive and trivial for adult humans, (ii) each skill cannot be decomposed further, or doing so would be unnatural, and (iii) the list of atomic visual skills should comprehensively cover the abilities required for perceiving geometric diagrams arising in mathematics at the level of high school or lower. While this definition is not a fully rigorous one, we found it to be sufficiently clear and substantive for our work. Figure 3 shows the list of the 36 skills. Their formal definitions and further illustrations are provided in Appendix B.

**Dataset format.** AVSD consists of three sub-datasets: AVSD-h, a handcrafted dataset designed for in-depth evaluation; AVSD-s, a procedurally generated synthetic dataset styled to resemble geometry problems found in mathematics textbooks and exams; and AVSD-c, a synthetic dataset with style and texture augmentations via ControlNet (Zhang et al.

| (a) Overall | | (b) AVSD-h | |
| --- | --- | --- | --- |
| Total questions | 13,188 | Number of skills | 36 |
| AVSD-h | 5,163 (39.1%) | Avg. # questions per skill | 143.4 |
| AVSD-s | 5,400 (40.9%) | # of "easy" questions | 2,136 |
| AVSD-c | 2,625 (19.9%) | # of "medium" questions | 2,087 |
| | | # of "hard" questions | 90 |
| (c) AVSD-s | | (d) AVSD-c | |
| Number of skills | 36 | Number of skills | 35 |
| Avg. # questions per skill | 150 | Avg. # questions per skill | 75 |
| Avg. # tasks per skill | 11.6 | Avg. # tasks per skill | 11.6 |
| | | # of ControlNet prompts | 130 |

Table 2: Statistics of the AVSD dataset. Note that AVSD-c has only 35 skills, as we observed that color problems frequently became unclear after style transformations.

(2023a)), aimed at evaluating VLMs' robustness to variations in styles. In total, the dataset provides an average of 366 problems per skill across the 36 atomic visual skills, to a total of 13,188 problems, as shown in Table 2.

Each problem consists of an image, a question, and an answer key. In the construction of the dataset, we paid attention to two key attributes: diversity and skill isolation.

**Diversity.** Although we focus on the set of only 36 skills, we make sure problems feature diverse expressions and formats, as illustrated by the sample problems in Figure 1. Moreover, since it is established in prior work that the performance of VLMs can heavily depend on the formatting of the prompt and the order of the choices (Zong et al. (2024)), we diversify text prompts and the alphabet labels (e.g. which letter to use to label a triangle's points).

**Skill isolation.** Each question is designed to target a specific atomic visual skill, minimizing the overlap with other skills. However, complete isolation is impossible. To address this, we create a diverse set of tasks for each skill, reducing the influence of each individual overlapping skill. For instance, when assessing cardinal perception, we minimize the impact of other skills by asking the count of a diverse range of concepts and objects, including colors, points, lines, and geometric figures.

### 4.1 AVSD-h

This sub-dataset comprises 5,163 newly handcrafted problems, offering significantly greater diversity than the synthetically generated data in AVSD-s and AVSD-c. Additionally, since all images and questions are newly created, they are free from data contamination concerns. The problems in this sub-dataset (but not for AVSD-s or AVSD-c) are categorized into three difficulty levels: easy, medium, and hard. Problems categorized as easy or medium are quickly solvable by humans, whereas hard questions are more time-consuming but still clear and easily verifiable. We clarify that the difficulty levels were determined by the authors, so there is a degree of subjectivity to the categorization.

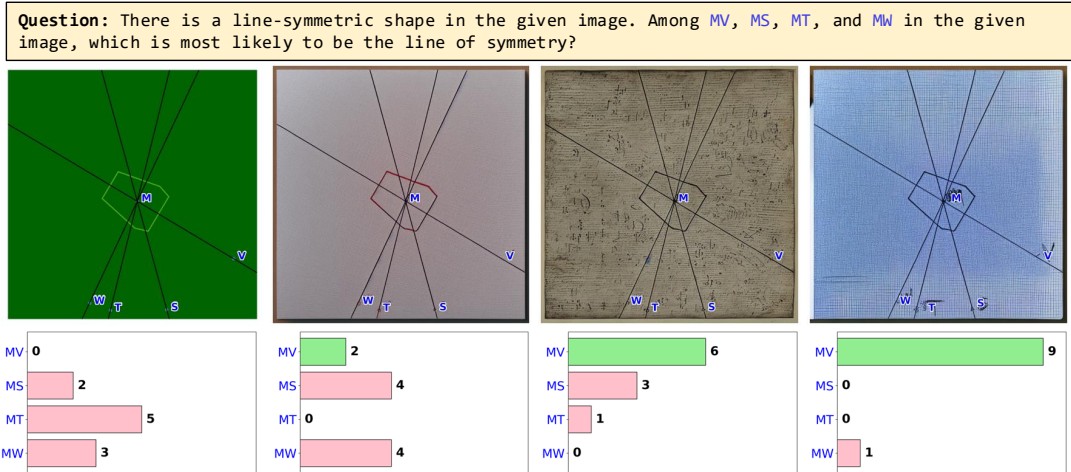

Figure 4: Statistics of GPT-4o response on the same question with different styles. This example shows that VLMs are sensitive to the variation in image style. This motivates the AVSD-c sub-dataset, designed to assess VLMs' robustness to perceive geometric features independent of image style.

## 4.2 AVSD-s

This sub-dataset is generated using the AlphaGeometry framework (Trinh et al. (2024)) in a manner similar to the $\nu$-geometry of Section 3. The precise generation process is detailed in Appendix C.

Our main AVSD-s sub-dataset consists of 150 problems per skill, totaling 5,400 across the 36 skills. However, because these problems are generated procedurally, there is no limit to the data size. Therefore, we also provide AVSD-s-train, which consists of 10,000 problems per skill, 360,000 total, intended for training and fine-tuning purposes. We also provide the generation code, which allows the user to generate an indefinite amount of data.

## 4.3 AVSD-c

We find that the geometric perception abilities of state-of-the-art VLMs are sensitive to changes in image style, as shown in the example of Figure 4. This is undesirable, and we would ideally want VLMs to robustly perceive geometric information independent of the image style. However, existing benchmarks such as Mathverse (Zhang et al. (2024b)) consist of problems that are very limited in the diversity of the image style.

Motivated by this, we introduce AVSD-c, a sub-dataset designed to evaluate the robustness of VLMs' geometric perception capabilities across varying styles. AVSD-c consists of 2,625 synthetically generated images with diverse styles imbued with ControlNet (Zhang et al. (2023a)). The base images are generated in the same manner as AVSD-s and then processed with a ControlNet model fine-tuned using the Flux diffusion model (Labs (2024)). Figure 5 illustrates this process.

Since the Flux diffusion model can utilize detailed natural language prompts, the style augmentation can have significant variety. Additionally, because ControlNet conditions the

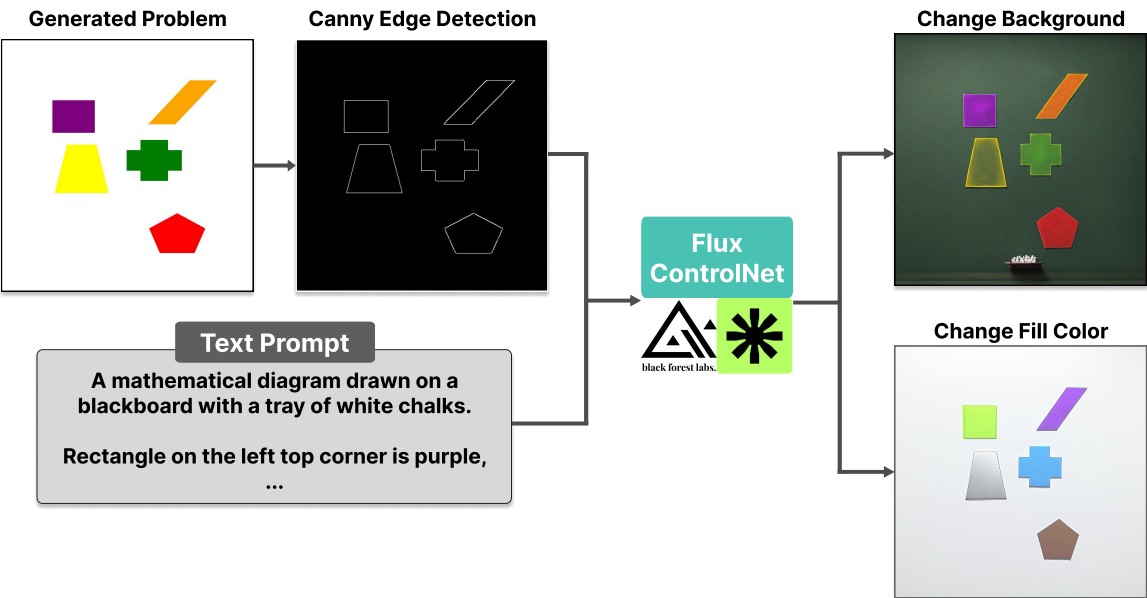

Figure 5: AVSD-c consists of synthetically generated images with diverse styles imbued with ControlNet. The generation content is conditioned on the Canny edges of the input image while the style is conditioned on the natural language prompt.

generation on the Canny edges of the input image, the style-augmented image does not lose any information or acquire extraneous detail that may interfere with answering the question. To further refine our generation process, we also employ a filtering mechanism by measuring the similarity of the Canny edges of the original and the augmented images. More details and examples can be found in Appendix C.3.

## 5 Experiments

In this section, we present experimental results. We start by describing some key experimental details and then present our results evaluating the VLM's performance and fine-tuning the VLMs on the AVSD dataset.

**Dataset verification.** We verify the synthetically generated AVSD-s and AVSD-c sub-datasets to check for any defects or ambiguities that may arise during the data generation process. The authors solved randomly chosen 30 images per skill, totaling 1080 across the 36 skills. The authors scored with 99% accuracy, confirming the solvability of the problems. The 1% of failures corresponded to cases where the generated diagrams were not adequately visible due to tightly overlapping components and when the ControlNet style augmentation removed some essential information from the diagram.

**VLMs.** We evaluate three types of VLMs on AVSD: (i) state-of-the-art proprietary models: GPT-4o (OpenAI (2024b,a)), Openai-o1 (OpenAI (Dec. 2024)), Openai-o3 (OpenAI (Apr. 2025)), Gemini 2.5 Flash (Deepmind (Apr. 2025a)), and Gemini 2.5 Pro (Deepmind (Apr. 2025b)), (ii) popular mid-sized open-weight models: LLaVA-Next (13B, 34B) (Liu

|        | Phi-3.5-VI | LN-13b | LN-34b | Math-L-13b | L-OV-7b | QVL-7b | GPT-4o | o1   | o3   | Flash | Pro  |
|--------|------------|--------|--------|------------|---------|--------|--------|------|------|-------|------|
| AVSD-h | 0.36       | 0.33   | 0.38   | 0.30       | 0.42    | 0.49   | 0.62   | 0.68 | 0.74 | 0.71  | 0.75 |
| AVSD-s | 0.36       | 0.31   | 0.35   | 0.32       | 0.38    | 0.45   | 0.55   | 0.56 | 0.62 | 0.71  | 0.72 |
| AVSD-c | 0.27       | 0.23   | 0.25   | 0.26       | 0.32    | 0.34   | 0.43   | 0.46 | 0.50 | 0.60  | 0.64 |
| Overall| 0.34       | 0.30   | 0.34   | 0.30       | 0.39    | 0.44   | 0.55   | 0.59 | 0.65 | 0.68  | 0.72 |

Table 3: Evaluation of state-of-the-art VLMs on AVSD-h, AVSD-s, and AVSD-c. 'LN' stands for LLaVA-NeXT, 'L' for LLaVA, 'OV' for OneVision, 'QVL' for Qwen2.5-VL, and 'Phi-3.5-VI' for Phi-3.5-Vision-Instruct. The difference between AVSD-s and AVSD-c is the style augmentations of AVSD-c via ControlNet, and the performance degradation indicates that the VLMs are not robust with respect to style changes. More details are in F.5.

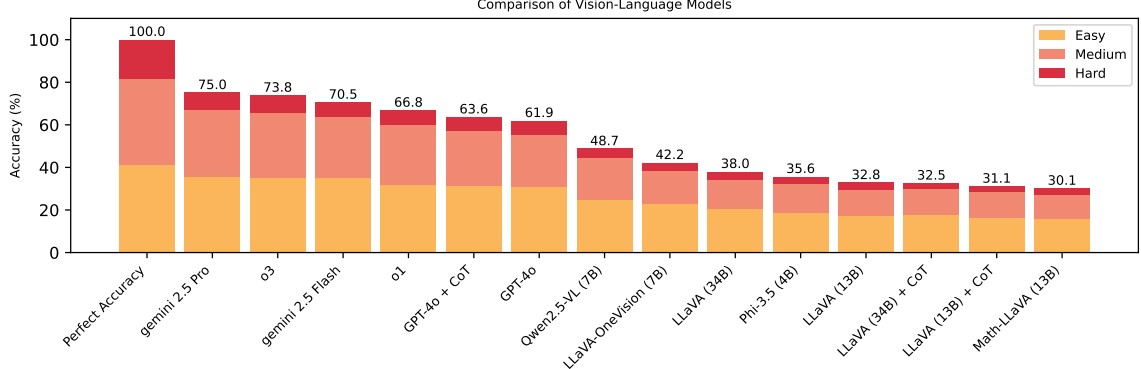

Figure 6: Evaluation results on AVSD-h. $+CoT$ implies the performance of the model on the right with chain-of-thought (CoT) prompting (Kojima et al. (2022)). The area ratios of each colored section are aligned with the actual ratio of problem counts. Further details are provided in Appendix F.5 and Section 5.1.

et al. (2023, 2024)), LLaVA-OneVision (7B) (Li et al. (2025)), Qwen2.5-VL (7B) (Bai et al. (2025)), Phi-3.5-Vision (4B) (Abdin et al. (2024)), and (iii) domain-specific VLMs specifically trained for geometry or mathematics: Math-LLaVA (13B) (Shi et al. (2024)), G-LLaVA (13B) (Gao et al. (2025)). Further details of model versions are provided in Appendix D.

**Evaluation protocol.** The evaluation protocol consists of three steps. First, we provide the VLM with the image-question pair and solicit a response. As we further discuss later, we also explore their performances with chain-of-thought (CoT) prompting (Wei et al. (2024); Kojima et al. (2022)). Second, we extract the answer from the VLM's response using GPT-4o-mini (OpenAI (2024b)). Third, we ask GPT-4o-mini to score the answer by comparing the extracted answer with the answer key. We award 1 point for a correct answer and 0 points otherwise, without any partial credit. We further validate this LLM-based scoring protocol with a Gemini 3 Flash scorer ablation; details are provided in Appendix E.

## 5.1 Evaluation of state-of-the-art VLMs

Figure 6, Figure 7, and Table 3 present summaries of the VLMs performance on AVSD. Further details and additional evaluation results are provided in Table 14.

**Finding: Models share strengths and weaknesses.** Figure 7 presents the accuracies of selected models on each skill. Performance varied significantly across skills. For example, most VLMs performed well on `OCR`, `Absolute Position`, and `Shapes`, but performed poorly on `Tangency`, `Parallel`, and `Angle`. Interestingly, we observed broad agreement across models regarding which skills were more or less challenging. To further examine these weaknesses, we manually inspected their incorrect predictions and we provide qualitative discussion in Appendix F.1.

**Finding: Geometry-specialized training does not guarantee better atomic perception.** Notably, Math-LLaVA(Shi et al. (2024)) and G-LLaVA(Gao et al. (2025)), which are specifically trained with geometry data, do not outperform general-purpose VLMs of similar size on almost any skills within AVSD. For G-LLaVA in particular, we observed frequent instruction-following and answer-formatting failures. This indicates that current specialization on composite geometry tasks does not automatically translate into stronger atomic visual perception. The specific results are shown in Table 3.

**Finding: Chain-of-thought is not helpful, but reasoning may.** We evaluated several non-reasoning models with chain-of-thought (CoT) prompting (Kojima et al. (2022)). We found that CoT did not help for most skills, and for some skills, it even worsened the performance, as shown in Table 4. This contrasts with prior work, which found CoT to be beneficial for certain visual reasoning tasks (Lu et al. (2024); Wang et al. (2024b)). We attribute this difference to our hypothesis that the atomic visual skills of AVSD require simple "perception" and, therefore, do not benefit from the additional "reasoning" steps afforded by CoT prompting. On the other hand, the closed-source Gemini 2.5 Flash model did meaningfully benefit from the use of reasoning, which is similar but not the same as CoT prompting. Figure 11 of the Appendix provides the full CoT output.

|  | LN-13b | LN-34b | GPT-4o | Gemini 2.5 Flash |
|---|---|---|---|---|
| without CoT | 0.33 | 0.38 | 0.62 | 0.66 |
| with CoT | 0.31 | 0.33 | 0.64 | 0.71 |

Table 4: Evaluation on AVSD-h with and without chain-of-thought (CoT) prompting. LN stands for LLaVA-NeXT. The first three results show that CoT is not helpful. Gemini 2.5 Flash is a reasoning model, and we can specify a "thinking budget." Flash without CoT was given a thinking budget of 0 tokens for reasoning, while Flash with CoT was given 1024. For Flash, reasoning does meaningfully improve the performance.

**Finding: VLMs are not robust against style changes.** The performance of Table 3 shows a gap between the performances on AVSD-s and AVSD-c, where the difference in the two sub-dataset is the style augmentations of AVSD-c via ControlNet. The gap is an

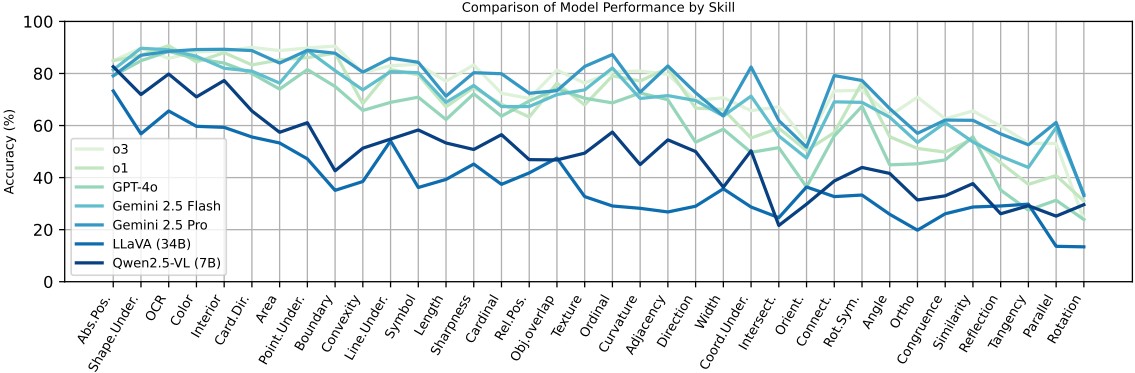

Figure 7: Evaluation results on AVSD-h across the skills. The skills are listed in the tendency of descending order of accuracy. We observe agreement among different models regarding which skills they found more or less challenging.

indication that the VLMs are not robust with respect to such style changes and that AVSD-c is effective at measuring this phenomenon. To show that this phenomenon is not merely a ControlNet artifact, we conduct a non-ControlNet style ablation using Qwen-Image-Edit-2511 (Wu et al. (2025)) in Appendix F.3.

## 5.2 Fine-tuning on atomic visual skills

The failure of VLMs on atomic visual skills is likely attributable to their training data. Vision encoders in VLMs may be pre-trained on datasets that prioritize natural images, although the information about the training data for proprietary and many open-weight models is not disclosed to the public. While such data may equip VLMs to effectively extract general scene-level information, it would not allow them to learn to understand precise geometric or scientific diagram features.

Thus, we hypothesize that the key to addressing this limitation lies in pre-training. Recent research on LLMs has shown that certain capabilities must be acquired during pre-training and cannot be easily learned through fine-tuning or post-training( Ovadia et al. (2024)). Similarly, geometric perception, which is qualitatively distinct from natural image scene understanding, should be explicitly incorporated into the pre-training process to be effectively learned.

As an indirect investigation of this question, we conducted a training experiment. We first fine-tuned LLaVA-Next-13B on the MathV360k dataset (Shi et al. (2024)), which primarily consists of composite geometry diagrams. Following the training recipe from the LLaVA repository (Liu et al. (2023)), we fine-tuned the VLM for 1–2 days on 8 H100 GPUs. Detailed hyperparameters are provided in Appendix E. We observed that additional training on composite diagrams alone did not improve the performance on AVSD. This suggests that training solely on composite geometry diagrams is insufficient to resolve the issue of atomic perception, at least in a data-efficient manner.

Next, we fine-tuned LLaVA-Next-13B on atomic perception tasks using the AVSD-s-train dataset, which contains 10,000 problems per skill, totaling 360,000 problems, matching the size of MathV360k. This training led to clear improvements, particularly on the handcrafted AVSD-h dataset, demonstrating a certain degree of out-of-distribution (OOD) generalization. Results are shown in Table 5.

While it is unsurprising that training on atomic visual skills improves performance on atomic visual skills, our results demonstrate that VLMs can indeed be trained to learn these capabilities. We hypothesize that incorporating such atomic perception data into the large-scale pre-training will lead to significant improvements in VLMs' geometric perception and reasoning abilities.

|        | LN-13b | + MathV360k | + AVSD-s |
|--------|--------|-------------|----------|
| AVSD-h | 0.33   | 0.31        | 0.45     |
| AVSD-s | 0.31   | 0.32        | 0.72     |
| AVSD-c | 0.23   | 0.24        | 0.55     |

Table 5: The performance of LLaVA-Next-13B on AVSD does not improve when further trained (fine-tuned) on the MathV360k dataset (Shi et al. (2024)), which contains mostly composite geometry diagrams. However, training LLaVA-Next-13B on the atomic perception tasks of the AVSD-s-train dataset leads to clear improvements.

## 6 Conclusion

In this work, we introduce the Atomic Visual Skills Dataset (AVSD) to evaluate VLMs on fundamental, indivisible geometric perception skills, which we refer to as atomic visual skills. Our findings show that state-of-the-art VLMs struggle with such skills. This limitation in accurately perceiving basic geometric features is concerning, as these capabilities are likely crucial for multimodal perception and reasoning tasks that require a precise understanding of the visual input. Enabling vision-language models to precisely perceive geometric diagrams and scientific figures would be valuable, significantly broadening the applicability of multimodal reasoning systems.

One limitation of this work is that we focus solely on 2D geometric perception. Expanding our approach to include 3D spatial reasoning would be an interesting direction for future research. Additionally, while our study focuses on geometric problems in mathematics, the ability of VLMs to precisely perceive diagrams and illustrations in scientific and everyday contexts would also be valuable. Such tasks may require a related but distinct set of atomic skills compared to the 36 skills we consider in this work. Exploring this broader scope presents another avenue for future work.

## 7 Limitations

AVSD is designed as a controlled diagnostic benchmark and therefore does not fully capture the complexity of real-world geometry diagrams. Textbook or classroom diagrams may include dense labels, auxiliary constructions, imperfect drawings, contextual clutter, and

perceptual ambiguities that are not fully represented in AVSD. To partially examine this gap, we conducted a small textbook-style validation using PGPS9K dataset (Zhang et al. (2023b)). AVSD also focuses on 2D geometric diagram perception and does not evaluate 3D geometric understanding, such as depth or projection. In addition, the easy/medium/hard labels in AVSD-h are author-defined difficulty indicators, not calibrated human difficulty estimates.

## 8 Broader Impact Statement

This work introduces the Atomic Visual Skills Dataset (AVSD), a benchmark designed to evaluate and analyze the fundamental perceptual capabilities of vision-language models (VLMs).

**Positive societal impacts.** The AVSD benchmark contributes to the development of more interpretable, reliable, and scientifically grounded multimodal models. Enhanced understanding of basic geometric perception could support education, visual reasoning research, and AI safety.

**Negative societal impacts.** Because AVSD is synthetic and composed of abstract mathematical diagrams and geometric figures–without any human, cultural, or sensitive content–it is not expected to pose ethical, privacy, or societal risks. The dataset does not involve personal data, and thus carries minimal potential for misuse.

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

## Appendix A. $\nu$-geometry

By the nature of geometry problems with visual inputs, accurate visual understanding is one of the most important prerequisites. To extensively measure the understanding of VLMs about plane geometry, we first introduce $\nu$-geometry, a novel dataset that visualizes a wide variety of plane-geometric diagrams. $\nu$-geometry consists of 1000 (image, question, answer) triplets, with two subtasks : True or False, and selecting the correct description of the image. The plane geometry in the image is generated by combining construction rules from our list of 77 rules, mostly motivated by AlphaGeometry (Trinh et al. (2024)).

**Motivation from AlphaGeometry.** As far as we know, the only existing VLM benchmarks that focus on plane geometric perception are : Euclid (Zhang et al. (2024a)) and Geoclidean (Hsu et al. (2022)). However, these tasks cover restricted portions of plane geometry; Euclids have 6 simple tasks, which is far less than our 36 atomic skills. Geoclidean offers infinitely many geometric situations based on code choices, but the tasks are purely non-verbal, thus not adequate to fully measure decoder-based VLMs such as LLaVA. Moreover, their geometric situations are mostly about recognizing intersections of points, lines, circles, and elementary polygons, and partially shape of elementary polygons, which are insufficient to illuminate diversity of plane geometry.

Therefore, we introduce $\nu$-**geometry**, a new dataset that aims to visually implement a dense subset of plane geometry. Our data generation is strongly motivated by AlphaGeometry (Trinh et al. (2024)), which consists of more than 50 construction rules, from adding fundamental shapes including point, circle, line, triangle, and square, to adding objects in complex relations including tangent line, midpoint, angle-trisector, incenter, and excenter. The dataset is rich and dense about plane geometry in that LLM trained on these data, together with symbolic prover, could solve plane geometry problems of all levels, up to the hardest problems from International Math Olympiad.

**Diagram construction.** To utilize the rich and diverse plane geometries within their construction, we implemented 47 rules from the AlphaGeometry rules. Additionally, we implemented new fundamental objects widely used in plane geometry, such as ovals or curves that are not closed. Lastly, motivated by Geoclidean (Hsu et al. (2022)), we added original rules that are more important when given in visual context, including positional relations, and rules about interaction of different objects, such as two circles and a line being tangent at one point. This resulted in 77 construction rules in total. Detailed descriptions of the construction rules are provided in Table A.

The diagram construction process is as follows. First, we start with an empty diagram. At each step, we randomly sample a construction rule from our 77 rules, and try adding a new feature on top of the current diagram following the sampled rule. If this is successful, repeat the step until the diagram undergoes a fixed number of construction steps. If this fails, leave the diagram unchanged and repeat another step.

Note that applying a construction rule with complex relations may sometimes fail, as the objects of the diagram are insufficient to implement the relation. Since this may cause imbalance among rules in data generation, we applied a simple trick of adding sufficient objects (but not counting as a step) when a complex rule is sampled.

Note that the complexity of a diagram can be controlled by using different number of construction steps. For rich complexity in $\nu$-geometry dataset, we constructed 50% of the images with one to three steps and 50% of the images with four to six steps. The examples of the visualization of our diagram constructions with different complexity are provided in Figure 14.

**Task design.** We introduce two tasks that measure the comprehension of VLMs about the constructed diagrams. To construct questions and answers about the diagram, we generated descriptions for each construction rule. When a rule is sampled for diagram construction, we also sample names to label new objects added to the diagram by the rule. Then we integrate the names and a (randomly chosen) description to complete a sentence that describes the diagram. Therefore, a collection of correct sentences of the diagram is naturally obtained as the diagram is constructed. Analogously, we can collect fake descriptions of the diagram, by (i) twisting correct descriptions, (ii) assigning descriptions from the rules unused in the diagram, and (iii) adding unused labels or permuting the used labels instead.

With the collections of correct and incorrect statements about the given diagram, we design two tasks with examples in Figure 2. The first task is the True or False question: to choose whether the given sentence is True or False. Second, given a correct sentence and an incorrect sentence, one should decide which sentence is correct.

**Visualization.** We applied some techniques to prevent the destruction of geometry by visualization process. For example, we forced the points to not be close to each other more than a certain threshold. In addition, we allowed lines with longer than certain threshold, to prevent them from looking like a point.

Combining all the process, we introduce a novel data generation pipeline that automatically generates tasks focusing on perception of plane geometry, together with visualization of diverse diagrams constructed by 77 rules motivated by AlphaGeometry. Generated with this pipeline, $\nu$-geometry consists of 1000 problems that effectively measure geometric perception of VLMs through the two tasks equally mixed. Note that by the design of both tasks, the random chance accuracy is exactly 0.5.

**Evaluation.** Since the ground truth for our tasks is straightforward–being either 'true' or 'false' (or choosing between '(i)' and '(ii)')–we evaluated the models by simply lower-casing their predictions and using exact string matching. Although we experimented with scoring via ChatGPT (OpenAI (2024a)), we found that string matching yielded more accurate results.

**Analysis.** These results indicate that the visual perception of current VLMs is overestimated when it comes to plane geometry. This shortfall motivates further investigation into methods for decomposing and identifying their perceptual abilities. We also remark that VLM training strategy, including data collection, should now focus more on geometric perception for an agent that generalized out of certain benchmarks and be able to handle geometry problems with complex visual inputs.

**List of construction rules.**

Table 6: Construction rules of $\nu$-geometry. $< n >$, where n is an integer, represents a label that will be assigned as each diagram is constructed.

| Construction Rule | Description |
| --- | --- |
| on_ellipse | Place point $<1>$ on the ellipse. |
| inside_cc1 | Place point $<3>$ in the intersection of circles $<1>$ and $<2>$. |
| inside_cc2 | Place point $<3>$ inside circle $<1>$ but outside circle $<2>$. |
| lll | Construct line $<1><2>$ and line $<1><3>$ meeting at point $<1>$. |
| ccl1 | Draw line $<3><4>$ connecting the point on circle $<1>$ ($<3>$) with the point on circle $<2>$ ($<4>$). |
| ccl2 | Place point $<3>$ on circle $<1>$ (and not on circle $<2>$) so that line $<3><4>$ is defined with point $<4>$. |
| ccl3 | Place points $<3>$ and $<4>$ inside circles $<1>$ and $<2>$ respectively, then join them to form line $<3><4>$. |
| convex_quad | Construct a convex quadrilateral with vertices $<2>$, $<3>$, $<4>$, and $<5>$. |
| centroid | Place point $<4>$ as the centroid of triangle $<1><2><3>$. |
| colinear | Ensure that points $<1>$, $<2>$, and $<3>$ are collinear. |
| eqangle3 | Set angle $<2><1><6>$ equal to angle $<3><4><5>$. |
| one_line_one_circle | Place point $<2>$ on circle $<1>$ and make circle $<4>$ tangent to both circle $<1>$ and line $<2><3>$. |
| two_lines_one_circle | Draw point $<2>$ on circle $<1>$; construct line $<2><3>$ tangent to circle $<4>$ at $<5>$, line $<3><8>$ tangent to circle $<4>$ at $<6>$, and ensure circles $<1>$ and $<4>$ are tangent at $<7>$. |
| on_dia | Make segment $<3><1>$ perpendicular to segment $<3><2>$. |
| trisect | Place points $<4>$ and $<5>$ on line $<1><3>$ to trisect angle $<1><2><3>$. |
| rotate_angle | Set angle $<2><1><3>$ to measure $<4>$. |
| risos | Construct right isosceles triangle $<1><2><3>$ with legs $<1><3>$ equal to $<1><2>$. |
| r_trapezoid | Construct right trapezoid $<1><2><3><4>$ with side $<1><2>$ perpendicular to sides $<1><4>$ and $<2><3>$. |

| Construction Rule | Description |
| --- | --- |
| shift1 | Ensure segments <1><2>, <3><4>, and <5> are equal, and segments <1><4>, <2><3>, and <6> are equal. |
| shift2 | Set segment <1><2> equal to segment <3><4> and segment <1><4> equal to segment <2><3>. |
| orthocenter | Place point <4> as the orthocenter of triangle <1><2><3>. |
| excenter | Place point <4> as the excenter of triangle <1><2><3> opposite vertex <1>. |
| midpointcircle | Mark points <4>, <5>, and <6> as the midpoints of triangle <1><2><3>'s sides and set point <7> as the circumcenter of triangle <4><5><6>. |
| on_circle2 | Place point <4> on circle <1>. |
| on_tline | Construct line <1><4> perpendicular to line <2><3>. |
| eqangle2 | Set angle <2><1><4> equal to angle <4><3><2> as indicated by <5>. |
| rotate90 | Rotate point <1> by 90° about point <2> to obtain point <3>. |
| line | Draw a line connecting points <1> and <2>. |
| labelled_line | Draw line <1><2> and label it with label <3>. |
| circle | Draw a circle centered at <1> with an unspecified radius. |
| infinite_line | Draw an infinite line passing through point <1>. |
| triangle | Draw triangle <1><2><3>. |
| angle_bisector | Construct line <2><4> that bisects angle <1><2><3>. |
| circle_center | Place point <4> as the circumcenter of the circumscribed circle of triangle <1><2><3>. |
| eq_quadrilateral | In quadrilateral <1><2><3><4>, ensure side <2><3> equals side <4><1>. |
| eq_trapezoid | Construct trapezoid <1><2><3><4> with all sides equal. |
| equilateral_triangle | Draw an equilateral triangle <1><2><3> with all sides equal. |
| isosceles_triangle | Construct isosceles triangle <1><2><3> with sides <1><2> and <2><3> equal. |
| eqdia | Ensure in quadrilateral <1><2><3><4> that diagonal <1><3> equals diagonal <2><4>. |
| circle_proj | Project point <1> onto circle <2> to obtain point <4>. |

| Construction Rule | Description |
| --- | --- |
| `mirror` | Reflect point <1> across point <2> to get point <3>. |
| `right_iso` | Construct right isosceles triangle <1><2><3> with sides <1><2> and <1><3> equal. |
| `on_bline` | Place point <3> on the perpendicular bisector of segment <1><2>. |
| `on_circle_with_r` | Draw a circle with center <1> and radius measured by <1><2> (length <4>) that passes through point <3>. |
| `on_circle` | Draw a circle centered at <1> that passes through point <3>. |
| `on_line` | Place point <3> on the line through points <1> and <2>. |
| `on_pline` | Construct lines <1><2> and <3><4> so that they are parallel. |
| `parallelogram` | Draw parallelogram <1><2><3><4>. |
| `pentagon` | Draw pentagon <1><2><3><4><5>. |
| `trapezoid` | Draw trapezoid <1><2><3><4>. |
| `r_triangle` | Draw right triangle <1><2><3> with the right angle at vertex <1>. |
| `rectangle` | Draw rectangle <1><2><3><4>. |
| `reflect` | Reflect point <1> across line <2><3> to obtain point <4>. |
| `square` | Draw square <1><2><3><4>. |
| `triangle12` | Construct triangle <1><2><3> with side ratio <1><2> : <2><3> = 1:<4>. |
| `inf_tangent` | Draw an infinite tangent line to the circle centered at <1> at point <2>. |
| `tangent` | Draw line <3><4> tangent to circle <1> at point <2>. |
| `trisect` | Trisect angle <5><1><2> by drawing lines <1><3> and <1><4>. |
| `trisegment` | Divide segment <1><2> into three equal parts by placing points <3> and <4>. |
| `c_tangent_with_r` | Draw a circle with center <1> and radius <2>, and from point <5> draw two tangents touching it at <3> and <4>. |
| `c_tangent` | Draw a circle centered at <1> and from point <4> draw two tangents touching it at <2> and <3>. |
| `cc_tangent` | Draw circles with centers <1> and <2>, and construct their common tangents <3><5> and <4><5>. |

| Construction Rule | Description |
| --- | --- |
| cc_tangent_with_r | Draw circles with center <1> (radius <2>) and center <3> (radius <4>), and construct common tangents <5><7> and <6><7>. |
| cc_tangent_one | Draw line <5><6> tangent to both circles <1> and <3>. |
| parallel | Ensure lines <1><2> and <3><4> are parallel. |
| intersect_ll | Draw lines <1><2> and <3><4> intersecting at point <5>. |
| intersect_cl | Construct circle <1> and line <5><6> intersecting at points <3> and <4>. |
| intersect_cc | Draw circles <1> and <3> intersecting at points <5> and <6>. |
| touches_cc | Construct circles <1> and <3> that touch at point <5>. |
| touches_clc | Draw circles <1> and <3> tangent at point <7> on line <5><6>. |
| touches_cc2 | Construct circle <1> surrounding circle <3> so that they touch at point <5>. |
| touches_clc2 | Draw a line tangent to both circles <1> and <3>, touching them at points <5> and <6> respectively. |
| rhombus | Draw a rhombus <1><3><2><4> with all sides equal. |
| quadrilateral | Draw quadrilateral <1><2><3><4>. |
| convex_quadrilateral | Construct convex quadrilateral <1><2><3><4>. |
| connect_center | Draw circles with centers <1> and <3> and connect the centers with a line. |
| parab_line_intersect | Draw a parabola and a line intersecting at points <1> and <2>. |
| ellipse_line_intersect | Draw an ellipse and a line intersecting at points <1> and <2>. |
| random_curve | Draw a random curve without a specific classification. |
| circle_with_radius | Draw a circle with center <1> and radius <2>. |
| circular_sector_with_rad | Draw a circular sector with center <1>, radius <2>, and central angle <3>. |
| circular_sector | Draw a circular sector with center <1> and angle <2>. |
| semicircle_with_radius | Draw a semicircle with center <1> and radius <2>. |
| semicircle | Draw a semicircle with center <1>. |

| Construction Rule | Description |
|---|---|
| l_in_c | Ensure line segment <3><4> is entirely contained within circle <1>. |
| l_out_c | Place line segment <3><4> completely outside circle <1>. |
| ll_angle | Construct lines <1><2> and <3><4> intersecting at point <5> with an angle of <6>°. |
| ellipse | Draw an ellipse with foci at <1> and <2>. |

## Appendix B. Further details on atomic visual skills

In this section, we provide detailed definitions of the 36 atomic visual skills, together with corresponding problem samples from AVSD-h. Before presenting the individual skills, we restate our criterion for atomicity. In AVSD, we treat a skill as atomic when the question directly asks for a single target visual judgment about the diagram, for example, whether two lines are parallel, or which angle is marked. Such a judgment may depend on lower-level visual cues, including points, lines, curves, colors, and labels, but the task does not require the solver to combine multiple relations, manipulate symbolic expressions, or apply geometric theorems.

1. `Angle` is a skill to understand how an angle is visually represented. Angle is the primary factor in how a polygon looks, how two or more objects are related, and many other situations.

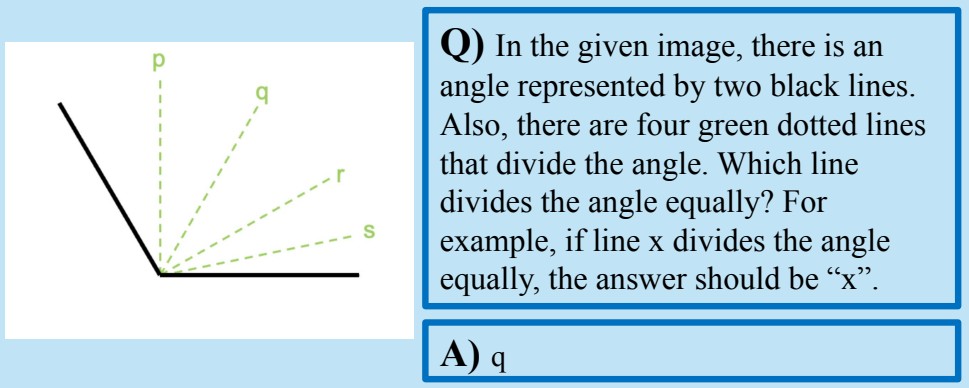

**Q)** In the given image, there is an angle represented by two black lines. Also, there are four green dotted lines that divide the angle. Which line divides the angle equally? For example, if line x divides the angle equally, the answer should be "x".

**A)** q

2. `Direction` is an ability to recognize linear direction in an image. It is a fundamental skill in human vision, supporting representation of linearity and multi-dimensional relations.

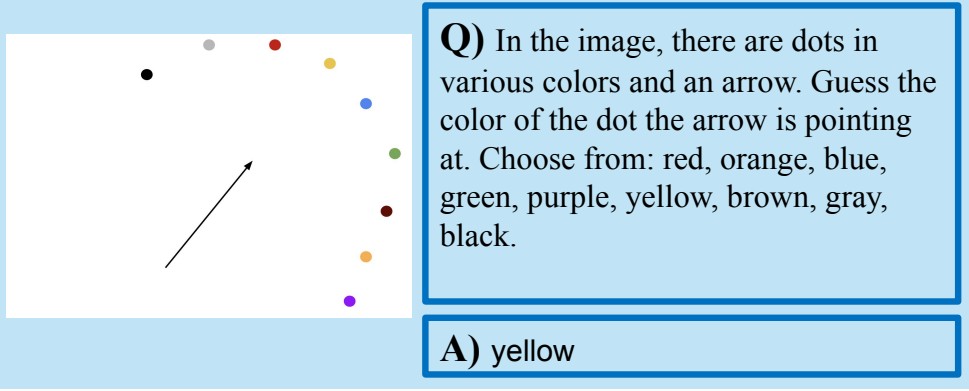

**Q)** In the image, there are dots in various colors and an arrow. Guess the color of the dot the arrow is pointing at. Choose from: red, orange, blue, green, purple, yellow, brown, gray, black.

**A)** yellow

3. `Boundary` is a skill to understand the ends of objects or areas, and to detect visual representation of edges. The skill is used in distinguishing between distinct objects, or detecting boundaries between spaces.

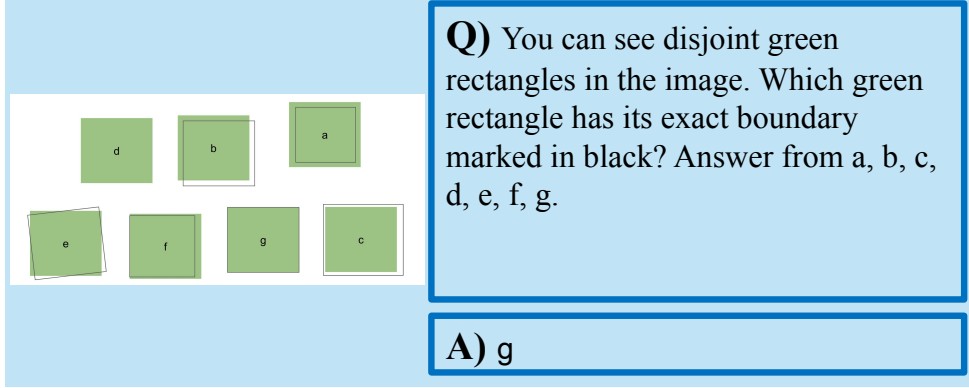

**Q)** You can see disjoint green rectangles in the image. Which green rectangle has its exact boundary marked in black? Answer from a, b, c, d, e, f, g.

**A)** g

4. **Cardinal** is a field about counting distinct objects or specified concepts. Mastery of cardinals implies measuring quantities or dealing with multiple objects. Especially, it should take into account everything that satisfies given conditions, giving a difference from the skill of understanding *Ordinals*.

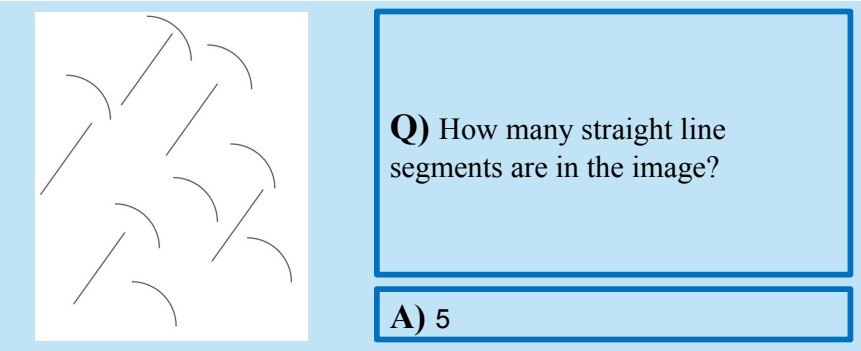

**Q)** How many straight line segments are in the image?

**A)** 5

5. **Congruence** is a skill of detecting objects with the exact same scale and shape, and understanding their correspondence. Congruence is a primary component of visualizing various symmetries including translation, rotation or flipping. Congruence is distinguished from other equivalence because it requires the objects to be equal at all levels of measurement.

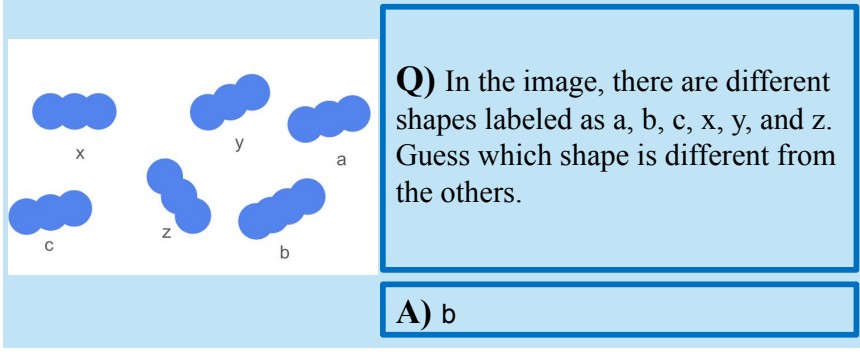

**Q)** In the image, there are different shapes labeled as a, b, c, x, y, and z. Guess which shape is different from the others.

**A)** b

6. **Convexity** is a skill of understanding convexity of given shapes. The skill is also closely related to detecting bumps or indentations and understanding convex and concave functions.

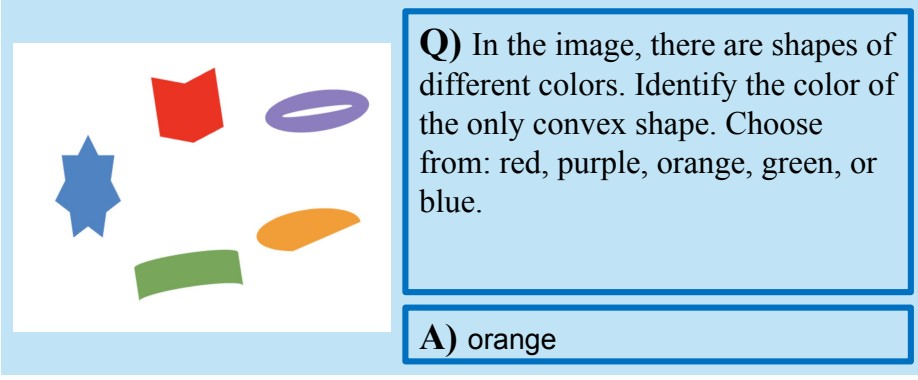

**Q)** In the image, there are shapes of different colors. Identify the color of the only convex shape. Choose from: red, purple, orange, green, or blue.

**A)** orange

7. `Intersection` is a mastery of detecting intersections of lines and curves. The skill is necessary for interpreting relationships among 1-dimensional objects, and also among higher dimensional objects from 1-dimensional representations of their boundaries.

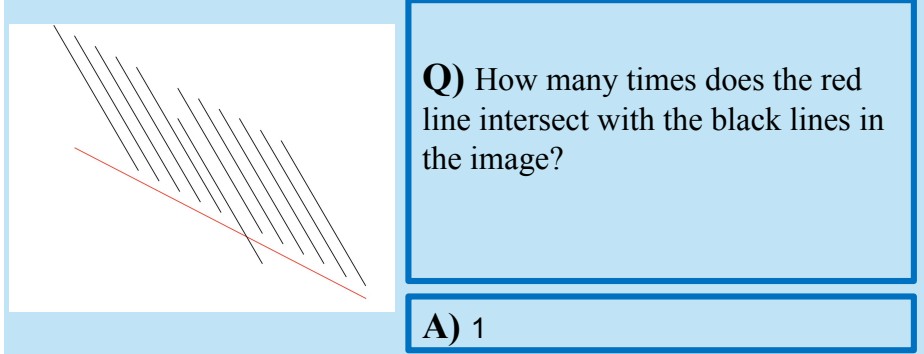

**Q)** How many times does the red line intersect with the black lines in the image?

**A)** 1

8. `Line` is a skill to detect line segments and understand their roles in the image. This skill is a fundamental unit in understanding various objects as polygons, graphs and diagrams.

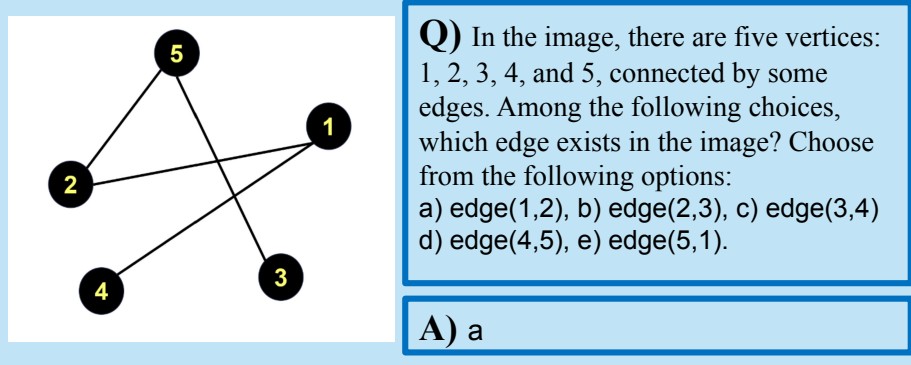

**Q)** In the image, there are five vertices: 1, 2, 3, 4, and 5, connected by some edges. Among the following choices, which edge exists in the image? Choose from the following options:
a) edge(1,2), b) edge(2,3), c) edge(3,4) d) edge(4,5), e) edge(5,1).

**A)** a

9. `OCR` is a skill to detect and read characters from visual inputs.

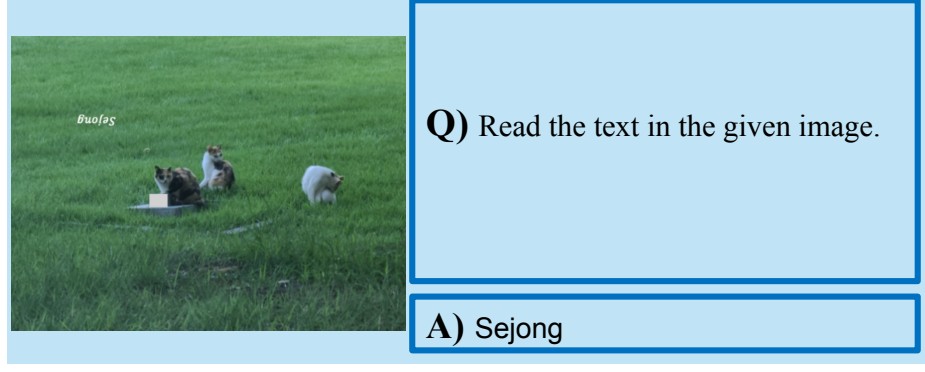

**Q)** Read the text in the given image.

**A)** Sejong

10. `Ordinal` is a skill to count certain objects or concepts in a given order. Mastery of this skill requires not just counting but also focusing on specific portions and order of targets, giving a difference from Cardinal Understanding.

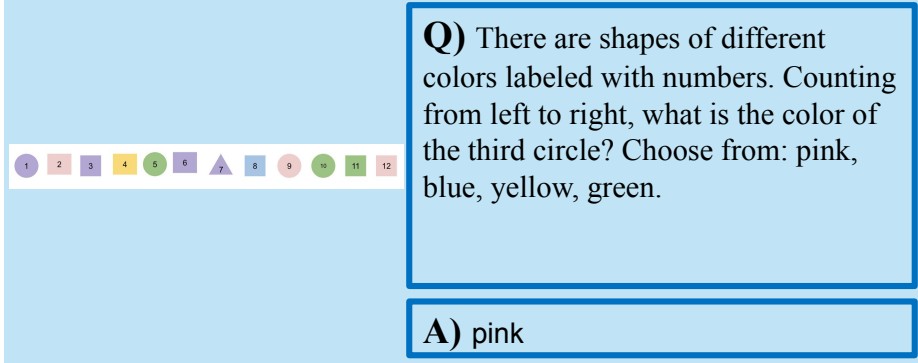

**Q)** There are shapes of different colors labeled with numbers. Counting from left to right, what is the color of the third circle? Choose from: pink, blue, yellow, green.

**A)** pink

11. `Overlap` skill is about correctly recognizing two or more objects sharing a common area. The skill is crucial in understanding overlapping shapes or complex shapes such as diagrams.

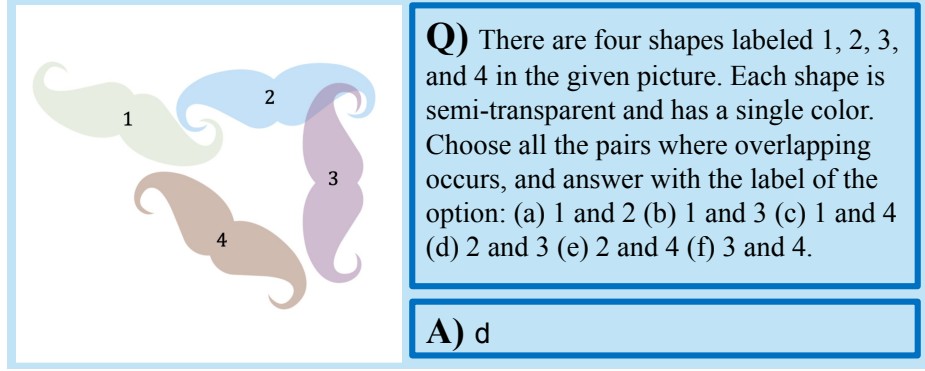

**Q)** There are four shapes labeled 1, 2, 3, and 4 in the given picture. Each shape is semi-transparent and has a single color. Choose all the pairs where overlapping occurs, and answer with the label of the option: (a) 1 and 2 (b) 1 and 3 (c) 1 and 4 (d) 2 and 3 (e) 2 and 4 (f) 3 and 4.

**A)** d

12. `Interior` is a skill of distinguishing between interior and exterior of the target area. This skill is essential in perceiving different areas.

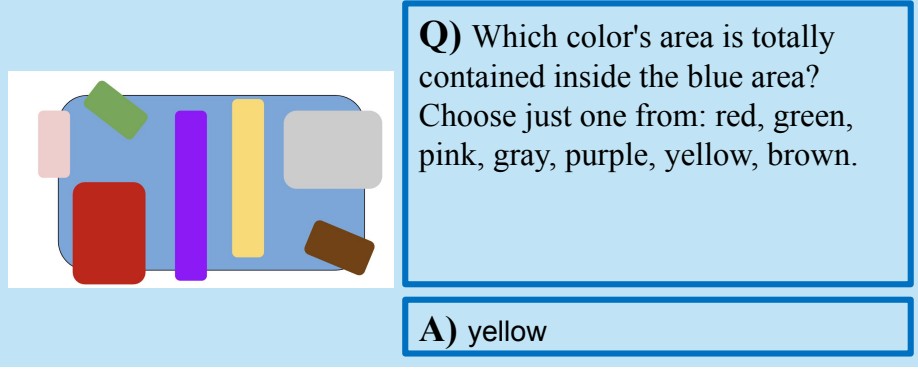

**Q)** Which color's area is totally contained inside the blue area? Choose just one from: red, green, pink, gray, purple, yellow, brown.

**A)** yellow

13. `Relative Position` is an ability to identify positional relationships between objects that cannot be simply described such as inside, outside, or moved in a certain direction. This skill requires comprehension of complex relationships such as "positioned in between," or "at the same side of."

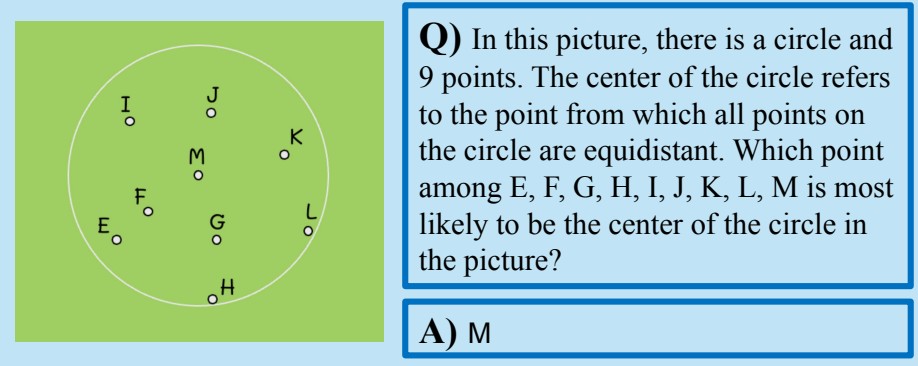

**Q)** In this picture, there is a circle and 9 points. The center of the circle refers to the point from which all points on the circle are equidistant. Which point among E, F, G, H, I, J, K, L, M is most likely to be the center of the circle in the picture?

**A)** M

14. `Reflection` is a field of recognizing linear symmetries. It requires detecting the axis of reflection and induced correspondence of objects.

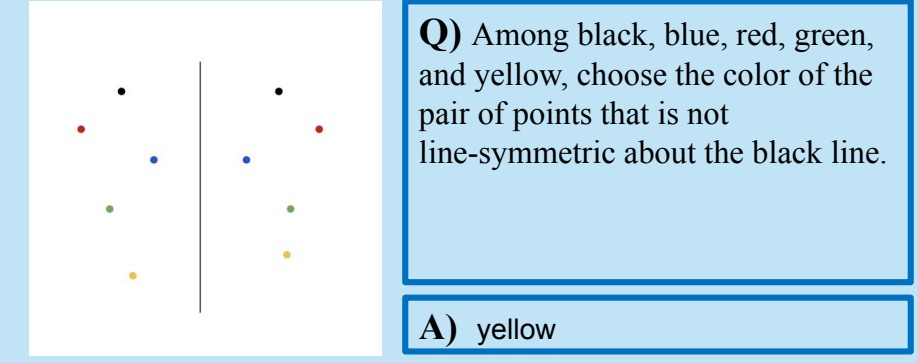

**Q)** Among black, blue, red, green, and yellow, choose the color of the pair of points that is not line-symmetric about the black line.

**A)** yellow

15. `Length` is a skill to handle lengths of different objects. It involves comparing different lengths and measuring distances of objects.

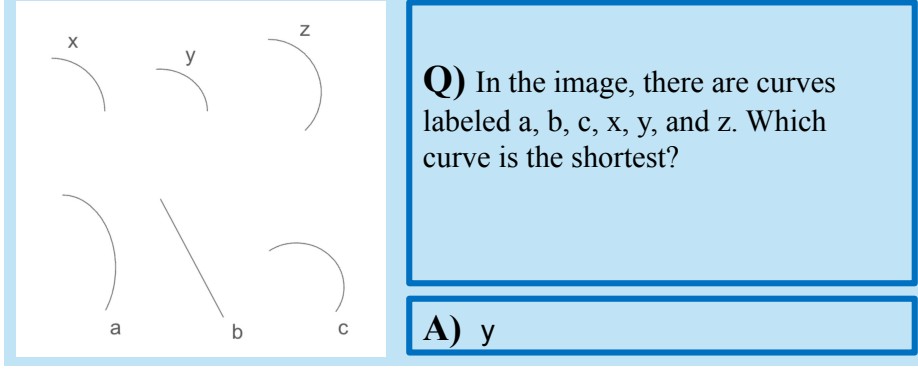

**Q)** In the image, there are curves labeled a, b, c, x, y, and z. Which curve is the shortest?

**A)** y

16. `Rotation` is an ability to identify changes in positions and angles induced by rotation, and detecting the axis of rotation.

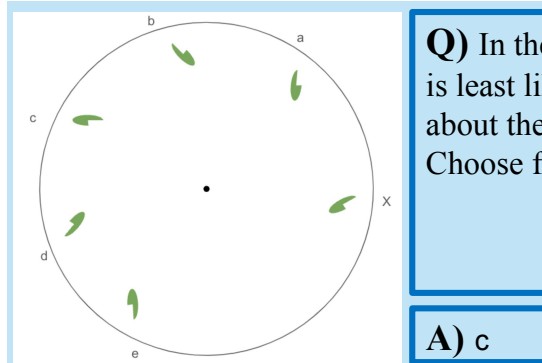

**Q)** In the image, which green shape is least likely to be a rotation of X about the center of the circle? Choose from: a, b, c, d, or e.

**A)** c

17. `Rotational Symmetry` is a field of symmetric representations with respect to rotations. The skill involves understanding invariant geometric features under specified rotations.

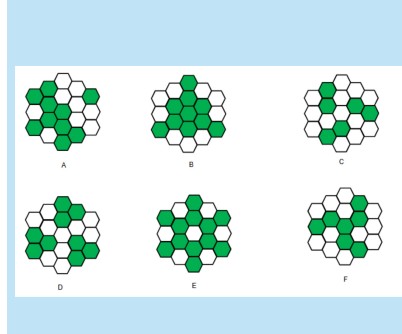

**Q)** Among shapes A, B, C, D, E, and F, which one does NOT have any 3-fold rotational symmetry?

**A)** A

18. `Symbol` is a skill to detect symbols, understand their roles in the image, and combine them with other visual information to attain the correct interpretation of the image.

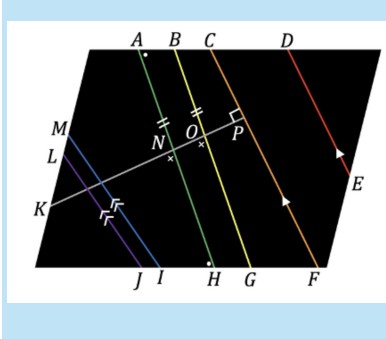

**Q)** Choose all the correct meanings of the symbols in the given picture.
(1) The angle JHA has the same size as the angle DAN.
(2) The angle KNH has the same size as the angle KOB.
(3) Two lines MI and CF are parallel.
(4) Two lines NP and CF are perpendicular.

**A)** 1,4

19. **Texture** is a skill to understand textures of objects in the image. The skill is essential as texture is another main component of visual representation of objects, and is used to distinguish different objects with same shapes, such as line styles.

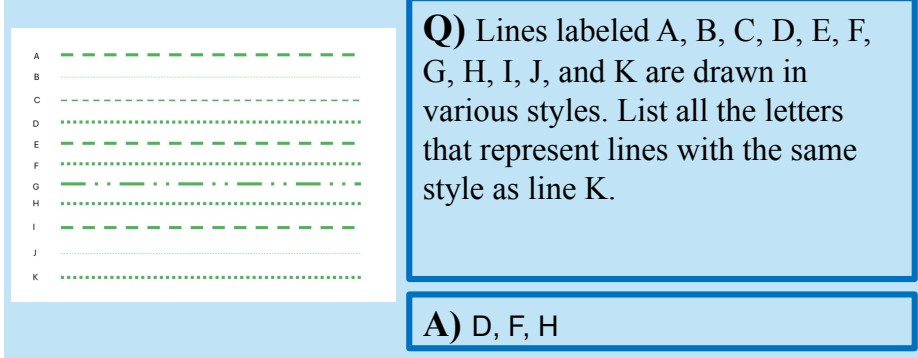

**Q)** Lines labeled A, B, C, D, E, F, G, H, I, J, and K are drawn in various styles. List all the letters that represent lines with the same style as line K.

**A)** D, F, H

20. **Width** is a skill to understand thickness and width of objects or areas. The skill is essential in measuring area or proportion of images together with length understanding.

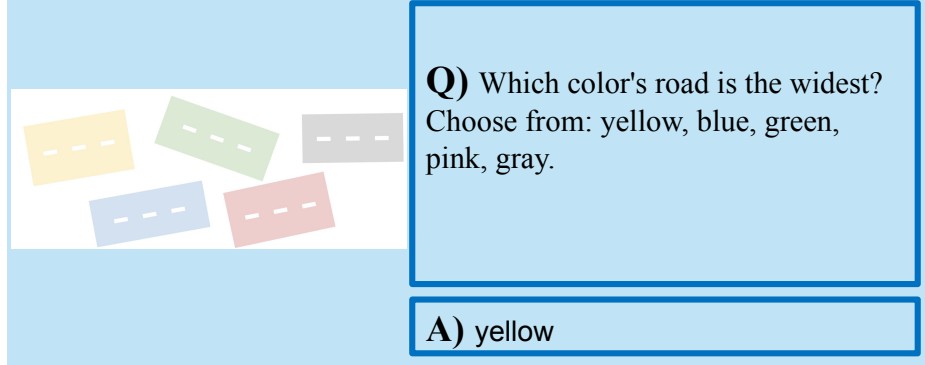

**Q)** Which color's road is the widest? Choose from: yellow, blue, green, pink, gray.

**A)** yellow

21. **Adjacency** is a skill to recognize when two or more objects are next to each other. The skill is crucial in understanding features induced by close positions such as forming clusters.

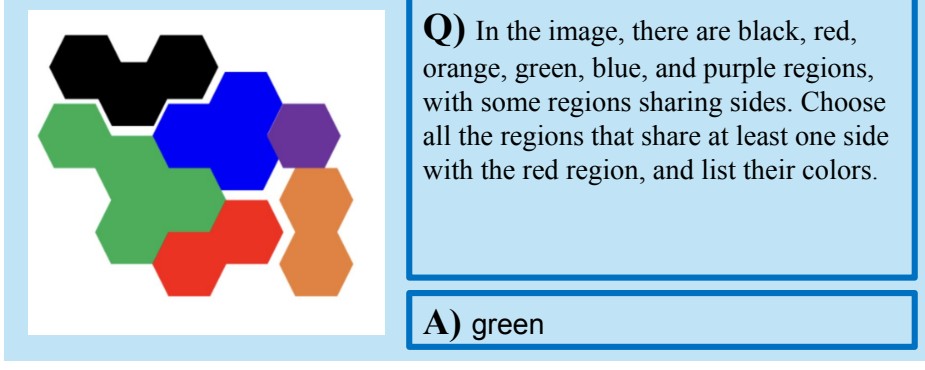

**Q)** In the image, there are black, red, orange, green, blue, and purple regions, with some regions sharing sides. Choose all the regions that share at least one side with the red region, and list their colors.

**A)** green

22. `Absolute Position` is a skill to correctly understand where the objects are represented as a part of the visual input, independently of other objects. This involves recognizing objects posited at corners of an image, or comparing heights of objects represented in the image.

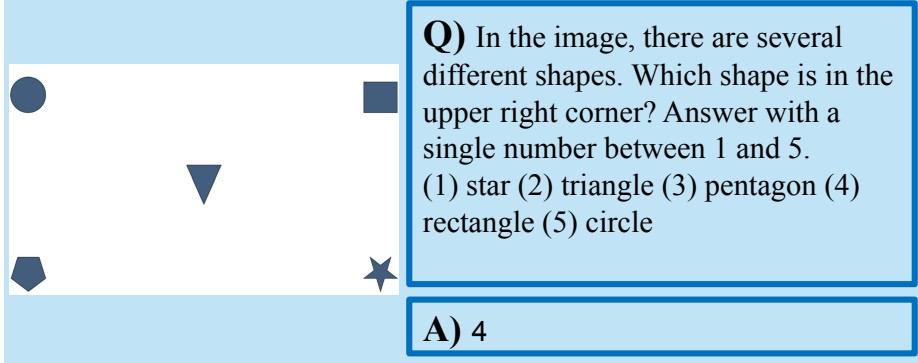

**Q)** In the image, there are several different shapes. Which shape is in the upper right corner? Answer with a single number between 1 and 5.
(1) star (2) triangle (3) pentagon (4) rectangle (5) circle

**A)** 4

23. `Area` is a skill to handle 2-dimensional volumes, including comparing areas.

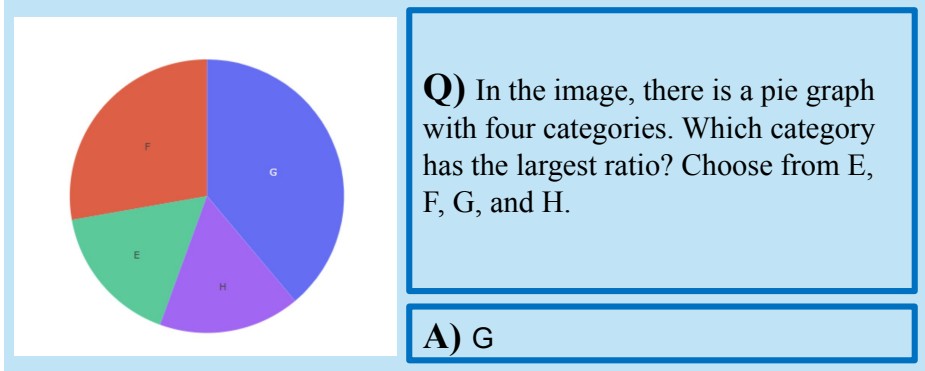

**Q)** In the image, there is a pie graph with four categories. Which category has the largest ratio? Choose from E, F, G, and H.

**A)** G

24. `Cardinal Direction` is a skill to understand primary directions including up, down, left, right, or diagonals. This involves recognizing North, South, West, and East directions.

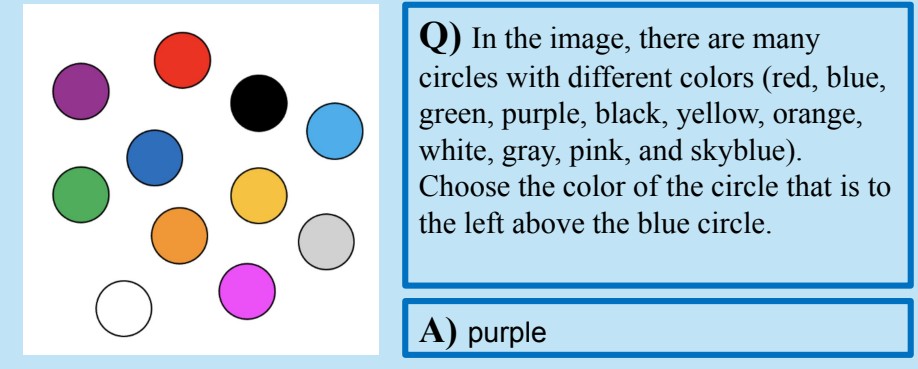

**Q)** In the image, there are many circles with different colors (red, blue, green, purple, black, yellow, orange, white, gray, pink, and skyblue). Choose the color of the circle that is to the left above the blue circle.

**A)** purple

25. **Orthogonality** is a skill to identify orthogonal relations of objects in the image, including a right angle formed by two lines. Understanding orthogonality is fundamental in geometry, design, and engineering.

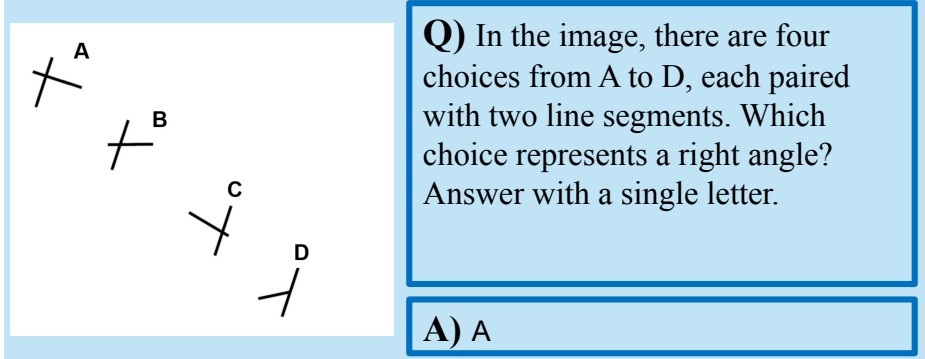

**Q)** In the image, there are four choices from A to D, each paired with two line segments. Which choice represents a right angle? Answer with a single letter.

**A)** A

26. **Tangency** is a skill to detect tangent objects. This skill focuses on geometric representation of tangent curves or boundaries, and is different from understanding adjacency that rather focuses on positional information.

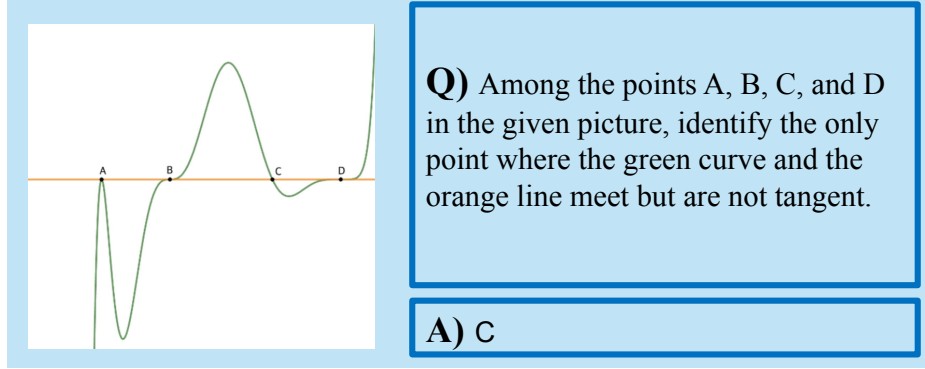

**Q)** Among the points A, B, C, and D in the given picture, identify the only point where the green curve and the orange line meet but are not tangent.

**A)** C

27. **Connectedness** is a skill to identify connected components and detect links between objects. This is crucial in understanding interactions and distinguishing distinct components.

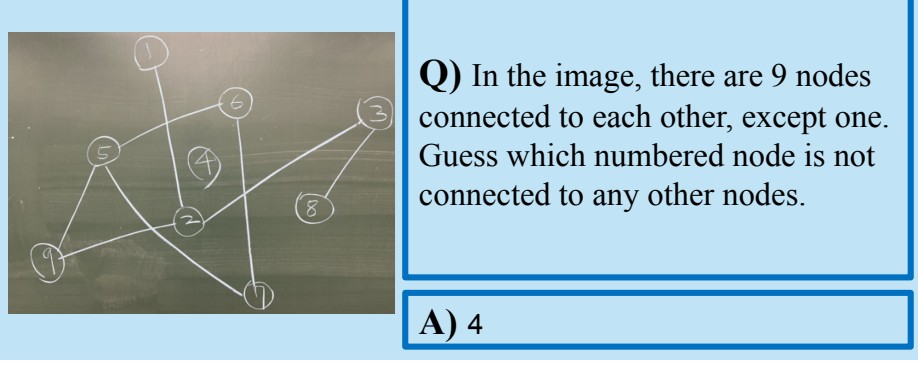

**Q)** In the image, there are 9 nodes connected to each other, except one. Guess which numbered node is not connected to any other nodes.

**A)** 4

28. `Parallel` is a skill to recognize parallel lines or curves. This is essential in identifying fundamental objects like squares.

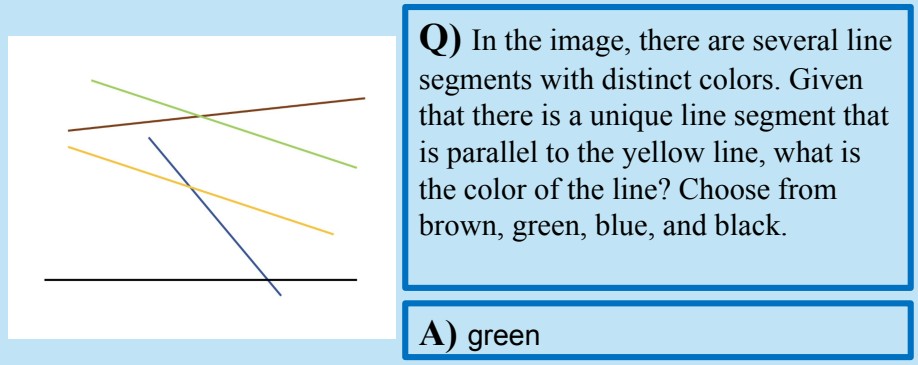

**Q)** In the image, there are several line segments with distinct colors. Given that there is a unique line segment that is parallel to the yellow line, what is the color of the line? Choose from brown, green, blue, and black.

**A)** green

29. `Similarity` is a skill to understand equivalence of geometric representations independent of scale. It also involves understanding of rescaling or comparing aspect ratios.

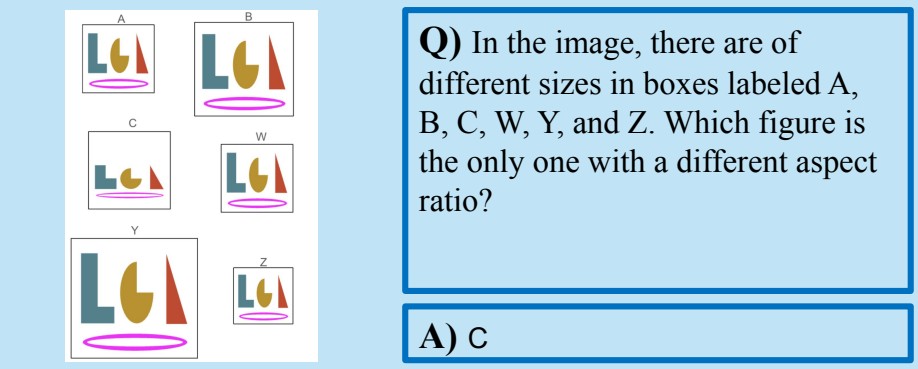

**Q)** In the image, there are of different sizes in boxes labeled A, B, C, W, Y, and Z. Which figure is the only one with a different aspect ratio?

**A)** C

30. `Color` is an ability to perceive, distinguish different colors, and understand the change in saturation and brightness.

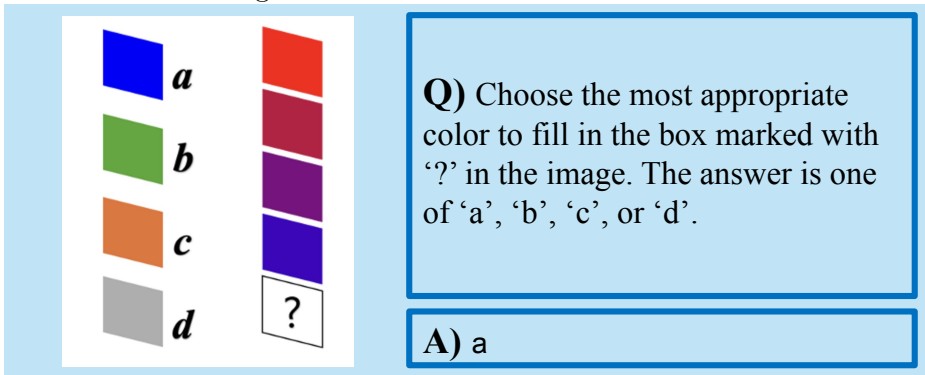

**Q)** Choose the most appropriate color to fill in the box marked with '?' in the image. The answer is one of 'a', 'b', 'c', or 'd'.

**A)** a

31. `Coordinate` is a skill to recognize and acquire correct information upon coordinate systems. We provide and acquire information about different systems such as polar coordinates.

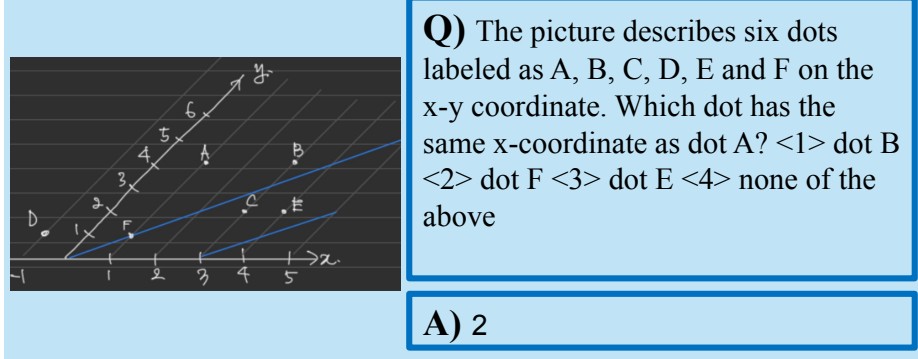

**Q)** The picture describes six dots labeled as A, B, C, D, E and F on the x-y coordinate. Which dot has the same x-coordinate as dot A? <1> dot B <2> dot F <3> dot E <4> none of the above

**A)** 2

32. `Point` is a fundamental capability to detect points and understand their roles in the image. It also involves understanding nodes in different graphs.

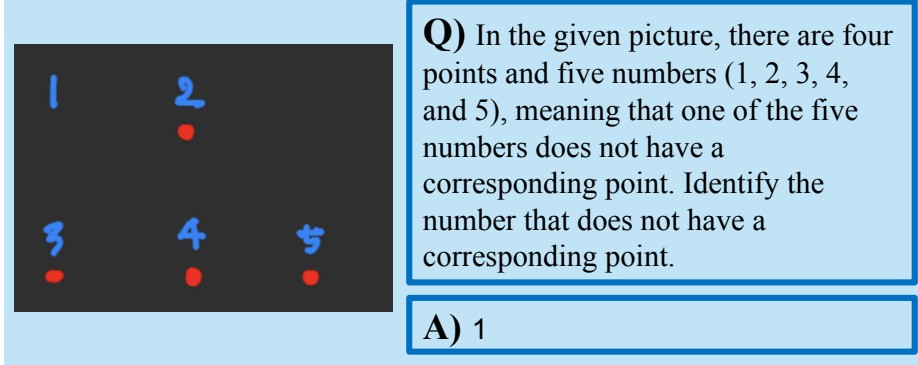

**Q)** In the given picture, there are four points and five numbers (1, 2, 3, 4, and 5), meaning that one of the five numbers does not have a corresponding point. Identify the number that does not have a corresponding point.

**A)** 1

33. `Shape` is a skill to understand details of shapes and compare different shapes independently of positions or tilts. It also involves identifying popular shapes such as triangles, rectangles, circles, and stars.

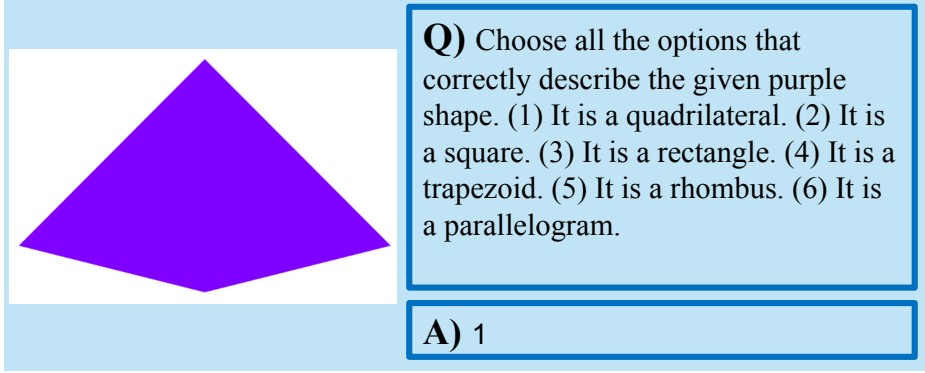

**Q)** Choose all the options that correctly describe the given purple shape. (1) It is a quadrilateral. (2) It is a square. (3) It is a rectangle. (4) It is a trapezoid. (5) It is a rhombus. (6) It is a parallelogram.

**A)** 1

34. **Curvature** is an ability to measure and compare curvatures of different curves. This involves distinguishing between straight lines and wavy curves, and detecting bends in a shape.

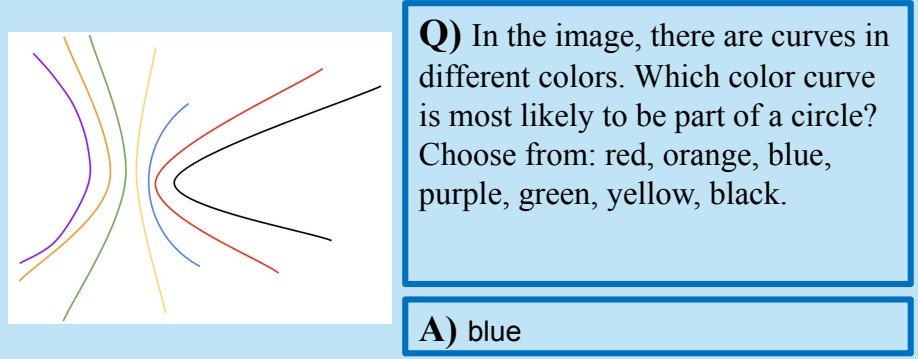

**Q)** In the image, there are curves in different colors. Which color curve is most likely to be part of a circle? Choose from: red, orange, blue, purple, green, yellow, black.

**A)** blue

35. **Sharpness** is a skill to detect pointy parts of a shape. This is essential in understanding the representations of non-smooth objects such as points of a function that are not differentiable.

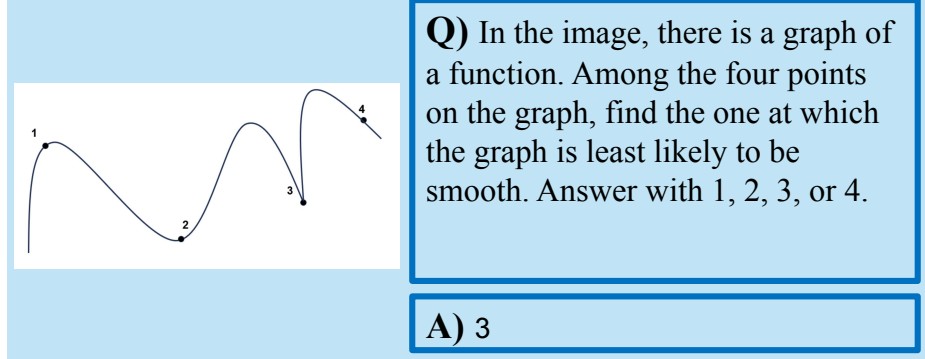

**Q)** In the image, there is a graph of a function. Among the four points on the graph, find the one at which the graph is least likely to be smooth. Answer with 1, 2, 3, or 4.

**A)** 3

36. **Orientation** is a skill to correctly distinguish clockwise and counterclockwise tendencies induced by not only rotations but also other movements that result in clockwise and counterclockwise directional change. The name originated from the mathematical definition of orientation in differential geometry.

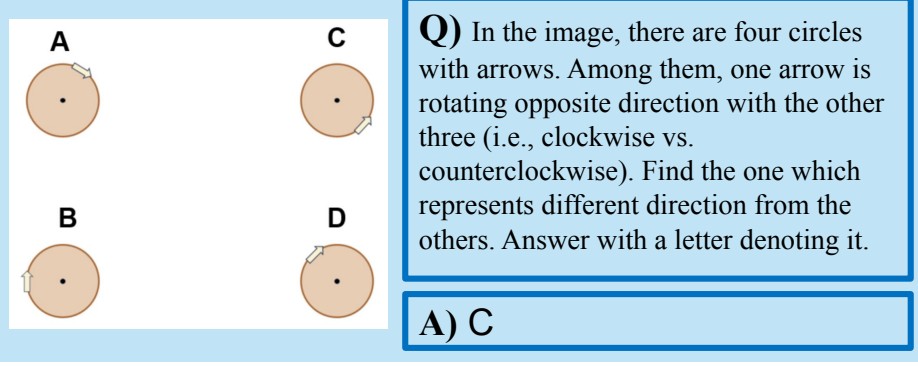

**Q)** In the image, there are four circles with arrows. Among them, one arrow is rotating opposite direction with the other three (i.e., clockwise vs. counterclockwise). Find the one which represents different direction from the others. Answer with a letter denoting it.

**A)** C

| Target skill | Possible lower-level cues | How AVSD isolates the target judgment |
|---|---|---|
| Angle | Vertex localization and line or ray localization | Angles are direct geometric objects in diagrams and are treated as single queryable units in elementary geometry. |
| Tangency | Curve localization and local contact judgment | The relevant objects are visually explicit, and the task focuses on whether they touch at exactly one point rather than on theorem application. |
| Parallel | Line detection and orientation comparison | Lines are presented in controlled configurations so that the question targets the visual relation of parallelism. |
| Symbol | Symbol detection and spatial association with nearby geometry | The task evaluates whether the model can identify diagram notation and bind it to the correct geometric object or relation. |
| Congruence and Similarity | Shape localization, scale comparison, and correspondence | These skills capture common equivalence relations that are visually judged before downstream symbolic reasoning begins. |

Table 7: Representative dependencies among AVSD skills. "Atomic" denotes the level at which AVSD evaluates a diagnostic visual judgment, not a claim that the skill has no lower-level perceptual prerequisites.

**Operational meaning of atomic visual skills.** We use the term *atomic* in an operational sense, rather than as a claim of formal psychological or mathematical indivisibility. In this work, a visual skill is treated as atomic if it is intuitive for human readers familiar with elementary geometry, can be evaluated with minimal task context, and is useful as a primitive diagnostic unit for understanding high-school-level 2D geometric diagrams. This does not mean that the skill has no lower-level perceptual prerequisites. For example, recognizing an angle requires localizing its vertex and two sides, and recognizing tangency requires localizing the relevant curves and judging whether they touch at exactly one point. Our claim is therefore not that these skills are irreducible in all possible perceptual models, but that they are useful diagnostic units at the level of geometric diagram understanding targeted by AVSD. The goal is to evaluate whether VLMs can perform these basic visual competencies before they are composed into more complex mathematical reasoning.

## Appendix C. Further details on AVSD

In this section, we introduce more details of AVSD-h, AVSD-s, and AVSD-c. We also provide examples from AVSD in Figures 16, 17, and 18. In Section C.4, we summarize our strategies for quality control.

### C.1 Further details on AVSD-h

**Difficulty details.** The problems of AVSD-h are classified into "easy," "medium," and "hard." We define each difficulty level with the following criteria:

1. Easy problems can be solved in a blink of an eye (Fu et al. (2024)).

2. Medium problems can be solved in 10 seconds.

3. Hard problems can be solved within a minute, and can be instantly verified when the answer is given.

Each problem was written by one person and independently evaluated by three other people. A problem was accepted if all three evaluators were able to solve it and if the three evaluators agreed on the difficulty level.

### C.2 Further details on AVSD-s

AVSD-s allows the generation of an infinite number of unique visual question-answer pairs for each of 36 skills in AVSD-h. For each skill, we designed an average of 11.6 task types, each specifically targeting different aspects of the skill. We mostly aimed for each synthetic task to depict some questions in the handcrafted dataset (AVSD-h). The visual component of each task involves the procedural addition of random geometric elements to a plain diagram. These geometric elements include points, lines, and polygons, along with paired simple styles (e.g., dashed/dotted lines), colors, and text labels. The position, scale, and shapes of the elements are determined by sampling random variables parameterizing them. The language question-answer component of each task consists of at least five different rephrased formats, while each format is parametrized based on the text labels in the diagram.

For example, task types related to the 'Length' skill include comparing the lengths of given lines, identifying the longest and shortest lines, comparing a line with a circle's radius, and comparing various distances. The length of each line, the radius of each circle, and the distance between diagrams are continuous parameters, with labels indicating them randomly chosen from distinct alphabets. Each problem in the dataset is then generated by selecting an available task type with uniform probability. We summarize other task types in Table 16 and provide further details along with code for sampling them.

### C.3 Further details on AVSD-c

**Model choice.** While the original ControlNet architecture was fine-tuned with the Stable Diffusion 1.5 model, Flux diffusion models are empirically known to be capable of taking more detailed input prompts in the format of natural language compared to the Stable

Diffusion 1.5 model, which makes it better suited for our purpose of reconstructing complex geometry images. Hence, we mainly use the ControlNet fine-tuned with Flux models (InstantX (2024)).

ControlNet's pipeline includes extracting the information of the input image to condition the image generation alongside the user's input prompt. Although there are many variants, we use the Canny Edge detection algorithm. For images that primarily consist of straight lines, algorithms like M-LSD straight-line detection might work better, but we have empirically verified that it does not make any difference in the quality of the generated images.

**Overall pipeline.** The main challenge in utilizing ControlNet in our pipeline lies in (i) enforcing the model to generate everything outlined by the canny edges and (ii) ensuring that the models do not generate any other details that can hurt the quality of the images or, even worse, change the answer to the question. For example, simply asking the model to generate 'a mathematical diagram' given the edges would sometimes not work.

To address this, we devise a filtering algorithm that rejects the generated image if the similarity between the canny edges of the original and the newly generated images is too low. This process is outlined in Figure 8. In this way, we empirically observe that we can filter out both the edge cases of (i) not generating everything and (ii) generating extraneous details.

Finally, while Flux diffusion models demonstrate remarkable capabilities in scribing text, the process of extracting a canny edge and then reconstructing text may be noisy and inefficient. Since augmenting the image with ControlNet does not alter the exact location of the geometric shapes, we can scribe text, such as the labels for the vertices, after the original image has been augmented. In this way, we ensure that the text is fully recognizable by both the models and humans, guaranteeing that the question remains solvable. Example images before and after applying the overall pipeline is depicted in Figure 19.

**Prompt design.** In addition to our proposed pipeline, we can vary the style of the generated images such as color, texture, and the background by manipulating the input prompt. For the most basic setup, we can utilize a prompt in the form of `"Diagram on {BACKGROUND}"` where `BACKGROUND` can be manually designed by humans, such as `"A blackboard with a tray of white chalks"` or sourced from LLMs. The prompts used to generate AVSD-c are summarized in Table 17.

### C.4 Quality control

**Image preprocessing.** Geometric perception tasks typically require a precise understanding of details in visual inputs. As the recognizability of certain geometric features depends on image properties (e.g., resolution), there may be concerns that preprocessing might damage the relevant information before the images are fed to visual encoders. While one can argue that preprocessing should be considered part of the VLM performance, we examined several images before and after the preprocessing from the open models used in our experiments. As

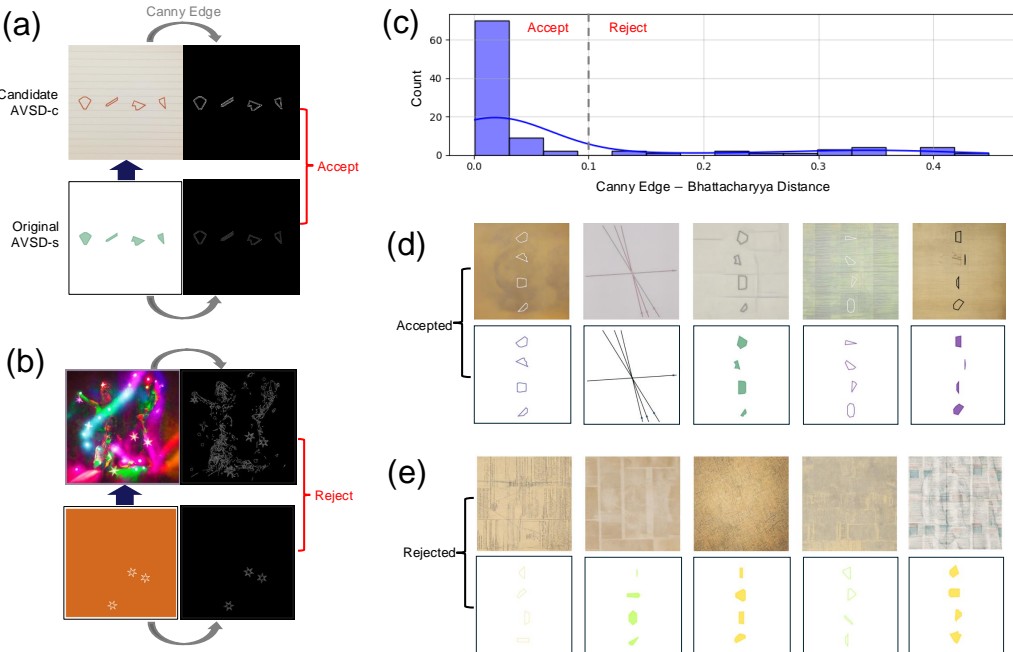

Figure 8: Our pipeline for filtering geometric distortions caused by ControlNet transformations. We reject images whose the Bhattacharyya Distance between the Canny edges before and after exceeds the threshold. (a) Example of the canny edges of accepted images. (b) Example of the canny edges of accepted images (c) Distribution of Bhattacharyya Distance during AVSD-c generation. (d) Examples of before and after of the accepted images. (e) Examples of before and after of the rejected images.

shown in Figure 9, we observed almost no degradation, suggesting that our dataset remains unaffected by this issue.

**Data filtering.** For synthetic generation, we used the following two measures to filter out the generation of invalid images.

1. During generation, we adjusted the parameters of the plotting code and set conditions to certify visibility of the images. For instance, we forced the distance between any point or text to not be too close and indistinguishable.

2. The automatic generation code chooses one of the predefined tasks, then generates a question, an answer, and an image based on the corresponding templates assigned to the task. The authors double-checked the templates to ensure that the generated result could be reasonable.

**Dataset verification.** Across AVSD-h, AVSD-s, AVSD-c, and $\nu$-geometry, we conducted the data verification process as described in Section 5. For AVSD-c, we further conducted the process summarized in Figure 8.

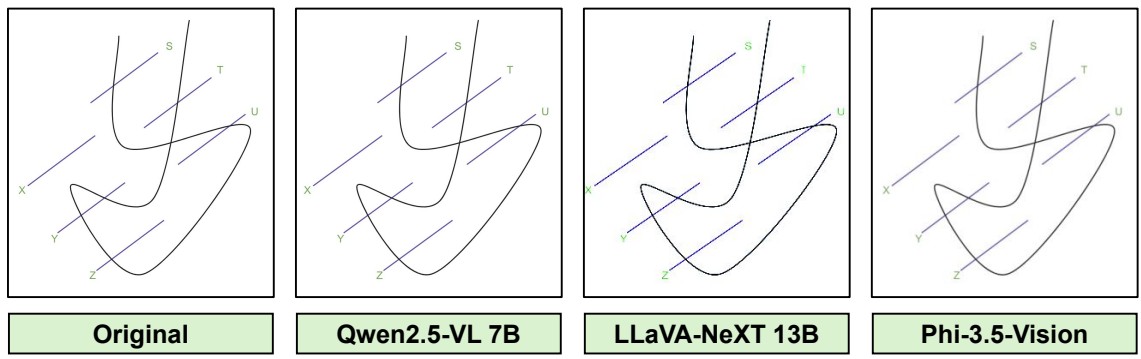

Figure 9: An example of before and after preprocessing our data, under the processors of Qwen2.5-VL 7B, LLaVA-NeXT 13B, and Phi-3.5-Vision-Instruct.

## Appendix D. Model versions

We evaluated closed-source models GPT-4o (OpenAI (2024b)), OpenAI o1 (OpenAI (Dec. 2024)), o3 (OpenAI (Apr. 2025)), Gemini 2.5 Flash (Deepmind (Apr. 2025a)) and Pro (Deepmind (Apr. 2025b)), and open-weight models LLaVA-NeXT (Liu et al. (2024)), LLaVA-OneVision (Li et al. (2025)), Math-LLaVA (Shi et al. (2024)), G-LLaVA (Gao et al. (2025)), Qwen2.5-VL (Bai et al. (2025)), and Phi-3.5-Vision (Abdin et al. (2024)). Tables 8 and 9 describe further details about the model sizes and versions. For closed-source models, we used the commercial APIs. Unless otherwise specified, models equipped with reasoning capabilities were evaluated with reasoning enabled, and their thinking budgets were set to the default value (e.g., medium for OpenAI models, and 1024 tokens for Gemini models). For open models, temperatures were set to 0, and for proprietary models accessed via API calls, we use the default temperature values.

Table 8: Versions of closed-source models

| Model Name | Version |
| --- | --- |
| ChatGPT | o3-2025-04-16 |
| | o1-2024-12-17 |
| | gpt-4o-2024-08-06 |
| | gpt-4o-mini-2024-07-18 |
| Gemini | gemini-2.5-pro-preview-03-25 |
| | gemini-2.5-flash-preview-04-17 |

Table 9: Versions and model sizes of open-weight models

| Version | Model Size(s) |
|---|---|
| LLaVA-NeXT | 13B, 34B |
| LLaVA-OneVision | 7B |
| Math-LLaVA | 13B |
| G-LLaVA | 13B |
| Qwen2.5-VL | 7B |
| Phi-3.5-Vision-Instruct | 4B |

## Appendix E. Further details on evaluation process

**Further Training Details.** We follow the official LLaVA repository (`https://github.com/haotian-liu/LLaVA`) as our fine-tuning pipeline. In particular, we train with Deep-Speed ZeRO-3 for 1 epoch, batch size 16 per GPU, learning rate `2e-5`, cosine schedule with warmup ratio 0.03, bf16 precision, and context length 2048. The vision encoder is not frozen during fine-tuning.

**$\nu$-geometry evaluation.** For $\nu$-geometry evaluation, we used an exact string matching method. After converting the model's prediction to lowercase, we provide the score depending on its task:

- **Task 1** We assign a score of 1 if the transformed prediction contains the ground truth string, either "true" or "false". Otherwise, the score is 0.

- **Task 2** We assign a score of 1 if the transformed prediction contains the ground truth string, either "(i)" or "(ii)". Otherwise, the score is 0.

Note that we did not observe responses like "i" or "ii" instead of "(i)" or "(ii)" across all models.

**AVSD evaluation.** For AVSD evaluation, we used GPT-4o-mini to extract answers from model responses and to judge correctness. Few-shot in-context learning prompts are provided to GPT-4o-mini as described in Tables 11 and 12. To verify the reliability of this pipeline, we randomly selected 150 problems from our dataset and compared the scores from GPT-4o-mini with human annotations. Reassuringly, GPT-4o-mini and the human annotators agreed on the scoring of the 149 problems. We attribute this high level of reliability, in part, to the straightforward and clear design of our questions and answers.

**Scorer ablation.** To test whether our results are sensitive to our model choice of GPT-4o-mini, we run a scorer ablation on a subset of 180 GPT-4o responses uniformly covering AVSD-h, AVSD-s, AVSD-c, and 18 target skills. We compare the original GPT-4o-mini-based scoring pipeline with two Gemini 3 Flash variants: a judgment-only variant, where Gemini judges the GPT-4o-mini-extracted answer against the answer key, and an end-to-end variant, where Gemini performs both answer extraction and correctness judgment. Overall

accuracy is 61.1% with the original GPT-4o-mini pipeline, 63.3% with Gemini judgment-only scoring, and 62.8% with Gemini end-to-end scoring; the Gemini end-to-end pipeline agrees with the original pipeline on 95.6% of examples. Although this ablation is performed on a subset of GPT-4o responses, it provides evidence that GPT-4o-mini is a reliable scorer for the simple, constrained answer format used in AVSD.

Table 10: Effect of varying the thinking budget on performance gain of Gemini 2.5 Flash on AVSD-h.

| Thinking budget | 0 | 1024 | 4096 | 16384 |
|---|---|---|---|---|
| Performance gain compared to 1024 | -4.7% | +0 | -0.2% | +0.3% |

# Appendix F. AVSD evaluation details

In this section, we provide full details of Section 5.

## F.1 Qualitative failure modes

To complement the aggregate accuracy results, we manually inspected incorrect predictions across several AVSD skills. Figure 10 shows representative examples. In (a) and (c), models fail to distinguish local geometric relations, such as whether two objects are disjoint, tangent, or cross each other. In (b), as well as the first problem in Figure 17, the model fails to compare angle magnitudes or classify the marked angle as acute, obtuse, or right. By contrast, (d) and (e) in Figure 10 involve global spatial relations, such as reflection and rotation, where the relevant evidence is distributed across multiple objects and their relative positions and orientations. Together, these examples suggest that AVSD exposes perceptual challenges in interpreting both local geometric features and global geometric structures, rather than failures of symbolic reasoning alone.

To further examine the local failure cases, we performed a lightweight diagnostic. For a targeted subset of originally incorrect predictions of GPT-4o about skills that depend on local cues, including tangency, intersection, and angle, we provided up to $3\times$ magnified views of the relevant regions and asked the same question. We manually verified that the magnified images preserved the intended geometric relation and fully contained necessary visual cues. Among 30 originally incorrect local-geometry predictions, 9 were corrected after zooming in. Since this is a small targeted sample focused on local geometries, we do not interpret the 9/30 recovery rate as an accurate benchmark-level estimate but treat it as a diagnostic observation. The limited recovery shows that scaling can affect some predictions, but the fact that most cases remained incorrect suggests that these failures cannot be fully explained by limited visibility or insufficient image scale. Together with the analysis in Section C.4, which suggests that inputs are not visibly degraded by preprocessing, this provides limited evidence of model difficulty in interpreting accessible geometric relations, rather than being fully explained by visibility issues such as preprocessing or image scale.

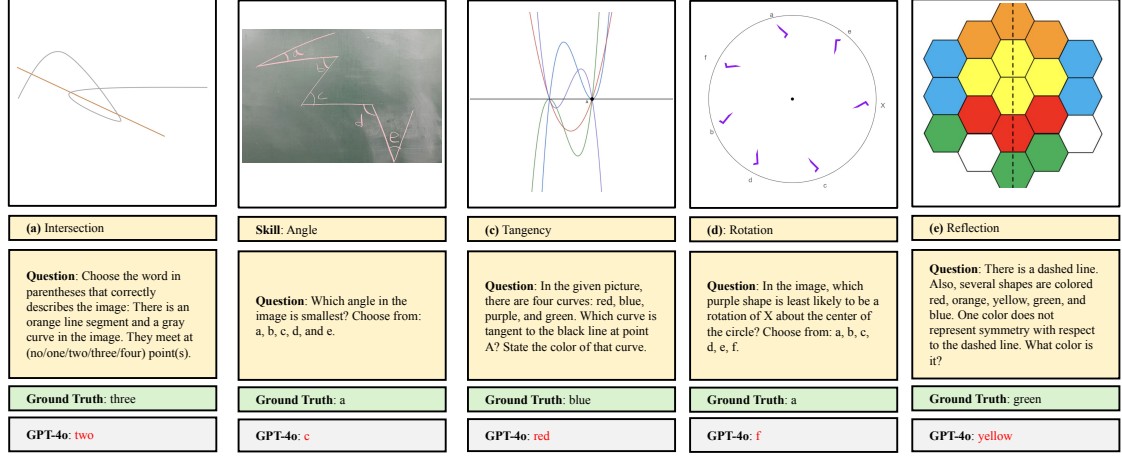

Figure 10: Examples of AVSD problems on which GPT-4o failed.

## F.2 Limited effectiveness of chain-of-thought and reasoning

As discussed in Section 5.1, CoT prompting did not provide meaningful performance gains. In AVSD-h, GPT-4o had only a 2% gain from applying CoT. However, CoT worsened performances of skills including `OCR`, `Length`, and `Symbol`. Models including LLaVA-Next scored worse with CoT on average. As in the case of Figure 11, by inspecting the responses of GPT-4o with and without CoT prompting, we observe that the additional reasoning steps are not generally helpful in comprehending visual inputs.

On the other hand, recent proprietary models with reasoning capabilities exhibit performance gains when reasoning is used. Specifically, we use the Gemini 2.5 Flash model on AVSD-h under different thinking budgets. Table 10 shows the evaluation results under different budgets. We observed an improvement with reasoning, but increasing the thinking budget did not provide further improvements beyond a certain point.

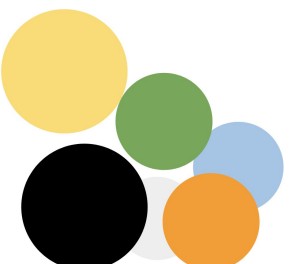

**Question**
In the image, there are circles with colors of gray, blue, green, yellow, orange, and black. Which color is the circle most likely to be tangent to the green circle?

**Ground Truth**
yellow

**GPT-4o with CoT**
To determine which circle is most likely to be tangent to the green circle, let's analyze the image step by step:\n\n1. **Position of Circles:**\n  - The green circle is positioned in contact with several other circles.\n\n2. **Proximity Analysis:**\n  - The green circle is overlapping slightly with the yellow and gray circles. \n  - However, the circle that appears to be almost non-overlapping and looks likely to touch the green circle without overlapping is the orange one.\n\n3. **Conclusion:**\n  - The orange circle is positioned in a way that suggests it may be tangent to the green circle, as they are close to touching yet not overlapping.\n\nTherefore, the **orange** circle is most likely to be tangent to the green circle.

**GPT-4o without CoT**
The **orange** circle is most likely to be tangent to the green circle.

**Gemini 2.5 Flash with reasoning**
Based on the image, the circle most likely to be tangent to the green circle is the \*\***blue**\*\* circle. They appear to touch at a single point.

**Gemini 2.5 Flash without reasoning**
The circle most likely to be tangent to the green circle is the \*\***blue**\*\* circle. Looking at the image, the blue circle is positioned to the right of the green circle and appears to be touching it. The other circles are either further away or positioned such that they are not tangent to the green circle.

Figure 11: Responses of GPT-4o and Gemini 2.5 Flash on a problem from AVSD with and without CoT or reasoning.

### F.3 Style-generation ablation.

AVSD-c uses ControlNet-based style augmentation to test whether VLMs are robust to geometry-preserving changes in visual appearance. To examine whether the style-sensitivity result depends on this particular model, we conducted an additional ablation using an independent image-editing pipeline based on Qwen-Image-Edit. We selected 90 AVSD-s examples covering 18 target skills, with 5 examples per skill, and generated five styled variants: chalkboard, notebook, blueprint, digital-screen, and textured-paper. This produced 450 styled variants in total. Figure 13 shows an example variant generated using Qwen-Image-Edit.

On this subset, GPT-4o achieves 58.9% accuracy on the original AVSD-s images and 47.1% accuracy on the styled variants, corresponding to an 11.8%p drop. This is close to the 12.0 percentage-point drop observed in the original AVSD-s to AVSD-c comparison. Therefore, this ablation partially supports our conclusion that VLMs are not robust to geometry-preserving style changes and that this effect is not restricted to the specific ControlNet pipeline used to construct AVSD-c.

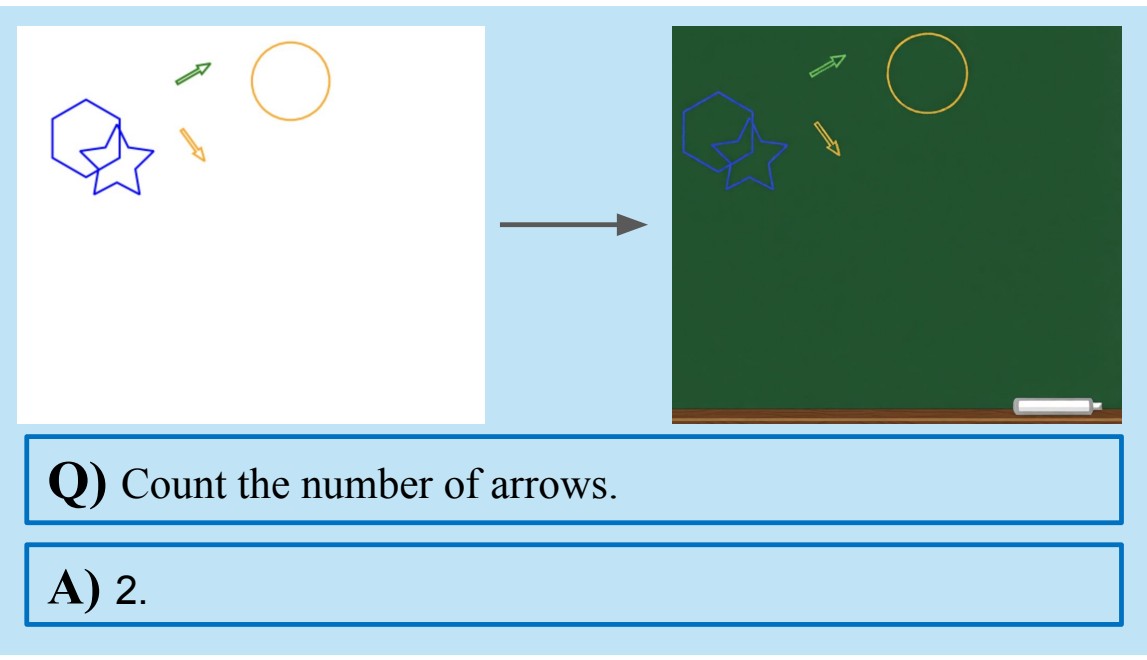

Figure 12: An example of style change of AVSD-s under Qwen-Image-Edit.

### F.4 Textbook-50 External Validation

To partially examine the gap between our controlled AVSD diagrams and textbook-style geometry diagrams, we constructed a small validation set, Textbook-50, using diagrams from PGPS9K (Zhang et al. (2023b)), which contains textbook-derived plane geometry diagrams and provides diagram annotations. For each image, we generated new questions targeting geometric perception that can be answered from its diagram annotation for accuracy.

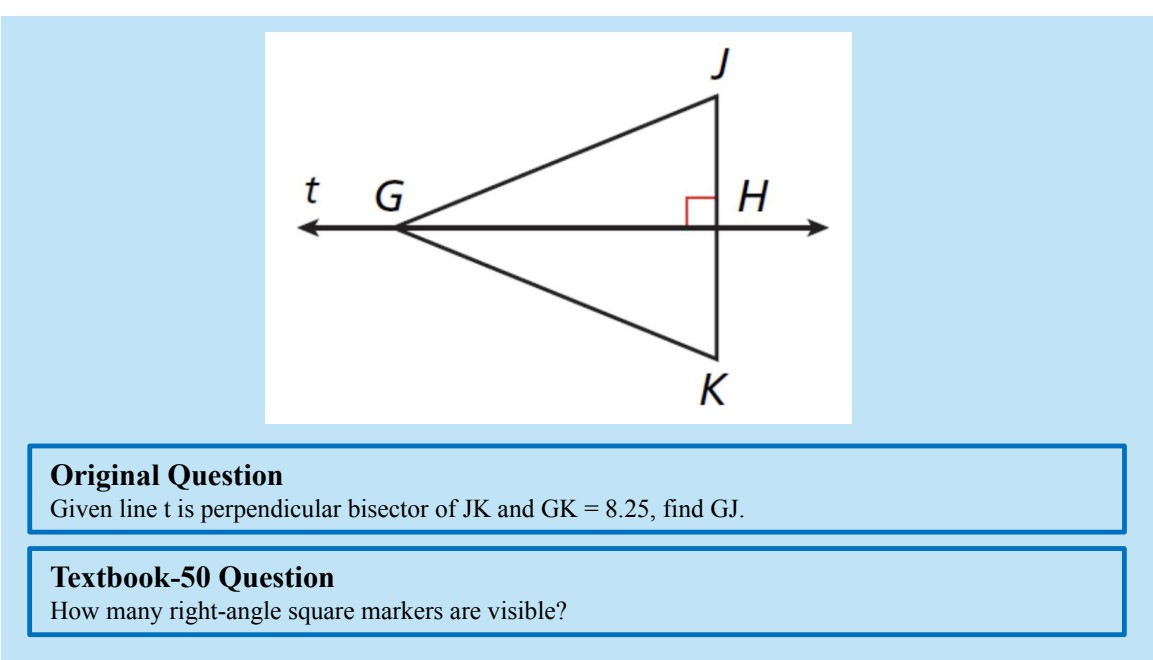

**Original Question**
Given line t is perpendicular bisector of JK and GK = 8.25, find GJ.

**Textbook-50 Question**
How many right-angle square markers are visible?

Figure 13: An example, including the image, original question, and redesigned question from Textbook-50.

We evaluated GPT-4o, GPT-4o with CoT, Qwen2.5-VL-7B-Instruct, and Gemini 2.5 Flash on Textbook-50. The results are shown in Table 13: GPT-4o achieved 70.0%, GPT-4o with CoT achieved 62.0%, Qwen2.5-VL-7B-Instruct achieved 52.0%, and Gemini 2.5 Flash achieved 74.0%. They suggest that the perceptual difficulties measured by AVSD also appear on textbook-style geometry diagrams. Notably, CoT prompting did not improve GPT-4o's performance, which aligns with our observation in 5.1. However, because Textbook-50 is small and manually constructed, we do not treat it as a fully comprehensive estimate of performance of models on real-world geometry.

### F.5 Detailed evaluation results

Table F presents our complete evaluation results on AVSD. The subcolumn named "Overall" indicates the accuracy across all problems of its corresponding skill from AVSD test subset. The column named "**TOTAL**" describes the overall accuracy across all problems in AVSD, across all skills. We also present the evaluation results on AVSD-h separately at Table 15, to provide the performance on different difficulty levels.

| Element | Prompt |
|---|---|
| **System prompt** | Imagine you are an intelligent teacher. Thoroughly read the provided instruction to ensure a solid understanding of the information provided |
| **Task description** | Please read the following example. Then extract the answer from the model response and type it at the end of the prompt. If the question requires a full sentence with a correct word filled in, please provide the word only.
{*examples*}
Question: {*question*}
Model response: {*model response*}
Extracted Answer: |
| **Examples** | **Question:** There is a single rectangle with multiple color layers in the image. What is the color of the boundary of the rectangle? The answer should be one of 'red', 'yellow', 'green', or 'blue'.
**Model response:** The color of the boundary of the circle is red.
**Extracted answer:** red

**Question:** How many line segments are in the image? Answer should be a number.
**Model response:** There are 4 dashed line segments in the image.
**Extracted answer:** 4

**Question:** Choose the word in parentheses that correctly describes the image. Rewrite the sentence with the chosen word.
In the image, shape (A/B) has sides curved inward. (Unit: $)
**Model response:** In the image, shape B has sides curved inward.
**Extracted answer:** B

**Question:** Choose the phrase in parentheses that correctly describes the image. Rewrite the sentence with the chosen phrase.
In the given image, the green arrow (is longer than/has the same length as/is shorter than) the black arrow.
**Model response:** In the given image, the green arrow is longer than the black arrow.
**Extracted answer:** is longer than

**Question:** In this image, choose the path which is a single line segment between points A and B from the following options. Provide your answer as a single uppercase letter: (A) the purple path (B) the blue path (C) the green path (D) the red path
**Model response:** B
**Extracted answer:** B

**Question:** Choose the most appropriate color to fill in the box marked with '?' in the image. The answer is one of 'a', 'b', 'c', or 'd'.
**Model response:** The correct color to fill in the box marked with '?' is (a) blue. The colors are following a gradient pattern from red, to a more purple hue, and finally to blue. The logical next color in the sequence would be blue, as it extends the progression seen in the previous squares.
**Extracted answer:** a

**Question:** There is a book in the image. What is the color of the book in the image? Choose answer from the number of the option and give your answer in "1", "2", "3", or "4". (1) red (2) yellow (3) blue (4) green
**Model response:** The color of the guitar in the image is (2) yellow.
**Extracted answer:** 2 |

Table 11: System prompt, task description, and examples used to prompt GPT-4o-mini for answer extraction.

| Element | Prompt |
|---|---|
| **System prompt** | Imagine you are an intelligent teacher. Thoroughly read the provided instruction to ensure a solid understanding of the information provided. |
| **Task description** | The [Standard Answer] is the correct answer to the question, and the [Model Answer] is the answer generated by a model for that question. |
| | Thoroughly read both the [Standard Answer] and the [Model Answer]. Assess the consistency of the information provided in these two responses. |
| | Although you do not know the specific question, you can still assess the consistency between the two responses by checking for logical conflicts if both responses are assumed to be correct. |
| | If the [Model Answer] is consistent with the [Standard Answer], please answer '1'. Otherwise, answer '0'. |
| | When the [Standard Answer] is provided as a list, answer '1' if the [Model Answer] is consistent with at least one item on the list. Otherwise, answer '0'. |
| | Below are the examples of the correct consistency judgment. |
| | Don't explain anything. Just answer in 0 or 1. |
| | ** |
| | Rememeber the format should be |
| | Judgment: 0 |
| | or |
| | Judgment: 1 |
| | ** |
| | You must keep the format!!! {*examples*} |
| | Now, below are two answers to a question. What is your judgment? |
| | [Standard Answer] {*standard answer*} |
| | [Model Answer] {*extracted answer*} |
| | Judgment: |
| **Examples** | **[Standard Answer]** a |
| | **[Model Answer]** a |
| | **Judgment:** 1 |
| | **[Standard Answer]** 1 |
| | **[Model Answer]** 4 |
| | **Judgment:** 0 |
| | **[Standard Answer]** circle |
| | **[Model Answer]** the circle |
| | **Judgment:** 1 |
| | **[Standard Answer]** 4 |
| | **[Model Answer]** shape 4 |
| | **Judgment:** 1 |
| | **[Standard Answer]** line segment B and C |
| | **[Model Answer]** B, C |
| | **Judgment:** 1 |
| | **[Standard Answer]** 2 |
| | **[Model Answer]** two |
| | **Judgment:** 1 |
| | **[Standard Answer]** three |
| | **[Model Answer]** 3 |
| | **Judgment:** 1 |
| | **[Standard Answer]** ['ac', 'ca'] |
| | **[Model Answer]** ca |
| | **Judgment:** 1 |

Table 12: System prompt, task description, and examples used to prompt GPT-4o-mini for judgment. Some of the blank lines are omitted for visibility.

Table 13: Results on Textbook-50, a small PGPS9K-based external validation set with 50 textbook-style geometry diagrams and perception questions.

| Model | Accuracy |
|---|---|
| GPT-4o | 35/50 (70.0%) |
| GPT-4o + CoT | 31/50 (62.0%) |
| Qwen2.5-VL-7B-Instruct | 26/50 (52.0%) |
| Gemini 2.5 Flash | 37/50 (74.0%) |

### Block 1

| Model | TOTAL Overall | AVSD-h | AVSD-s | AVSD-c | OCR Overall | AVSD-h | AVSD-s | AVSD-c | Abs. pos. Overall | AVSD-h | AVSD-s | AVSD-c | Adjacency Overall | AVSD-h | AVSD-s | AVSD-c | Angle Overall | AVSD-h | AVSD-s | AVSD-c | Area Overall | AVSD-h | AVSD-s | AVSD-c | Boundary Overall | AVSD-h | AVSD-s | AVSD-c |
|---|---|---|---|---|---|---|---|---|---|---|---|---|---|---|---|---|---|---|---|---|---|---|---|---|---|---|---|---|
| o3 | 64.5 | 73.8 | 62.5 | 50.3 | 71.6 | 85.8 | 65.3 | 49.3 | 67.8 | 84.9 | 60.7 | 62.7 | 77.3 | 79.7 | 67.3 | 93.3 | 48.3 | 63.8 | 40.7 | 25.3 | 85.0 | 88.8 | 90.7 | 65.3 | 77.5 | 90.5 | 72.0 | 62.7 |
| Gemini 2.5 Flash | 68.4 | 70.5 | 70.7 | 59.5 | 80.6 | 89.1 | 77.3 | 65.3 | 77.2 | 81.4 | 73.3 | 78.7 | 69.5 | 71.5 | 66.7 | 69.3 | 54.9 | 63.2 | 48.7 | 48.0 | 84.0 | 76.3 | 92.7 | 64.0 | 77.7 | 81.1 | 80.0 | 66.7 |
| Gemini 2.5 Pro | 71.7 | 75.0 | 72.3 | 63.9 | 82.1 | 88.5 | 82.0 | 66.7 | 79.7 | 79.1 | 80.3 | 80.0 | 76.4 | 82.9 | 66.7 | 82.7 | 57.3 | 66.5 | 52.7 | 48.0 | 83.2 | 84.0 | 87.3 | 73.3 | 78.8 | 87.8 | 75.3 | 68.0 |
| GPT-4o | 55.2 | 61.9 | 54.7 | 43.1 | 70.6 | 88.5 | 66.0 | 36.0 | 65.9 | 79.1 | 60.0 | 62.7 | 55.2 | 69.9 | 50.0 | 41.3 | 36.8 | 44.9 | 32.0 | 26.7 | 76.4 | 74.0 | 82.7 | 69.3 | 63.3 | 75.0 | 73.3 | 20.0 |
| o1 | 58.5 | 67.9 | 55.7 | 45.6 | 74.5 | 90.7 | 68.7 | 46.7 | 66.6 | 84.9 | 58.0 | 62.7 | 73.0 | 82.1 | 58.7 | 86.7 | 41.0 | 55.7 | 30.7 | 25.3 | 78.9 | 85.2 | 79.3 | 64.0 | 75.1 | 87.8 | 74.7 | 50.7 |
| LLaVA-NeXT (13B) | 30.0 | 32.8 | 30.8 | 23.0 | 45.6 | 54.6 | 42.7 | 29.3 | 54.3 | 65.1 | 48.7 | 53.3 | 14.9 | 26.8 | 12.0 | 1.3 | 22.0 | 22.2 | 25.3 | 14.7 | 42.1 | 52.7 | 40.0 | 22.7 | 32.7 | 33.8 | 48.0 | 0.0 |
| LLaVA-NeXT (34B) | 34.2 | 38.0 | 35.1 | 25.2 | 52.2 | 65.6 | 46.0 | 32.0 | 61.7 | 73.3 | 56.0 | 60.0 | 17.5 | 26.8 | 16.7 | 4.0 | 24.1 | 25.9 | 24.7 | 18.7 | 49.2 | 53.3 | 55.3 | 28.0 | 32.2 | 35.1 | 44.0 | 2.7 |
| Math-LLaVA | 30.0 | 30.1 | 32.0 | 25.9 | 33.6 | 37.2 | 32.7 | 26.7 | 49.8 | 54.7 | 45.3 | 53.3 | 15.5 | 28.5 | 12.0 | 1.3 | 23.7 | 16.8 | 29.3 | 29.3 | 44.2 | 44.4 | 52.7 | 26.7 | 27.9 | 31.1 | 36.7 | 4.0 |
| LLaVA-OneVision (7B) | 38.5 | 42.2 | 37.9 | 32.3 | 57.4 | 66.1 | 56.0 | 38.7 | 60.1 | 75.6 | 48.7 | 65.3 | 23.6 | 44.7 | 12.0 | 12.0 | 27.8 | 21.6 | 32.0 | 34.7 | 56.9 | 57.4 | 59.3 | 50.7 | 38.1 | 48.6 | 45.3 | 2.7 |
| Qwen2.5-VL (7B) | 44.3 | 44.7 | 45.1 | 33.9 | 65.2 | 79.8 | 58.7 | 42.7 | 65.9 | 82.6 | 57.3 | 64.0 | 32.2 | 54.5 | 23.3 | 13.3 | 29.8 | 41.6 | 19.3 | 21.3 | 67.0 | 57.4 | 82.7 | 57.3 | 42.1 | 42.6 | 56.0 | 13.3 |
| Phi-3.5-Vision-Instruct | 34.3 | 35.6 | 36.5 | 27.0 | 48.0 | 55.7 | 44.7 | 36.0 | 61.1 | 76.7 | 52.7 | 60.0 | 19.3 | 30.9 | 16.7 | 5.3 | 24.1 | 25.9 | 23.3 | 21.3 | 52.3 | 40.2 | 72.7 | 38.7 | 33.8 | 38.5 | 46.0 | 0.0 |

### Block 2

| Model | Cardinal Overall | AVSD-h | AVSD-s | AVSD-c | Cardinal Direction Overall | AVSD-h | AVSD-s | AVSD-c | Color Overall | AVSD-h | AVSD-s | AVSD-c | Congruence Overall | AVSD-h | AVSD-s | AVSD-c | Connectedness Overall | AVSD-h | AVSD-s | AVSD-c | Convexity Overall | AVSD-h | AVSD-s | AVSD-c | Coordinate Overall | AVSD-h | AVSD-s | AVSD-c |
|---|---|---|---|---|---|---|---|---|---|---|---|---|---|---|---|---|---|---|---|---|---|---|---|---|---|---|---|---|
| o3 | 75.9 | 72.4 | 84.0 | 69.3 | 73.7 | 90.0 | 77.3 | 46.7 | 79.8 | 88.3 | 66.7 | - | 50.9 | 62.6 | 36.7 | 48.0 | 63.7 | 73.3 | 61.3 | 49.3 | 61.7 | 80.3 | 53.3 | 49.3 | 54.5 | 65.9 | 49.3 | 42.7 |
| Gemini 2.5 Flash | 73.6 | 67.3 | 82.7 | 57.3 | 93.0 | 80.9 | 98.0 | 97.3 | 85.3 | 86.6 | 83.3 | - | 58.4 | 61.1 | 54.7 | 42.7 | 63.4 | 69.1 | 65.3 | 46.7 | 64.2 | 73.7 | 56.7 | 57.3 | 48.0 | 71.3 | 49.3 | 46.7 |
| Gemini 2.5 Pro | 74.7 | 79.9 | 79.3 | 60.0 | 91.5 | 88.8 | 68.7 | 94.7 | 85.3 | 89.2 | 72.8 | - | 58.9 | 65.0 | 61.7 | 49.3 | 69.6 | 79.2 | 60.7 | 50.7 | 70.1 | 80.5 | 64.3 | 61.3 | 65.8 | 82.4 | 54.3 | 52.0 |
| GPT-4o | 68.8 | 63.6 | 79.3 | 62.7 | 69.2 | 80.0 | 71.3 | 52.0 | 79.3 | 85.7 | 69.3 | - | 46.5 | 46.8 | 45.3 | 48.0 | 45.9 | 56.0 | 38.0 | 41.3 | 53.2 | 65.8 | 49.3 | 41.3 | 39.7 | 49.7 | 32.0 | 36.0 |
| o1 | 71.3 | 68.2 | 80.0 | 62.7 | 70.8 | 83.3 | 74.0 | 49.3 | 76.9 | 84.4 | 65.3 | - | 43.7 | 49.8 | 36.7 | 41.3 | 52.5 | 57.3 | 46.7 | 54.7 | 54.4 | 68.4 | 50.0 | 41.3 | 41.8 | 55.2 | 31.3 | 37.3 |
| LLaVA-NeXT (13B) | 45.8 | 34.6 | 70.0 | 29.3 | 49.2 | 43.3 | 58.7 | 37.3 | 42.3 | 52.4 | 26.7 | - | 28.5 | 22.2 | 33.3 | 36.0 | 24.5 | 29.3 | 21.3 | 21.3 | 26.3 | 41.9 | 20.7 | 13.3 | 13.3 | 18.9 | 8.7 | 12.0 |
| LLaVA-NeXT (34B) | 44.6 | 37.4 | 58.7 | 37.3 | 53.7 | 55.6 | 59.3 | 40.0 | 49.6 | 59.7 | 34.0 | - | 33.2 | 26.1 | 36.7 | 45.3 | 24.8 | 32.7 | 21.3 | 16.0 | 26.9 | 38.5 | 24.7 | 13.3 | 21.5 | 28.7 | 16.7 | 17.3 |
| Math-LLaVA | 46.9 | 33.6 | 70.0 | 38.7 | 38.4 | 38.9 | 37.3 | 40.0 | 42.5 | 45.9 | 37.3 | - | 28.0 | 19.2 | 34.7 | 38.7 | 18.7 | 21.3 | 16.0 | 18.7 | 27.8 | 36.8 | 24.7 | 20.0 | 14.7 | 22.4 | 9.3 | 10.7 |
| LLaVA-OneVision (7B) | 53.8 | 41.6 | 78.7 | 38.7 | 56.8 | 56.7 | 56.7 | 57.3 | 54.6 | 67.5 | 34.7 | - | 30.4 | 28.1 | 30.7 | 36.0 | 23.5 | 28.7 | 22.7 | 14.7 | 33.9 | 47.9 | 26.0 | 28.0 | 22.6 | 30.8 | 17.3 | 17.3 |
| Qwen2.5-VL (7B) | 64.7 | 56.5 | 84.0 | 49.3 | 59.7 | 65.6 | 64.7 | 42.7 | 62.2 | 71.0 | 48.7 | - | 35.5 | 33.0 | 37.3 | 38.7 | 30.9 | 38.7 | 27.3 | 22.7 | 38.0 | 51.3 | 34.7 | 24.0 | 33.2 | 50.3 | 22.0 | 22.7 |
| Phi-3.5-Vision-Instruct | 53.1 | 37.9 | 79.3 | 44.0 | 50.2 | 47.8 | 54.7 | 44.0 | 48.0 | 55.0 | 37.3 | - | 26.4 | 22.2 | 28.7 | 33.3 | 23.5 | 25.3 | 24.0 | 18.7 | 28.4 | 40.2 | 28.0 | 10.7 | 20.7 | 30.8 | 15.3 | 12.0 |

### Block 3

| Model | Curvature Overall | AVSD-h | AVSD-s | AVSD-c | Direction Overall | AVSD-h | AVSD-s | AVSD-c | Interior Overall | AVSD-h | AVSD-s | AVSD-c | Intersection Overall | AVSD-h | AVSD-s | AVSD-c | Length Overall | AVSD-h | AVSD-s | AVSD-c | Line Overall | AVSD-h | AVSD-s | AVSD-c | Overlap Overall | AVSD-h | AVSD-s | AVSD-c |
|---|---|---|---|---|---|---|---|---|---|---|---|---|---|---|---|---|---|---|---|---|---|---|---|---|---|---|---|---|
| o3 | 62.6 | 80.9 | 60.7 | 34.7 | 60.1 | 69.6 | 58.7 | 45.3 | 76.0 | 88.7 | 68.7 | 65.3 | 68.5 | 67.2 | 64.0 | 60.0 | 61.1 | 77.0 | 56.7 | 44.0 | 72.5 | 78.1 | 82.0 | 40.3 | 65.6 | 81.3 | 56.0 | 53.3 |
| Gemini 2.5 Flash | 61.1 | 70.4 | 53.3 | 29.3 | 68.0 | 69.6 | 72.0 | 57.3 | 73.7 | 82.0 | 65.3 | 58.7 | 61.9 | 56.0 | 60.2 | 74.7 | 68.6 | 68.9 | 68.4 | 62.7 | 77.1 | 80.7 | 84.0 | 56.0 | 81.5 | 71.9 | 90.7 | 81.3 |
| Gemini 2.5 Pro | 60.3 | 72.8 | 67.4 | 30.7 | 70.6 | 72.6 | 76.7 | 54.7 | 74.4 | 89.3 | 67.3 | 58.7 | 65.2 | 61.9 | 60.0 | 81.3 | 70.6 | 71.3 | 70.7 | 69.3 | 76.9 | 85.9 | 85.3 | 60.0 | 84.8 | 73.4 | 95.3 | 85.7 |
| GPT-4o | 59.8 | 72.5 | 62.7 | 32.0 | 51.5 | 53.6 | 53.3 | 44.0 | 72.3 | 84.0 | 70.7 | 52.0 | 54.6 | 51.5 | 47.3 | 74.7 | 47.8 | 62.3 | 44.0 | 32.0 | 67.8 | 68.9 | 78.0 | 45.3 | 52.2 | 74.1 | 41.3 | 33.3 |
| o1 | 57.6 | 77.1 | 58.7 | 21.3 | 56.5 | 66.7 | 51.3 | 48.0 | 75.2 | 88.0 | 69.3 | 61.3 | 58.2 | 59.0 | 49.3 | 74.7 | 50.4 | 67.2 | 46.0 | 32.0 | 71.7 | 81.5 | 78.0 | 41.3 | 56.9 | 76.3 | 54.0 | 26.7 |
| LLaVA-NeXT (13B) | 25.6 | 26.7 | 24.7 | 25.3 | 23.7 | 23.2 | 27.3 | 17.3 | 32.5 | 41.3 | 26.7 | 26.7 | 24.8 | 20.9 | 26.7 | 28.0 | 31.1 | 40.2 | 31.3 | 16.0 | 33.1 | 37.0 | 32.7 | 26.7 | 29.7 | 33.8 | 24.7 | 32.0 |
| LLaVA-NeXT (34B) | 27.2 | 28.2 | 25.3 | 29.3 | 28.7 | 29.0 | 26.7 | 32.0 | 50.1 | 59.3 | 44.0 | 44.0 | 28.7 | 24.6 | 29.3 | 34.7 | 30.3 | 39.3 | 33.3 | 9.3 | 45.8 | 54.1 | 48.7 | 25.3 | 34.9 | 47.5 | 32.0 | 17.3 |
| Math-LLaVA | 19.4 | 18.3 | 20.0 | 20.0 | 29.8 | 27.5 | 32.0 | 29.3 | 35.7 | 39.3 | 36.0 | 28.0 | 30.9 | 23.9 | 32.7 | 40.0 | 33.1 | 38.5 | 34.7 | 21.3 | 29.2 | 29.6 | 30.0 | 26.7 | 26.1 | 30.2 | 20.7 | 29.3 |
| LLaVA-OneVision (7B) | 33.1 | 42.0 | 28.7 | 26.7 | 39.7 | 40.6 | 37.3 | 42.7 | 54.1 | 69.3 | 45.3 | 41.3 | 35.1 | 23.1 | 40.7 | 45.3 | 36.9 | 50.8 | 36.0 | 16.0 | 45.6 | 52.6 | 50.0 | 24.0 | 39.3 | 41.0 | 42.0 | 30.7 |
| Qwen2.5-VL (7B) | 33.4 | 45.0 | 30.0 | 20.0 | 44.9 | 50.0 | 47.3 | 30.7 | 65.6 | 77.3 | 60.7 | 52.0 | 30.6 | 21.6 | 34.7 | 38.7 | 42.1 | 53.3 | 37.3 | 33.3 | 55.6 | 54.8 | 68.0 | 32.0 | 40.4 | 46.8 | 48.7 | 12.0 |
| Phi-3.5-Vision-Instruct | 29.2 | 32.1 | 31.3 | 20.0 | 26.4 | 27.5 | 30.0 | 17.3 | 40.5 | 43.3 | 43.0 | 34.7 | 36.2 | 32.8 | 36.7 | 41.3 | 39.5 | 42.6 | 38.7 | 36.0 | 35.6 | 38.5 | 38.5 | 24.0 | 32.4 | 34.5 | 38.0 | 17.3 |

### Block 4

| Model | Ordinal Overall | AVSD-h | AVSD-s | AVSD-c | Orientation Overall | AVSD-h | AVSD-s | AVSD-c | Orthogonality Overall | AVSD-h | AVSD-s | AVSD-c | Parallel Overall | AVSD-h | AVSD-s | AVSD-c | Point Overall | AVSD-h | AVSD-s | AVSD-c | Reflection Overall | AVSD-h | AVSD-s | AVSD-c | Relative Position Overall | AVSD-h | AVSD-s | AVSD-c | Rotation Overall | AVSD-h | AVSD-s | AVSD-c |
|---|---|---|---|---|---|---|---|---|---|---|---|---|---|---|---|---|---|---|---|---|---|---|---|---|---|---|---|---|---|---|---|---|
| o3 | 80.2 | 80.6 | 82.7 | 74.7 | 66.2 | 53.7 | 84.7 | 49.3 | 55.9 | 70.9 | 57.3 | 36.0 | 60.8 | 53.1 | 66.0 | 64.0 | 78.3 | 89.8 | 77.3 | 57.7 | 50.9 | 59.8 | 48.7 | 32.0 | 61.6 | 70.4 | 64.7 | 44.0 | 22.3 | 23.9 | 18.7 | 26.7 |
| Gemini 2.5 Flash | 85.2 | 82.1 | 91.3 | 78.7 | 53.4 | 47.5 | 57.3 | 53.3 | 61.7 | 53.5 | 70.7 | 53.3 | 67.5 | 59.2 | 69.3 | 78.7 | 81.6 | 89.5 | 75.3 | 81.3 | 54.2 | 48.2 | 62.0 | 50.7 | 71.7 | 67.3 | 84.7 | 53.3 | 31.8 | 33.8 | 30.0 | 32.0 |
| Gemini 2.5 Pro | 86.4 | 87.3 | 87.3 | 82.7 | 65.5 | 51.7 | 80.7 | 57.3 | 63.0 | 57.0 | 71.3 | 53.3 | 69.1 | 61.2 | 71.3 | 80.0 | 91.6 | 89.5 | 92.0 | 96.0 | 56.6 | 56.8 | 58.7 | 52.0 | 75.5 | 72.4 | 88.7 | 54.7 | 29.7 | 33.1 | 23.3 | 36.0 |
| GPT-4o | 75.2 | 68.7 | 86.0 | 65.3 | 39.3 | 36.4 | 42.7 | 37.3 | 42.1 | 45.3 | 48.7 | 25.3 | 55.9 | 31.3 | 76.0 | 64.0 | 68.5 | 81.5 | 77.3 | 32.0 | 39.4 | 35.2 | 48.7 | 32.0 | 54.5 | 69.4 | 52.0 | 40.0 | 21.3 | 23.9 | 17.3 | 24.0 |
| o1 | 79.7 | 79.1 | 84.0 | 72.0 | 61.8 | 50.4 | 81.3 | 41.3 | 44.1 | 51.2 | 52.7 | 18.7 | 48.9 | 40.8 | 48.0 | 66.7 | 68.5 | 86.1 | 74.0 | 32.0 | 43.6 | 45.7 | 45.3 | 34.7 | 49.5 | 63.3 | 47.3 | 36.0 | 25.9 | 31.0 | 20.7 | 26.7 |
| LLaVA-NeXT (13B) | 25.9 | 25.4 | 26.0 | 26.7 | 37.9 | 32.2 | 48.7 | 25.3 | 23.2 | 12.8 | 27.3 | 26.7 | 14.8 | 12.9 | 15.3 | 17.3 | 33.0 | 34.3 | 42.7 | 12.0 | 22.9 | 27.6 | 18.7 | 18.7 | 27.9 | 36.7 | 26.0 | 20.0 | 21.8 | 19.7 | 22.7 | 24.0 |
| LLaVA-NeXT (34B) | 33.1 | 29.1 | 36.0 | 34.7 | 40.2 | 36.4 | 50.0 | 26.7 | 26.0 | 19.8 | 36.0 | 13.3 | 21.5 | 13.6 | 23.3 | 33.3 | 51.4 | 47.2 | 67.3 | 25.3 | 24.8 | 29.1 | 20.7 | 21.3 | 31.9 | 41.8 | 32.0 | 18.7 | 17.4 | 13.4 | 18.0 | 24.0 |
| Math-LLaVA | 23.7 | 22.4 | 22.0 | 29.3 | 38.7 | 34.7 | 50.7 | 21.3 | 23.5 | 14.0 | 30.7 | 20.0 | 13.2 | 22.4 | 8.7 | 4.0 | 42.3 | 34.3 | 54.7 | 29.3 | 34.3 | 25.1 | 40.0 | 46.7 | 26.9 | 31.6 | 26.7 | 21.3 | 24.3 | 32.4 | 16.7 | 24.0 |
| LLaVA-OneVision (7B) | 35.4 | 24.6 | 41.3 | 42.7 | 32.7 | 33.1 | 38.0 | 21.3 | 29.6 | 23.3 | 34.7 | 26.7 | 17.7 | 12.9 | 18.0 | 26.7 | 47.1 | 45.4 | 54.0 | 36.0 | 32.3 | 34.2 | 32.0 | 28.0 | 35.6 | 43.9 | 32.7 | 30.7 | 22.9 | 30.3 | 16.7 | 21.3 |
| Qwen2.5-VL (7B) | 57.1 | 57.5 | 62.0 | 46.7 | 36.7 | 29.8 | 43.3 | 34.7 | 29.6 | 31.4 | 34.7 | 17.3 | 43.8 | 25.2 | 51.3 | 65.3 | 56.2 | 61.1 | 66.0 | 29.3 | 24.8 | 26.1 | 25.3 | 20.0 | 44.6 | 46.9 | 47.3 | 36.0 | 25.1 | 26.7 | 23.3 | 20.0 |
| Phi-3.5-Vision-Instruct | 33.7 | 28.4 | 36.7 | 37.3 | 35.8 | 29.8 | 40.7 | 36.0 | 28.0 | 26.7 | 32.7 | 20.0 | 28.5 | 21.1 | 30.7 | 38.7 | 40.5 | 43.5 | 50.0 | 17.3 | 24.8 | 28.6 | 24.7 | 14.7 | 26.6 | 27.6 | 29.3 | 20.0 | 21.3 | 21.1 | 22.7 | 18.7 |

### Block 5

| Model | Rotational Symmetry Overall | AVSD-h | AVSD-s | AVSD-c | Shape Overall | AVSD-h | AVSD-s | AVSD-c | Sharpness Overall | AVSD-h | AVSD-s | AVSD-c | Similarity Overall | AVSD-h | AVSD-s | AVSD-c | Symbol Overall | AVSD-h | AVSD-s | AVSD-c | Tangency Overall | AVSD-h | AVSD-s | AVSD-c | Texture Overall | AVSD-h | AVSD-s | AVSD-c | Width Overall | AVSD-h | AVSD-s | AVSD-c |
|---|---|---|---|---|---|---|---|---|---|---|---|---|---|---|---|---|---|---|---|---|---|---|---|---|---|---|---|---|---|---|---|---|
| o3 | 49.3 | 73.5 | 31.3 | 42.7 | 70.9 | 91.2 | 62.7 | 50.7 | 67.2 | 83.3 | 64.0 | 46.7 | 53.0 | 65.6 | 50.0 | 38.7 | 60.8 | 83.5 | 53.3 | 37.3 | 55.6 | 53.2 | 59.3 | 53.3 | 61.2 | 76.3 | 57.3 | 37.3 | 57.6 | 70.7 | 56.7 | 32.0 |
| Gemini 2.5 Flash | 55.0 | 68.9 | 42.7 | 52.0 | 76.3 | 90.9 | 74.0 | 54.7 | 70.9 | 75.4 | 68.7 | 68.0 | 51.3 | 53.7 | 49.3 | 40.0 | 69.3 | 80.3 | 65.3 | 58.7 | 53.9 | 43.9 | 58.7 | 68.0 | 73.2 | 73.7 | 72.7 | 52.0 | 62.3 | 63.7 | 67.3 | 50.7 |
| Gemini 2.5 Pro | 64.4 | 77.3 | 50.7 | 69.3 | 77.4 | 87.0 | 78.7 | 56.0 | 74.4 | 80.3 | 70.7 | 72.0 | 50.6 | 62.0 | 47.3 | 38.7 | 76.4 | 84.3 | 72.7 | 70.7 | 58.3 | 52.6 | 60.7 | 66.7 | 76.9 | 82.7 | 78.0 | 62.7 | 65.4 | 63.7 | 70.0 | 60.0 |
| GPT-4o | 42.9 | 67.4 | 24.7 | 36.0 | 67.1 | 84.9 | 57.3 | 52.0 | 62.7 | 72.2 | 62.0 | 48.0 | 49.3 | 55.7 | 45.3 | 46.7 | 48.6 | 70.9 | 37.3 | 33.3 | 41.7 | 27.5 | 52.0 | 53.3 | 55.9 | 70.5 | 49.3 | 38.7 | 52.6 | 66.7 | 46.7 | 29.3 |
| o1 | 45.7 | 76.5 | 25.3 | 32.0 | 69.8 | 87.0 | 62.0 | 52.0 | 63.8 | 74.6 | 56.7 | 60.0 | 49.6 | 54.9 | 51.3 | 37.3 | 59.9 | 79.5 | 50.7 | 45.3 | 41.2 | 37.4 | 44.0 | 44.0 | 56.2 | 67.9 | 54.7 | 34.7 | 52.6 | 66.2 | 47.3 | 34.7 |
| LLaVA-NeXT (13B) | 19.3 | 15.9 | 21.3 | 21.3 | 39.6 | 54.8 | 32.7 | 24.0 | 29.1 | 31.7 | 31.3 | 20.0 | 24.5 | 26.2 | 25.3 | 20.0 | 30.1 | 38.6 | 28.0 | 20.0 | 32.8 | 19.9 | 40.7 | 46.7 | 24.4 | 31.4 | 21.3 | 16.0 | 30.9 | 35.7 | 29.3 | 24.0 |
| LLaVA-NeXT (34B) | 21.3 | 33.3 | 17.3 | 8.0 | 41.8 | 56.8 | 34.0 | 28.0 | 37.3 | 45.2 | 38.0 | 22.7 | 29.4 | 28.7 | 30.7 | 28.0 | 32.1 | 36.2 | 33.3 | 22.7 | 32.8 | 29.8 | 36.7 | 32.0 | 23.9 | 32.7 | 20.7 | 12.0 | 33.2 | 35.7 | 35.3 | 24.0 |
| Math-LLaVA | 25.2 | 19.7 | 30.7 | 24.0 | 44.2 | 47.3 | 42.7 | 41.3 | 35.0 | 34.1 | 35.3 | 36.0 | 25.9 | 31.5 | 24.0 | 20.0 | 32.1 | 36.2 | 33.3 | 22.7 | 34.3 | 25.1 | 40.0 | 40.0 | 26.2 | 24.4 | 30.0 | 22.7 | 31.9 | 36.3 | 27.3 | 32.0 |
| LLaVA-OneVision (7B) | 29.4 | 31.1 | 30.0 | 25.3 | 61.2 | 71.2 | 52.7 | 58.7 | 42.5 | 52.4 | 37.3 | 36.0 | 36.9 | 28.7 | 47.3 | 29.3 | 39.8 | 62.2 | 27.3 | 26.7 | 37.1 | 29.2 | 49.3 | 30.7 | 27.6 | 32.7 | 24.7 | 22.7 | 36.9 | 39.5 | 34.0 | 37.3 |
| Qwen2.5-VL (7B) | 42.0 | 43.9 | 37.3 | 48.0 | 61.7 | 71.9 | 58.7 | 48.0 | 44.7 | 50.8 | 44.7 | 34.7 | 37.8 | 37.7 | 40.7 | 32.0 | 44.6 | 58.3 | 38.0 | 34.7 | 34.6 | 24.6 | 44.0 | 38.7 | 37.5 | 49.4 | 29.3 | 29.3 | 32.2 | 36.3 | 30.0 | 28.0 |
| Phi-3.5-Vision-Instruct | 24.6 | 23.5 | 20.7 | 34.7 | 46.4 | 57.5 | 42.7 | 32.0 | 37.9 | 42.9 | 40.0 | 25.3 | 25.4 | 15.6 | 33.3 | 25.3 | 31.3 | 41.7 | 28.7 | 18.7 | 36.1 | 23.4 | 46.7 | 44.0 | 31.2 | 40.4 | 28.7 | 17.3 | 32.7 | 38.9 | 27.3 | 30.7 |

Table 14: Evaluation results on AVSD. Each value represents the model's accuracy(%) for the corresponding row's task, evaluated on the problems of each dataset and skill type in the column.

| | TOTAL | | | | OCR | | | | Absolute position | | | | Adjacency | | | | Angle | | | | Area | | | | Boundary | | | |
|---|---|---|---|---|---|---|---|---|---|---|---|---|---|---|---|---|---|---|---|---|---|---|---|---|---|---|---|---|
| | overall | easy | medium | hard | overall | easy | medium | hard | overall | easy | medium | hard | overall | easy | medium | hard | overall | easy | medium | hard | overall | easy | medium | hard | overall | easy | medium | hard |
| o3 | 73.8 | 86.3 | 75.8 | 39.3 | 85.8 | 93.2 | 94.7 | 50.0 | 84.9 | 100.0 | 96.3 | 47.8 | 79.7 | 91.8 | 83.0 | 42.9 | 63.8 | 75.7 | 50.0 | 35.0 | 88.8 | 98.5 | 91.3 | 65.7 | 90.5 | 98.3 | 96.3 | 67.6 |
| Gemini 2.5 Flash | 70.5 | 83.7 | 70.8 | 37.3 | 89.1 | 96.6 | 96.6 | 59.5 | 81.4 | 100.0 | 100.0 | 30.4 | 71.5 | 75.7 | 80.0 | 38.1 | 63.2 | 78.0 | 42.6 | 40.0 | 76.3 | 93.4 | 82.2 | 34.3 | 81.1 | 96.6 | 87.5 | 44.1 |
| Gemini 2.5 Pro | 75.0 | 86.8 | 76.7 | 44.4 | 88.5 | 98.3 | 97.7 | 51.4 | 79.1 | 100.0 | 100.0 | 21.7 | 82.9 | 78.4 | 90.8 | 66.7 | 66.5 | 78.9 | 51.9 | 40.0 | 84.0 | 98.4 | 87.7 | 51.4 | 87.8 | 100.0 | 87.5 | 67.6 |
| GPT-4o | 61.9 | 74.2 | 61.4 | 35.2 | 88.5 | 98.6 | 96.1 | 50.0 | 79.1 | 94.4 | 96.3 | 34.8 | 69.9 | 75.5 | 79.2 | 33.3 | 44.9 | 53.2 | 31.5 | 35.0 | 74.0 | 90.8 | 71.0 | 48.6 | 75.0 | 93.3 | 64.8 | 58.8 |
| o1 | 67.9 | 79.4 | 69.6 | 38.0 | 90.7 | 100.0 | 96.1 | 58.8 | 84.9 | 100.0 | 100.0 | 43.5 | 82.1 | 91.8 | 84.9 | 52.4 | 55.7 | 67.6 | 29.6 | 60.0 | 85.2 | 93.8 | 87.0 | 65.7 | 87.8 | 100.0 | 92.6 | 58.8 |
| LLaVA-NeXT (13B) | 32.8 | 42.0 | 29.9 | 18.1 | 54.6 | 79.5 | 52.6 | 5.9 | 65.1 | 83.3 | 81.5 | 17.4 | 26.8 | 20.4 | 35.8 | 19.0 | 22.2 | 28.8 | 13.0 | 10.0 | 52.7 | 80.0 | 46.4 | 14.3 | 33.8 | 36.7 | 44.4 | 11.8 |
| LLaVA-NeXT (34B) | 38.0 | 49.1 | 34.8 | 19.7 | 65.6 | 91.8 | 65.8 | 8.8 | 73.3 | 100.0 | 81.5 | 21.7 | 26.8 | 26.5 | 28.3 | 23.8 | 25.9 | 23.4 | 33.3 | 20.0 | 53.3 | 80.0 | 49.3 | 11.4 | 35.1 | 45.0 | 35.2 | 17.6 |
| Math-LLaVA | 30.1 | 38.2 | 28.0 | 16.2 | 37.2 | 47.9 | 39.5 | 8.8 | 54.7 | 63.9 | 74.1 | 17.4 | 28.5 | 24.5 | 35.8 | 19.0 | 16.8 | 13.5 | 22.2 | 20.0 | 44.4 | 70.8 | 33.3 | 17.1 | 31.1 | 31.7 | 38.9 | 17.6 |
| LLaVA-OneVision (7B) | 42.2 | 55.3 | 38.5 | 20.4 | 66.1 | 90.4 | 60.5 | 26.5 | 75.6 | 100.0 | 88.9 | 21.7 | 54.5 | 63.3 | 56.6 | 28.6 | 41.6 | 52.3 | 31.5 | 10.0 | 57.4 | 87.7 | 46.4 | 22.9 | 42.6 | 53.3 | 46.3 | 11.8 |
| Qwen2.5-VL (7B) | 48.7 | 60.7 | 47.8 | 23.4 | 79.8 | 90.4 | 89.5 | 35.3 | 82.6 | 100.0 | 96.3 | 39.1 | 54.5 | 63.3 | 56.6 | 28.6 | 41.6 | 52.3 | 31.5 | 10.0 | 57.4 | 83.1 | 52.2 | 20.0 | 42.6 | 53.3 | 46.3 | 17.6 |
| Phi-3.5-Vision-Instruct | 35.6 | 45.0 | 34.3 | 17.2 | 55.7 | 78.1 | 53.9 | 11.8 | 76.7 | 97.2 | 88.9 | 30.4 | 30.9 | 24.5 | 39.6 | 23.8 | 25.9 | 24.3 | 29.6 | 25.0 | 40.2 | 61.5 | 31.9 | 17.1 | 38.5 | 50.0 | 46.3 | 5.9 |

| Cardinal | | | | Cardinal Direction | | | | Color | | | | Congruence | | | | Connectedness | | | | Convexity | | | | Coordinate | | | |
|---|---|---|---|---|---|---|---|---|---|---|---|---|---|---|---|---|---|---|---|---|---|---|---|---|---|---|---|
| overall | easy | medium | hard | overall | easy | medium | hard | overall | easy | medium | hard | overall | easy | medium | hard | overall | easy | medium | hard | overall | easy | medium | hard | overall | easy | medium | hard |
| 72.4 | 89.6 | 74.2 | 38.6 | 90.0 | 100.0 | 90.6 | 70.0 | 88.3 | 96.5 | 92.9 | 46.9 | 62.6 | 80.2 | 50.0 | 52.3 | 73.3 | 78.8 | 77.1 | 53.6 | 80.3 | 87.1 | 79.4 | 61.9 | 65.9 | 71.7 | 71.4 | 30.5 |
| 67.3 | 89.8 | 64.8 | 44.7 | 80.9 | 97.4 | 90.6 | 35.0 | 88.6 | 97.4 | 87.1 | 46.9 | 61.1 | 77.6 | 61.5 | 35.6 | 69.1 | 77.3 | 76.8 | 35.7 | 73.7 | 87.1 | 67.6 | 45.5 | 71.3 | 84.3 | 77.9 | 25.0 |
| 79.9 | 98.3 | 82.4 | 51.1 | 88.8 | 100.0 | 96.9 | 55.0 | 89.2 | 97.4 | 91.8 | 53.1 | 65.0 | 83.6 | 58.2 | 37.8 | 79.2 | 90.9 | 88.9 | 35.7 | 80.5 | 90.3 | 79.4 | 54.5 | 82.4 | 98.0 | 86.8 | 35.0 |
| 63.6 | 83.1 | 65.6 | 25.0 | 80.0 | 89.5 | 90.6 | 45.0 | 85.7 | 93.9 | 90.5 | 43.8 | 46.8 | 77.8 | 23.1 | 31.8 | 56.0 | 65.4 | 55.7 | 39.3 | 65.8 | 75.8 | 61.8 | 42.9 | 49.7 | 58.5 | 52.9 | 15.0 |
| 68.2 | 90.9 | 64.5 | 36.4 | 83.3 | 92.1 | 96.9 | 45.0 | 84.4 | 93.0 | 86.9 | 46.9 | 49.8 | 72.8 | 35.9 | 31.8 | 57.3 | 75.0 | 61.4 | 14.3 | 68.4 | 83.9 | 64.7 | 28.6 | 55.2 | 54.7 | 65.7 | 20.0 |
| 34.6 | 62.3 | 24.7 | 6.8 | 43.3 | 52.6 | 37.5 | 35.0 | 52.4 | 67.0 | 41.7 | 28.1 | 22.2 | 29.6 | 17.9 | 15.9 | 29.3 | 48.1 | 20.0 | 17.9 | 41.9 | 48.4 | 38.2 | 28.6 | 18.9 | 13.2 | 25.7 | 10.0 |
| 37.4 | 54.5 | 33.3 | 15.9 | 55.6 | 84.2 | 43.8 | 20.0 | 59.7 | 70.4 | 58.3 | 25.0 | 26.1 | 30.9 | 23.1 | 22.7 | 32.7 | 46.2 | 27.1 | 21.4 | 38.5 | 51.6 | 32.4 | 9.5 | 28.7 | 32.1 | 30.0 | 15.0 |
| 33.6 | 54.5 | 28.0 | 9.1 | 38.9 | 55.3 | 31.2 | 20.0 | 45.9 | 65.2 | 32.1 | 12.5 | 19.2 | 23.5 | 17.9 | 13.6 | 21.3 | 34.6 | 12.9 | 17.9 | 36.8 | 51.6 | 23.5 | 14.3 | 22.4 | 20.8 | 24.3 | 20.0 |
| 41.6 | 66.2 | 36.6 | 9.1 | 56.7 | 76.3 | 56.2 | 20.0 | 67.5 | 82.6 | 61.9 | 28.1 | 28.1 | 40.7 | 20.5 | 18.2 | 28.7 | 28.8 | 32.9 | 17.9 | 47.9 | 62.9 | 29.4 | 33.3 | 30.8 | 34.0 | 35.7 | 5.0 |
| 56.5 | 76.6 | 55.9 | 22.7 | 65.6 | 78.9 | 68.8 | 35.0 | 71.0 | 85.2 | 66.7 | 31.2 | 33.0 | 49.4 | 25.6 | 15.9 | 38.7 | 46.2 | 41.4 | 17.9 | 51.3 | 72.6 | 32.4 | 19.0 | 50.3 | 58.5 | 55.7 | 10.0 |
| 37.9 | 67.5 | 28.0 | 6.8 | 47.8 | 63.2 | 50.0 | 15.0 | 55.0 | 74.8 | 41.7 | 18.8 | 22.2 | 28.4 | 20.5 | 13.6 | 25.3 | 36.5 | 25.7 | 3.6 | 40.2 | 50.0 | 44.1 | 4.8 | 30.8 | 41.5 | 31.4 | 0.0 |

| Curvature | | | | Direction | | | | Interior | | | | Intersection | | | | Length | | | | Line | | | | Overlap | | | |
|---|---|---|---|---|---|---|---|---|---|---|---|---|---|---|---|---|---|---|---|---|---|---|---|---|---|---|---|
| overall | easy | medium | hard | overall | easy | medium | hard | overall | easy | medium | hard | overall | easy | medium | hard | overall | easy | medium | hard | overall | easy | medium | hard | overall | easy | medium | hard |
| 80.9 | 100.0 | 69.0 | 79.4 | 69.6 | 89.5 | 75.0 | 20.8 | 88.7 | 98.5 | 89.1 | 55.0 | 67.2 | 83.3 | 63.9 | 45.0 | 77.0 | 87.5 | 79.6 | 52.0 | 78.1 | 87.5 | 81.4 | 70.0 | 81.3 | 94.6 | 84.7 | 41.7 |
| 70.4 | 94.3 | 66.1 | 50.0 | 69.6 | 88.9 | 73.3 | 29.2 | 82.0 | 95.4 | 86.2 | 25.0 | 56.0 | 71.4 | 53.5 | 33.3 | 68.9 | 91.1 | 64.0 | 40.0 | 80.7 | 83.3 | 86.0 | 60.0 | 71.9 | 91.1 | 69.5 | 33.3 |
| 72.8 | 94.3 | 71.0 | 50.0 | 72.6 | 91.7 | 80.0 | 20.8 | 89.3 | 96.9 | 89.1 | 65.0 | 61.9 | 69.0 | 69.0 | 23.8 | 71.3 | 86.7 | 76.0 | 36.0 | 85.9 | 84.7 | 90.7 | 80.0 | 73.4 | 92.9 | 69.5 | 37.5 |
| 72.5 | 84.6 | 67.2 | 67.6 | 53.6 | 89.5 | 48.7 | 12.5 | 84.0 | 92.4 | 89.1 | 40.0 | 51.5 | 66.7 | 50.0 | 25.0 | 62.3 | 68.8 | 65.3 | 44.0 | 68.9 | 77.8 | 60.5 | 55.0 | 74.1 | 91.1 | 71.2 | 41.7 |
| 77.1 | 92.3 | 74.1 | 64.7 | 66.7 | 84.2 | 75.0 | 12.5 | 88.0 | 95.5 | 95.3 | 40.0 | 59.0 | 73.8 | 61.1 | 20.0 | 67.2 | 79.2 | 65.3 | 48.0 | 81.5 | 81.9 | 88.4 | 65.0 | 76.3 | 94.6 | 79.7 | 25.0 |
| 26.7 | 28.2 | 27.6 | 23.5 | 23.2 | 34.2 | 17.1 | 25.0 | 41.3 | 50.0 | 40.6 | 15.0 | 20.9 | 23.8 | 20.8 | 15.0 | 40.2 | 47.9 | 46.9 | 12.0 | 37.0 | 47.2 | 27.9 | 20.0 | 33.8 | 44.6 | 25.4 | 29.2 |
| 28.2 | 48.7 | 20.7 | 17.6 | 29.0 | 44.7 | 26.3 | 12.5 | 59.3 | 75.8 | 51.6 | 30.0 | 24.6 | 35.7 | 23.6 | 5.0 | 39.3 | 50.0 | 38.8 | 20.0 | 54.1 | 63.9 | 41.9 | 45.0 | 47.5 | 69.6 | 40.7 | 12.5 |
| 18.3 | 12.8 | 25.9 | 11.8 | 27.5 | 47.4 | 22.4 | 12.5 | 39.3 | 45.5 | 42.2 | 10.0 | 23.9 | 26.2 | 23.6 | 20.0 | 38.5 | 47.9 | 40.8 | 16.0 | 29.6 | 37.5 | 16.3 | 30.0 | 30.2 | 35.7 | 23.7 | 33.3 |
| 42.0 | 74.4 | 32.8 | 20.6 | 40.6 | 73.7 | 30.3 | 20.8 | 69.3 | 84.8 | 67.2 | 25.0 | 25.1 | 33.3 | 22.2 | 5.0 | 50.8 | 66.7 | 49.0 | 24.0 | 52.6 | 65.3 | 34.9 | 45.0 | 41.0 | 51.8 | 37.3 | 25.0 |
| 45.0 | 76.9 | 32.8 | 29.4 | 50.0 | 81.6 | 42.1 | 25.0 | 77.3 | 92.4 | 76.6 | 30.0 | 21.6 | 33.3 | 16.7 | 15.0 | 53.3 | 58.3 | 59.2 | 32.0 | 54.8 | 59.7 | 60.5 | 25.0 | 46.8 | 66.1 | 35.6 | 29.2 |
| 32.1 | 46.2 | 29.3 | 20.6 | 27.5 | 55.3 | 21.1 | 4.2 | 43.3 | 50.0 | 37.5 | 40.0 | 32.8 | 28.6 | 34.7 | 35.0 | 42.6 | 54.2 | 49.0 | 8.0 | 38.5 | 45.8 | 25.6 | 40.0 | 34.5 | 32.1 | 37.3 | 33.3 |

| Ordinal | | | | Orientation | | | | Orthogonality | | | | Parallel | | | | Point | | | | Reflection | | | | Relative Position | | | | Rotation | | | |
|---|---|---|---|---|---|---|---|---|---|---|---|---|---|---|---|---|---|---|---|---|---|---|---|---|---|---|---|---|---|---|---|
| overall | easy | medium | hard | overall | easy | medium | hard | overall | easy | medium | hard | overall | easy | medium | hard | overall | easy | medium | hard | overall | easy | medium | hard | overall | easy | medium | hard | overall | easy | medium | hard |
| 80.6 | 100.0 | 96.4 | 36.3 | 53.7 | 64.4 | 47.6 | 35.0 | 70.9 | 65.7 | 80.6 | 65.0 | 53.1 | 56.2 | 65.1 | 29.0 | 89.8 | 100.0 | 96.8 | 58.3 | 59.8 | 72.7 | 55.7 | 36.0 | 70.4 | 88.6 | 48.6 | 68.4 | 23.9 | 57.1 | 14.0 | 12.0 |
| 82.1 | 100.0 | 89.3 | 45.5 | 47.5 | 59.6 | 46.5 | 15.0 | 53.5 | 60.6 | 54.5 | 40.0 | 59.2 | 80.8 | 53.5 | 16.1 | 89.5 | 98.0 | 100.0 | 54.2 | 48.2 | 52.7 | 51.4 | 24.0 | 67.3 | 81.8 | 52.9 | 60.0 | 33.8 | 47.6 | 43.7 | 14.0 |
| 87.3 | 100.0 | 94.6 | 57.6 | 51.7 | 59.6 | 58.1 | 15.0 | 57.0 | 69.7 | 57.6 | 35.0 | 61.2 | 79.5 | 62.8 | 16.1 | 89.5 | 100.0 | 100.0 | 50.0 | 56.8 | 62.3 | 62.9 | 16.0 | 72.4 | 90.9 | 50.0 | 70.0 | 33.1 | 61.9 | 38.0 | 14.0 |
| 68.7 | 91.3 | 76.4 | 24.2 | 36.4 | 30.5 | 42.9 | 40.0 | 45.3 | 60.0 | 48.4 | 15.0 | 31.3 | 35.6 | 34.9 | 16.1 | 81.5 | 94.3 | 90.3 | 41.7 | 35.2 | 44.2 | 37.1 | 0.0 | 69.4 | 90.9 | 48.6 | 57.9 | 23.9 | 31.4 | 24.6 | 18.0 |
| 79.1 | 100.0 | 92.7 | 27.3 | 50.4 | 59.3 | 47.6 | 30.0 | 51.2 | 51.4 | 67.7 | 25.0 | 40.8 | 39.7 | 55.8 | 22.6 | 86.1 | 92.5 | 96.8 | 58.3 | 45.7 | 62.3 | 39.2 | 40.0 | 63.0 | 75.0 | 40.0 | 78.9 | 31.0 | 54.3 | 33.3 | 12.0 |
| 25.4 | 41.3 | 20.0 | 12.1 | 32.2 | 25.4 | 28.6 | 60.0 | 12.8 | 17.1 | 0.0 | 25.0 | 12.9 | 15.1 | 7.0 | 16.1 | 34.3 | 41.5 | 41.9 | 8.3 | 27.6 | 26.0 | 26.8 | 36.0 | 36.7 | 47.7 | 34.3 | 15.8 | 19.7 | 20.0 | 15.8 | 24.0 |
| 29.1 | 47.8 | 23.6 | 12.1 | 36.4 | 32.2 | 28.6 | 65.0 | 19.8 | 28.6 | 19.4 | 5.0 | 13.6 | 9.6 | 23.3 | 9.7 | 47.2 | 52.8 | 61.3 | 16.7 | 29.1 | 37.7 | 20.6 | 36.0 | 41.8 | 59.1 | 37.1 | 10.5 | 13.4 | 20.0 | 7.0 | 16.0 |
| 22.4 | 32.6 | 21.8 | 9.1 | 34.7 | 37.3 | 31.0 | 35.0 | 14.0 | 22.9 | 6.5 | 10.0 | 22.4 | 24.7 | 14.0 | 29.0 | 34.3 | 45.3 | 41.9 | 0.0 | 21.6 | 31.2 | 15.5 | 16.0 | 31.6 | 43.2 | 28.6 | 10.5 | 32.4 | 48.6 | 33.3 | 20.0 |
| 24.6 | 41.3 | 20.0 | 9.1 | 33.1 | 25.4 | 31.0 | 60.0 | 23.3 | 34.5 | 16.1 | 15.0 | 12.9 | 11.0 | 14.0 | 16.1 | 45.4 | 50.9 | 54.5 | 45.0 | 34.2 | 33.8 | 40.2 | 12.0 | 43.9 | 63.6 | 31.4 | 21.1 | 30.3 | 45.7 | 28.1 | 21.1 |
| 57.5 | 84.8 | 56.4 | 21.2 | 29.8 | 33.9 | 28.6 | 20.0 | 31.4 | 48.6 | 19.4 | 20.0 | 25.2 | 20.5 | 39.5 | 16.1 | 61.1 | 54.7 | 80.6 | 50.0 | 26.1 | 33.8 | 20.6 | 24.0 | 46.9 | 68.2 | 34.3 | 21.1 | 29.6 | 34.3 | 28.1 | 28.0 |
| 28.4 | 34.8 | 34.5 | 9.1 | 29.8 | 27.1 | 31.0 | 35.0 | 26.7 | 45.7 | 16.1 | 10.0 | 21.1 | 11.0 | 37.2 | 22.6 | 43.5 | 47.2 | 61.3 | 12.5 | 28.6 | 33.8 | 23.7 | 32.0 | 27.6 | 40.9 | 20.0 | 10.5 | 21.1 | 22.9 | 31.6 | 8.0 |

| Rotational Symmetry | | | | Shape | | | | Sharpness | | | | Similarity | | | | Symbol | | | | Tangency | | | | Texture | | | | Width | | | |
|---|---|---|---|---|---|---|---|---|---|---|---|---|---|---|---|---|---|---|---|---|---|---|---|---|---|---|---|---|---|---|---|
| overall | easy | medium | hard | overall | easy | medium | hard | overall | easy | medium | hard | overall | easy | medium | hard | overall | easy | medium | hard | overall | easy | medium | hard | overall | easy | medium | hard | overall | easy | medium | hard |
| 73.5 | 100.0 | 71.8 | 25.0 | 91.2 | 98.1 | 93.2 | 55.0 | 83.3 | 93.5 | 94.6 | 44.4 | 65.6 | 86.0 | 68.4 | 27.3 | 83.5 | 92.0 | 92.9 | 38.1 | 53.2 | 60.2 | 49.1 | 33.3 | 76.3 | 86.0 | 73.5 | 42.9 | 70.7 | 68.5 | 77.5 | 52.2 |
| 68.9 | 97.6 | 66.2 | 20.0 | 90.9 | 97.9 | 89.9 | 70.0 | 75.4 | 87.8 | 89.1 | 29.6 | 53.7 | 67.9 | 61.1 | 9.5 | 80.3 | 88.6 | 95.2 | 19.0 | 43.9 | 59.6 | 31.7 | 13.6 | 73.7 | 84.2 | 72.9 | 38.1 | 63.7 | 74.1 | 58.8 | 56.5 |
| 77.3 | 97.6 | 78.9 | 30.0 | 87.0 | 97.9 | 87.3 | 60.0 | 80.3 | 89.8 | 93.5 | 40.7 | 62.0 | 78.6 | 70.8 | 9.5 | 84.3 | 93.2 | 93.5 | 38.1 | 52.6 | 64.0 | 38.3 | 45.5 | 82.7 | 90.8 | 81.4 | 57.1 | 63.7 | 70.4 | 57.5 | 69.6 |
| 67.4 | 68.3 | 76.1 | 35.0 | 84.9 | 100.0 | 83.8 | 50.0 | 72.2 | 87.1 | 78.4 | 29.6 | 55.7 | 86.0 | 40.4 | 36.4 | 70.9 | 84.0 | 80.4 | 14.3 | 27.5 | 31.2 | 21.1 | 28.6 | 70.5 | 81.4 | 65.3 | 38.1 | 58.6 | 66.7 | 58.8 | 39.1 |
| 76.5 | 100.0 | 74.6 | 35.0 | 87.0 | 96.2 | 87.8 | 60.0 | 74.6 | 88.7 | 81.1 | 33.3 | 54.9 | 69.8 | 50.9 | 36.4 | 79.5 | 92.0 | 92.9 | 14.3 | 37.4 | 43.0 | 36.8 | 14.3 | 67.9 | 81.4 | 61.2 | 28.6 | 66.2 | 63.0 | 75.0 | 43.5 |
| 15.9 | 29.3 | 8.5 | 15.0 | 54.8 | 67.3 | 52.7 | 30.0 | 31.7 | 45.2 | 27.0 | 7.4 | 26.2 | 30.2 | 29.8 | 9.1 | 38.6 | 48.0 | 44.6 | 0.0 | 19.9 | 24.7 | 14.0 | 14.3 | 31.4 | 36.0 | 32.7 | 9.5 | 35.7 | 50.0 | 28.7 | 26.1 |
| 33.3 | 51.2 | 25.4 | 25.0 | 56.8 | 75.0 | 52.7 | 25.0 | 45.2 | 56.5 | 45.9 | 18.5 | 28.7 | 30.2 | 21.1 | 45.5 | 36.2 | 46.0 | 39.3 | 4.8 | 29.8 | 30.1 | 29.8 | 28.6 | 32.7 | 40.7 | 22.4 | 23.8 | 35.7 | 42.6 | 36.2 | 17.4 |
| 19.7 | 29.3 | 9.9 | 35.0 | 47.3 | 69.2 | 37.8 | 25.0 | 34.1 | 45.2 | 29.7 | 14.8 | 17.2 | 9.3 | 26.3 | 9.1 | 36.6 | 46.0 | 48.2 | 0.0 | 25.1 | 25.8 | 22.8 | 28.6 | 24.4 | 38.4 | 8.2 | 4.8 | 36.3 | 33.3 | 46.2 | 8.7 |
| 31.1 | 65.9 | 15.5 | 15.0 | 71.2 | 90.4 | 66.2 | 40.0 | 52.4 | 72.6 | 37.8 | 25.9 | 28.7 | 41.9 | 15.8 | 36.4 | 62.2 | 66.0 | 76.8 | 14.3 | 24.6 | 31.2 | 17.5 | 14.3 | 32.7 | 44.2 | 24.5 | 4.8 | 39.5 | 44.4 | 40.0 | 26.1 |
| 43.9 | 46.3 | 45.1 | 35.0 | 71.9 | 88.5 | 73.0 | 25.0 | 50.8 | 67.7 | 54.1 | 7.4 | 37.7 | 30.2 | 47.4 | 27.3 | 58.3 | 66.0 | 69.6 | 9.5 | 29.2 | 35.5 | 21.1 | 23.8 | 49.4 | 66.3 | 34.7 | 14.3 | 36.3 | 31.5 | 47.5 | 8.7 |
| 23.5 | 43.9 | 11.3 | 25.0 | 57.5 | 82.7 | 47.3 | 30.0 | 42.9 | 53.2 | 40.5 | 22.2 | 15.6 | 11.6 | 15.8 | 22.7 | 41.7 | 46.0 | 51.8 | 4.8 | 23.4 | 24.7 | 21.1 | 23.8 | 40.4 | 50.0 | 36.7 | 9.5 | 38.9 | 40.7 | 41.2 | 26.1 |

Table 15: Full details of evaluation results on AVSD-h. Each value represents the accuracy of the model of its row, on problems with the difficulty and for the skill of its column.

Table 16: Summary of task descriptions used in ASVD-s and AVSD-c.

| Key | Description |
| --- | --- |
| **absolute_position** | |
| abs_position_right | Determine whether the given point is located on the right side of the figure. |
| abs_position_left | Determine whether the given point is located on the left side of the figure. |
| abs_position_top | Determine whether the given point is located on the top side of the figure. |
| abs_position_bottom | Determine whether the given point is located on the bottom side of the figure. |
| abs_position_topright | Determine whether the given point is located in the top right quadrant of the figure. |
| abs_position_topleft | Determine whether the given point is located in the top left quadrant of the figure. |
| abs_position_bottomright | Determine whether the given point is located in the bottom right quadrant of the figure. |
| abs_position_bottomleft | Determine whether the given point is located in the bottom left quadrant of the figure. |
| C_abs_position_right | Determine whether the given colored point(s) are located on the right side of the figure. |
| C_abs_position_left | Determine whether the given colored point(s) are located on the left side of the figure. |
| C_abs_position_top | Determine whether the given colored point(s) are located on the top side of the figure. |
| C_abs_position_bottom | Determine whether the given colored point(s) are located on the bottom side of the figure. |
| C_abs_position_topright | Determine whether the given colored point(s) are located in the top right quadrant of the figure. |
| C_abs_position_topleft | Determine whether the given colored point(s) are located in the top left quadrant of the figure. |
| C_abs_position_bottomright | Determine whether the given colored point(s) are located in the bottom right quadrant of the figure. |

| Key | Description |
| --- | --- |
| C_abs_position_bottomleft | Determine whether the given colored point(s) are located in the bottom left quadrant of the figure. |
| abs_position_line_right | Determine whether the given line is located on the right side of the figure. |
| abs_position_line_left | Determine whether the given line is located on the left side of the figure. |
| abs_position_line_top | Determine whether the given line is located on the top side of the figure. |
| abs_position_line_bottom | Determine whether the given line is located on the bottom side of the figure. |
| abs_position_line_tr | Determine whether the given line is located in the top right quadrant of the figure. |
| abs_position_line_tl | Determine whether the given line is located in the top left quadrant of the figure. |
| abs_position_line_br | Determine whether the given line is located in the bottom right quadrant of the figure. |
| abs_position_line_bl | Determine whether the given line is located in the bottom left quadrant of the figure. |
| C_abs_position_line_right | Determine whether the given colored line(s) are located on the right side of the figure. |
| C_abs_position_line_left | Determine whether the given colored line(s) are located on the left side of the figure. |
| C_abs_position_line_top | Determine whether the given colored line(s) are located on the top side of the figure. |
| C_abs_position_line_bottom | Determine whether the given colored line(s) are located on the bottom side of the figure. |
| C_abs_position_line_tr | Determine whether the given colored line(s) are located in the top right quadrant of the figure. |
| C_abs_position_line_tl | Determine whether the given colored line(s) are located in the top left quadrant of the figure. |
| C_abs_position_line_br | Determine whether the given colored line(s) are located in the bottom right quadrant of the figure. |

| Key | Description |
| --- | --- |
| `C_abs_position_line_bl` | Determine whether the given colored line(s) are located in the bottom left quadrant of the figure. |
| `abs_position_circle_right` | Determine whether the given circle is located on the right side of the figure. |
| `abs_position_circle_left` | Determine whether the given circle is located on the left side of the figure. |
| `abs_position_circle_top` | Determine whether the given circle is located on the top side of the figure. |
| `abs_position_circle_bottom` | Determine whether the given circle is located on the bottom side of the figure. |
| `abs_position_circle_tr` | Determine whether the given circle is located in the top right quadrant of the figure. |
| `abs_position_circle_tl` | Determine whether the given circle is located in the top left quadrant of the figure. |
| `abs_position_circle_br` | Determine whether the given circle is located in the bottom right quadrant of the figure. |
| `abs_position_circle_bl` | Determine whether the given circle is located in the bottom left quadrant of the figure. |
| `C_abs_position_circle_right` | Determine whether the given colored circle is located on the right side of the figure. |
| `C_abs_position_circle_left` | Determine whether the given colored circle is located on the left side of the figure. |
| `C_abs_position_circle_top` | Determine whether the given colored circle is located on the top side of the figure. |
| `C_abs_position_circle_bottom` | Determine whether the given colored circle is located on the bottom side of the figure. |
| `C_abs_position_circle_tr` | Determine whether the given colored circle is located in the top right quadrant of the figure. |
| `C_abs_position_circle_tl` | Determine whether the given colored circle is located in the top left quadrant of the figure. |
| `C_abs_position_circle_br` | Determine whether the given colored circle is located in the bottom right quadrant of the figure. |
| `C_abs_position_circle_bl` | Determine whether the given colored circle is located in the bottom left quadrant of the figure. |

| Key | Description |
|---|---|
| `abs_position_object` | Determine which object, among several objects, is located in a specified direction of the image. |
| `abs_position_object_colored` | Determine which object, and its color, among several objects is located in a specified direction of the image. |
| `abs_position_object_quadrant` | Determine which object, among several objects, is located in a specified quadrant of the image. |
| `abs_position_object_colored_quadrant` | Determine which object, and its color, among several objects is located in a specified quadrant of the image. |
| **adjacency** | |
| `adjacency1` | Determine which region colors in a colored grid are adjacent by a shared edge to a specified region. |
| `adjacency2` | Determine which elements in a matrix are immediately adjacent (sharing an edge) to a designated element. |
| **angle** | |
| `angle1` | Determine whether the constructed angle is acute. |
| `C_angle1` | Determine whether the colored constructed angle is acute. |
| `angle2` | Determine whether the constructed angle is obtuse. |
| `C_angle2` | Determine whether the colored constructed angle is obtuse. |
| `angle3` | Determine whether the constructed angle is a right angle. |
| `C_angle3` | Determine whether the colored constructed angle is a right angle. |
| `angle4` | Determine whether the given triangle is acute-angled. |
| `C_angle4` | Determine whether the colored triangle is acute-angled. |

| Key | Description |
| --- | --- |
| angle5 | Determine whether the given triangle is obtuse-angled. |
| C_angle5 | Determine whether the colored triangle is obtuse-angled. |
| angle6 | Determine whether the given triangle is right-angled. |
| C_angle6 | Determine whether the colored triangle is right-angled. |
| angle7 | Compare two angles and determine which one is acute. |
| C_angle7 | Compare two colored angles and determine which one is acute. |
| angle7-2 | Compare two angles (alternate format) and determine which is acute. |
| C_angle7-2 | Compare two colored angles (alternate format) and determine which is acute. |
| angle8 | Classify three angles as acute, obtuse, or right. |
| C_angle8 | Classify three colored angles as acute, obtuse, or right. |
| angle9 | Identify the acute angles among four given angles. |
| C_angle9 | Identify the acute angles among four colored angles. |
| angle10 | Compare four angles and determine which are obtuse and which are acute. |
| C_angle10 | Compare four colored angles and determine which are obtuse and which are acute. |
| angle11 | Identify the right angle in the given triangle. |
| C_angle11 | Identify the right angle in the colored triangle. |
| angle12 | Determine whether the given triangle contains an obtuse angle. |
| C_angle12 | Determine whether the colored triangle contains an obtuse angle. |
| angle13 | Evaluate the triangle to classify its angles as acute or obtuse. |

| Key | Description |
| --- | --- |
| C_angle13 | Evaluate the colored triangle to classify its angles as acute or obtuse. |
| angle14 | Determine if the sum of two given angles is less than or equal to 90°. |
| C_angle14 | Determine if the sum of two colored angles is less than or equal to 90°. |
| angle15 | Determine if the sum of two given angles is greater than 90°. |
| C_angle15 | Determine if the sum of two colored angles is greater than 90°. |
| angle16 | Verify that all angles in the set are acute. |
| C_angle16 | Verify that all colored angles in the set are acute. |
| angle17 | Identify which angles among four given angles are obtuse. |
| C_angle17 | Identify which colored angles among four given angles are obtuse. |
| angle18-2-1 | Determine which of the two given lines bisects the specified angle. |
| angle18-3-1 | Determine which of the provided lines trisects the specified angle. |
| angle18-2-2 | Determine which line bisects the set of lines forming the specified angle (variant). |
| angle18-3-2 | Determine which line trisects the set of lines forming the specified angle (variant). |
| angle18-2-3 | Identify the color of the line that bisects the given angle. |
| angle18-3-3 | Identify the colors of the lines that trisect the given angle. |
| angle19-smallest-1 | Determine which of the two given angles is the smallest. |
| angle19-largest-1 | Determine which of the two given angles is the largest. |
| angle19-smallest-2 | Determine the smallest angle in the figure (alternate format). |
| angle19-largest-2 | Determine the largest angle in the figure (alternate format). |

| Key | Description |
| --- | --- |
| angle20-1 | Identify the angle that is equal in measure to the reference angle. |
| angle20-2 | Identify the angle that matches the measure of the reference angle (alternate format). |
| angle22-1 | Determine which labeled angle in the matrix is the largest. |
| angle22-2 | Determine which colored angle in the matrix is the largest. |
| **area** | |
| area_1_1 | Determine which category in a four-category pie chart has the largest ratio. |
| area_1_2 | Determine which category in a five-category pie chart has the largest ratio. |
| area_1_3 | Determine which category in a three-category pie chart has the largest ratio. |
| area_1_4 | Determine which category in a six-category pie chart has the largest ratio. |
| area_1_5 | Determine which category in a two-category pie chart has the largest ratio. |
| area_2_1 | Compare two areas and choose the adjective 'larger' or 'smaller' to describe their sizes. |
| area_2_2 | Compare three areas and choose the correct adjectives to describe their relative sizes. |
| **boundary** | |
| boundary1 | Determine if the colored line precisely forms the boundary of a shape filled with a specific color (Yes). |
| boundary2 | Determine if the colored line accurately outlines the boundary of a shape with a given colored interior (No). |
| boundary3 | Determine whether the shape has an outline that is colored differently from its filled interior (Yes). |
| boundary4 | Determine whether the shape's outline is indistinguishable in color from its interior (No). |

| Key | Description |
| --- | --- |
| `boundary5` | Identify the label of the line segment that divides one area from an adjacent area. |
| `boundary6` | Determine the color of the outermost boundary among overlapping shapes. |
| **cardinal** | |
| `cardinal_1` | Count the total number of points present in the image. |
| `cardinal_2` | Count the total number of line segments present in the image. |
| `cardinal_3` | Count the occurrences of a specified shape type (denoted by $<1>$) in the image. |
| `cardinal_4` | Determine the number of unique shapes depicted in the image. |
| `cardinal_5` | Count the distinct colors used in the image, excluding the background. |
| `cardinal_6` | Count the occurrences of a specified letter (denoted by $<1>$) in the image, choosing from the options 3, 4, 5, or 6. |
| **cardinal_direction** | |
| `cardinal_dir_horizontal` | Determine whether the target object is located on the left or right side of the reference object. |
| `cardinal_dir_horizontal_colored` | Determine, using color cues, whether the target object is located on the left or right side of the reference object. |
| `cardinal_dir_vertical` | Determine whether the target object is located on the top or bottom side of the reference object. |
| `cardinal_dir_vertical_colored` | Determine, using color cues, whether the target object is located on the top or bottom side of the reference object. |
| `cardinal_dir_both` | Determine the diagonal quadrant (top right, top left, bottom right, or bottom left) in which the target object is located relative to the reference object. |

| Key | Description |
|---|---|
| `cardinal_dir_both_colored` | Determine, using color cues, the diagonal quadrant (top right, top left, bottom right, or bottom left) in which the target object is located relative to the reference object. |
| `cardinal_direction_obj_horizontal` | Determine which of two objects is positioned to the left or right of the other. |
| `cardinal_direction_obj_horizontal_colored` | Determine, using color cues, which of two objects is positioned to the left or right of the other. |
| `cardinal_direction_obj_vertical` | Determine which of two objects is positioned above or below the other. |
| `cardinal_direction_obj_vertical_colored` | Determine, using color cues, which of two objects is positioned above or below the other. |
| `cardinal_direction_obj_both` | Determine which of two objects is located in a specific diagonal quadrant relative to the other. |
| `cardinal_direction_obj_both_colored` | Determine, using color cues, which of two objects is located in a specific diagonal quadrant relative to the other. |
| **color** | |
| `color1` | Determine the color of a given object. |
| `color2_a` | Determine the colors of multiple objects of the same type and compare them. |
| `color2_b` | Identify and list the colors that compose a polygon's sides. |
| `color3` | Extrapolate the gradation of colors to choose the appropriate color for a missing cell. |
| `color5` | Determine the relative brightness of objects by selecting the brightest or darkest one. |
| `color6` | Determine the relative saturation of objects by choosing the one with the highest or lowest saturation. |
| `color7` | Determine the background color of the image. |
| `color8` | Determine the color of the shape that an arrow is pointing to. |
| `color9` | Identify the color that is most similar or most different compared to another given color. |

| Key | Description |
|---|---|
| color10 | Choose the color that is significantly distinct from the others in the image. |
| color11 | Identify the color of a specified line. |
| color12 | Determine the number of lines interrupting a gradation of colors. |
| color13 | Identify the distinct color among several colors where one differs from the others. |
| **congruence** | |
| congruence1 | Determine whether the two given figures are congruent. |
| congruence2 | Determine whether the two given figures are not congruent. |
| congruence3 | Determine which side in one of the congruent figures corresponds to a specified side in the other. |
| congruence4 | Determine which vertex in one of the congruent figures corresponds to a specified vertex in the other. |
| **connectedness** | |
| connectedness1 | Determine whether the line connecting two points of a specified color is continuous. |
| connectedness2 | Determine whether two specified points (by label) are connected by an unbroken line. |
| connectedness3 | Determine whether the line connecting two points of a specified color is not continuous. |
| connectedness4 | Determine whether two specified points (by label) are not connected by an unbroken line. |
| connectedness5 | Count the total number of connected shapes in the given figure. |
| connectedness6 | Determine the number of connected components (or 'islands') in a graph of nodes. |
| connectedness7 | Provide a list of connected components in the graph, formatted as specified. |
| connectedness8 | Identify the nodes that are directly connected to a specified node in the graph. |
| **convexity** | |

| Key | Description |
| --- | --- |
| convexity1 | Determine whether the given polygon is convex or concave. |
| convexity2 | Determine which polygon is convex or concave from a set of options, and answer with its label. |
| convexity3 | Determine the color of the polygon that is convex or concave among the provided options. |
| convexity4 | Determine all polygons that are convex or concave from the given list by selecting their labels. |
| convexity5 | Determine the colors of all polygons that are convex or concave among the given options. |
| convexity6 | Determine whether the given function is convex or concave. |
| convexity7 | Determine which rectangular regions (marked by shaded boxes) in the function are convex or concave based on their labels. |
| convexity8 | Determine which regions, divided by specific points in the function, are convex or concave based on their labels. |
| convexity9 | Determine whether the given lens is convex or concave. |
| convexity10 | Determine which lens is convex or concave from a set of options, and answer with its label. |
| convexity11 | Determine the color of the lens that is convex or concave from the provided options. |
| convexity12 | Determine all lenses that are convex or concave from the given list by selecting their labels. |
| convexity13 | Determine the colors of all lenses that are convex or concave among the provided options. |
| **coordinate** | |
| coordinate2 | Identify the (x, y) coordinates of a specified point in the figure. |
| coordinate3 | Identify the (x, y) coordinates of the point that has a given color. |

| Key | Description |
| --- | --- |
| coordinate4 | Determine the label of the point that shares the same specified coordinate (x or y) as another point. |
| coordinate5 | Determine the color of the point that shares the same specified coordinate with a point of a given color. |
| coordinate6 | Identify the pair of points that share the same specified coordinate (x or y). |
| coordinate7 | Identify the colors of the two points that share the same specified coordinate (x or y). |
| coordinate8 | Determine the point that has a specific combination of coordinates as indicated. |
| coordinate9 | Determine the color of the point that has a specific combination of coordinates as indicated. |
| coordinate10 | Determine the quadrant in which a specified point lies. |
| coordinate11 | Determine the quadrant in which a point with a given color is located. |
| coordinate12 | Determine the specified intercept (e.g., y-intercept) of a graphed function. |
| coordinate13 | Determine the quadrant in which a given shape is located. |
| coordinate14 | Determine all quadrants that a given shape spans. |
| coordinate17 | Determine the polar coordinates (r, theta) of a specified point. |
| coordinate18 | Determine the polar coordinates (r, theta) of the point with a given color. |
| coordinate19 | Determine which point shares the same polar coordinate (r or theta) as a given point. |
| coordinate20 | Determine which point shares the same polar coordinate (r or theta) as a point of a specified color. |
| coordinate21 | Identify the pair of points in polar coordinates that share the same coordinate (r or theta). |

| Key | Description |
| --- | --- |
| coordinate22 | Identify the colors of the two points in polar coordinates that share the same coordinate (r or theta). |
| coordinate23 | Determine the polar coordinates of a specified point and identify the point sharing a given coordinate attribute. |
| coordinate24 | Determine the polar coordinate attribute of a given point and identify the corresponding point with a matching attribute and its color. |
| **curvature** | |
| curvature1 | Determine which lines on the figure are straight segments (not curves) by identifying their colors. |
| curvature2 | Determine which lines on the figure are curves (not straight lines) by identifying their colors. |
| curvature3 | Determine which colored curve in the figure is part of a circle. |
| curvature4 | Determine which circle among several has the largest absolute curvature, and state its color. |
| curvature5 | Determine which circle among several has the smallest absolute curvature, and state its color. |
| curvature6 | Among several marked points on a curve, determine which point has the greatest absolute curvature (answer by label). |
| curvature7 | Among several marked points on a curve, determine which point has the smallest absolute curvature (answer by label). |
| **direction** | |
| direction1 | Determine which arrow among several has the most distinct direction compared to the others. |
| direction2 | Determine the direction in which a given arrow is pointing. |
| direction3 | Determine which arrow among multiple arrows is pointing in a specified direction. |

| Key | Description |
| --- | --- |
| `direction4` | Determine which object (or its label) is indicated by the central arrow's direction. |
| `direction5` | Determine the two objects that a two-headed arrow is pointing to. |
| `direction6` | Determine which object from a set is indicated by an arrow's direction. |
| `direction7` | Determine the sequence of textual elements arranged along the path of an arrow. |
| `direction8` | Determine the relative orientation of one arrow from the perspective of another arrow. |
| `direction9` | Determine the arrow's primary horizontal and vertical directional components based on its orientation. |
| `direction10` | Determine which arrow among several is pointing either towards or away from the center of the image. |
| **interior** | |
| `interior1` | Determine which points are inside a given shape. |
| `interior2` | Determine which points are outside a given shape. |
| `interior3` | Determine the colors of the points inside a given shape. |
| `interior4` | Determine the colors of the points outside a given shape. |
| `interior5` | Determine which shapes are inside a given shape. |
| `interior6` | Determine which shapes are outside a given shape. |
| `interior7` | Determine the colors of the shapes inside a given shape. |
| `interior8` | Determine the colors of the shapes outside a given shape. |
| `interior9` | Determine which points lie within a specified region. |
| `interior10` | Determine the colors of the points inside a specified region. |

| Key | Description |
| --- | --- |
| `interior11` | Determine which shapes are inside a specified region. |
| `interior12` | Determine the colors of the shapes inside a specified region. |
| **intersection** | |
| `intersection1` | Determine whether two given objects (line segments or curves) intersect. |
| `intersection2` | Determine whether two given objects (line segments or curves) do not intersect. |
| `intersection3` | Determine whether two given objects, specified by type and color, intersect. |
| `intersection4` | Determine whether two given objects, specified by type and color, do not intersect. |
| `intersection5` | Determine the number of intersection points between two given objects and select the correct count from the provided options. |
| `intersection6` | Determine the number of intersection points between two given objects, each specified by a color. |
| `intersection7` | Determine which line segment among several has the greatest number of intersections with a given curve. |
| `intersection8` | Determine how many line segments intersect with a given curve. |
| `intersection9` | Determine which curve among several has the highest number of intersections with a given line segment. |
| `intersection10` | Determine how many curves intersect with a given line segment. |
| `intersection11` | Determine whether the boundaries of two given shapes intersect. |
| `intersection12` | Determine whether the boundaries of two given shapes do not intersect. |
| `intersection13` | Determine the number of intersection points along the boundaries of two given shapes. |
| **length** | |

| Key | Description |
| --- | --- |
| length_1 | Determine which of two given line segments (formed by two pairs of points) is longer. |
| length_2 | Determine which of two given line segments (without point labels) is longer. |
| length_3 | Identify the longest line segment among a set of line segments, including one intentionally longer than the others. |
| length_4 | Compare the length of a given line segment with the radius of a circle and decide which is longer. |
| length_5 | Determine whether a given line segment is shorter than the radius of a specified circle. |
| length_6 | Verify that a given line segment has a length equal to the distance between two specified points. |
| length_7 | Determine whether a given line segment is longer than the diameter of a circle. |
| length_8 | Identify, among several points, which one is the farthest from a designated anchor point. |
| length_9 | Decide whether a given line segment is shorter than the diameter of a circle centered at a given point. |
| length_10 | Compare the length of a given line segment with the distance between two other points. |
| length_11 | Determine if a given line segment is exactly equal in length to the distance between two specified points. |
| length_12 | Decide whether a given line segment is longer than the distance between two designated points. |
| length_13 | Determine whether all three edges of a given triangle are equal in length (i.e. if the triangle is equilateral). |
| length_14_1 | Determine whether a triangle has two equal sides with the third side differing in length. |
| length_14_2 | Determine whether a triangle has two equal sides with one side distinctly shorter or longer, under a specific angular condition. |

| Key | Description |
| --- | --- |
| length_15 | Determine that a given triangle is scalene by verifying that all its sides differ in length. |
| length_16 | Identify which of two circles has a larger radius. |
| length_17_1 | Determine whether one line segment is exactly twice as long as another. |
| length_17_2 | Determine whether one line segment is exactly three times as long as another. |
| length_17_3 | Determine the ratio (between 2 and 5) by which one line segment is longer than another. |
| length_18 | Identify, among several lines, which line is exactly twice as long as a given line segment. |
| length_20 | Determine whether the height or the width is larger in a given rectangle. |
| length_21 | Identify the object (circle, rectangle, triangle, or bar) that is either the smallest or the tallest among a set of similar objects. |
| **line** | |
| line6 | Determine the colors of two connections: identify the color of the straight line versus the color of the curved line connecting given pairs of points. |
| line5 | List (or count) the points that are directly connected to a given point by a line segment. |
| line1-exists | From a set of options, select the one that correctly refers to a line that exists in the image. |
| line1-not | From a set of options, select the one that does NOT refer to any line present in the image. |
| line2-line | Identify which option represents a straight (non-curved) line among several curves. |
| line2-curve | Identify which option represents a curved line (as opposed to a straight line) among given choices. |
| line3-exists | List all the lines present in the image (or count how many lines there are). |
| line3-not | Select the option(s) that do NOT correspond to any line present in the image. |

| Key | Description |
| --- | --- |
| `line3-points` | List all the points that are connected by any line in the image and answer connectivity questions. |
| `line4` | Identify the endpoints of a specific line segment in the image. |
| **OCR** | |
| `ocr1` | Determine whether a specific element (number or alphabet) exists in the image and list all such elements present. |
| `ocr2` | Extract and verify the word that is shown in the image. |
| `ocr3` | Perform optical character recognition to read and output the text present in the image. |
| `ocr7` | Identify and read the word that appears flipped (upside down) in the image. |
| `ocr9` | Identify the pairs of alphabets and numbers in the image and answer related pairing questions. |
| `ocr14` | Extract the word formed by letters having a specified attribute (such as a particular color or bold style) from the image. |
| `ocr15` | Determine which letter in the given word has a circle drawn on it. |
| `ocr16` | Identify the letters that are circled in the displayed word in the image. |
| `ocr21` | Determine which letter, among the given options, does not appear in the image. |
| `ocr22` | Read and verify the text written in a specified color in the image. |
| `ocr23` | List all the letters that are visible in the image. |
| **ordinal** | |
| `ordinal_1` | Determine which shape occupies a specified ordinal position from the bottom among a set of distinct shapes. |
| `ordinal_2` | Determine which letter appears at a specified ordinal position from the top of the image, taking its case into account. |

| Key | Description |
| --- | --- |
| `ordinal_3` | Determine which labeled point holds a specified ordinal (e.g., second-highest) position on a 2D graph. |
| `ordinal_4` | Determine the label of the shape that ranks at a given ordinal position (e.g., third smallest) in terms of size. |
| `ordinal_5` | Determine which letter appears at a specified ordinal position from the top of the image, considering its case. |
| `ordinal_6` | Determine the (row, column) location of a specified element within a table in the image. |
| **orientation** | |
| `orientation_1` | Determine the correct reading direction (clockwise or counterclockwise) for letters arranged on a circle to form a specified word. |
| `orientation_2` | Determine the sequence of letters obtained by reading those arranged on a circle starting from a given letter in a specified order. |
| `orientation_2_1` | Determine the numerical sequence formed by reading the vertices of a rhombus in clockwise order starting from a specified vertex. |
| `orientation_3` | Determine whether a cycle in the image is oriented in a clockwise or counterclockwise direction. |
| `orientation_4` | Determine whether a set of arrows arranged on a circle are pointing clockwise or counterclockwise. |
| `orientation_5` | Identify the arrow that deviates from the common directional orientation among arrows on a circle. |
| **orthogonality** | |
| `ortho1-color` | Determine the color of the foot (projection point) of a given dot on a line. |
| `ortho1-label` | Determine the label of the foot (projection point) of a given dot on a line. |

| Key | Description |
| --- | --- |
| ortho2-color | Determine the color of the point that is obtained by orthogonally projecting a given dot onto the line. |
| ortho2-label | Determine the label of the point that results from the orthogonal projection of a given dot onto the line. |
| ortho3-color | Determine the color of the line that is perpendicular to a given line. |
| ortho3-label | Determine the label of the line that is perpendicular to a given line. |
| ortho4 | A test task for orthogonality, used as a placeholder. |
| **overlap** | |
| overlap_1 | Determine whether the interiors of two circles overlap. |
| overlap_2 | Determine whether the interiors of two triangles overlap. |
| overlap_2_1 | Determine whether the interiors of two squares overlap. |
| overlap_5 | Determine the shape of the intersection formed by the overlapping purple and orange regions. |
| **parallel** | |
| parallel_1 | Determine which line in the image is parallel to a given line. |
| C_parallel_1 | Determine which colored line in the image is parallel to a specified colored line. |
| C_parallel_1-2 | Determine which colored line is parallel to a given colored line in the image. |
| parallel_2 | Count how many additional lines (excluding the given line) are parallel to a specified line in the image. |
| parallel_2-2 | Count the number of lines, excluding the reference line, that are parallel to it. |
| C_parallel_2-2 | Count how many colored lines, excluding the reference line, are parallel to it in the image. |

| Key | Description |
| --- | --- |
| C_parallel_2 | Determine how many additional colored lines are parallel to a given colored line. |
| parallel_3 | Construct a parallelogram by finding the fourth vertex so that the opposite sides are parallel. |
| C_parallel_3 | Construct a colored parallelogram by determining the fourth vertex ensuring opposite sides are parallel. |
| parallel_4 | Identify the parallel edges of a trapezoid in the image. |
| C_parallel_4 | Identify the colored parallel edges of a trapezoid in the image. |
| C_parallel_4-2 | Identify the specific colored parallel edges of a trapezoid in the image. |
| parallel_5-3 | Determine the number of parallel pairs of lines present in a polygon shown in the image. |
| parallel_5-4 | Determine how many pairs of parallel lines exist in a polygon in the image. |
| parallel_5-5 | Determine the count of parallel line pairs in a polygon depicted in the image. |
| parallel_5-6 | Determine the number of parallel pairs of lines in a polygon in the image. |
| parallel_6 | Determine whether two specific lines in the image are not parallel. |
| parallel_7 | Determine that none of the lines from a given set are parallel to a specified line. |
| parallel_8 | Identify which side of a triangle is parallel to a given segment in the image. |
| parallel_9 | Evaluate a complex diagram to compare multiple parallelity relationships among lines. |
| parallel_10 | Determine that two specified lines in the image are not parallel. |
| **point** | |
| point_1 | Determine which label among a given set does not correspond to any point in the image. |

| Key | Description |
| --- | --- |
| point_2 | Identify the label that represents the only point present in the image. |
| **reflection** | |
| reflection_1 | Determine which candidate line is the reflection axis such that one point is the mirror image of another. |
| reflection_2 | Identify the sub-image that lacks line symmetry among several labeled options. |
| reflection_3 | Select the sub-image that exhibits line symmetry from a set of labeled figures. |
| reflection_4 | Determine which candidate line serves as the reflection axis for a pair of mirror-image shapes. |
| reflection_5 | Identify the reflection line (axis of symmetry) for a line-symmetric shape from multiple candidates. |
| reflection_6 | Determine which figure contains two shapes that are mirror reflections of each other with respect to a dividing line. |
| **rel_pos** | |
| rel_pos_above | Determine whether the given object is positioned above the reference object. |
| rel_pos_below | Determine whether the given object is positioned below the reference object. |
| rel_pos_left_of | Determine whether the given object is positioned to the left of the reference object. |
| rel_pos_right_of | Determine whether the given object is positioned to the right of the reference object. |
| rel_pos_between | Determine whether the given object is located between two reference objects. |
| rel_pos_adjacent | Determine which object is immediately adjacent to the specified reference object. |
| C_rel_pos_above | Determine whether the given colored object is positioned above the reference object. |
| C_rel_pos_below | Determine whether the given colored object is positioned below the reference object. |

| Key | Description |
| --- | --- |
| `C_rel_pos_left_of` | Determine whether the given colored object is positioned to the left of the reference object. |
| `C_rel_pos_right_of` | Determine whether the given colored object is positioned to the right of the reference object. |
| `C_rel_pos_between` | Determine whether the given colored object is located between two reference objects. |
| **rotation** | |
| `rotation_1` | Determine which triangle among a set of triangles is least likely to be a rotation of a given triangle about a specified point. |
| `rotation_2` | Determine which shape among several can be formed solely by rotating a given shape on the 2D plane. |
| `rotation_3` | Determine which shape among several cannot be produced solely by rotating a given shape on the 2D plane. |
| `rotation_4` | Determine which triangle can be obtained by rotating a given triangle by 90°. |
| `rotation_5` | Determine which heart shape is formed by rotating a given heart shape by 45°. |
| **rotational_symmetry** | |
| `rotational_symmetry_1` | Determine which point among a given set is most likely the center of 2-fold (point) rotational symmetry between two triangles. |
| `rotational_symmetry_2` | Determine which point (by its label) among several candidates serves as the center of rotational symmetry for a given shape. |
| `rotational_symmetry_3` | Determine the color of the point that marks the center of rotational symmetry for a given shape. |
| `rotational_symmetry_4` | Determine which shape among two candidates is most likely to exhibit 4-fold rotational symmetry (i.e. appearance preserved after a 90° rotation). |

| Key | Description |
| --- | --- |
| rotational_symmetry_4_1 | Identify the numbered shape that most likely possesses 4-fold rotational symmetry, given that 'Rotated <1>' represents a 90° rotation of shape <1>. |
| rotational_symmetry_5 | Determine which shape among several is symmetric to a given shape with respect to a point (i.e. exhibits point symmetry rather than mirror symmetry). |
| **shape** | |
| shape1 | Determine the overall shape of the single diagram shown in the image. |
| shape2 | List the names of the three shapes displayed in the image from left to right. |
| shape3 | Count the number of occurrences of a specified shape type present in the image. |
| shape4 | Identify the shape that is distinguished by a different color compared to the others in the image. |
| shape5 | Determine the name of the shape that is labeled with a given identifier. |
| shape6 | Determine which shape the arrow in the image is pointing to. |
| shape7 | Count the number of sides of the polygon shown in the image and state its name accordingly. |
| shape8 | Identify the common shape that appears in both parts of the image when it is divided into two sections. |
| shape9 | Identify the shape in the image that is not filled with any color. |
| **sharpness** | |
| sharp1 | Determine which part of the given curve is pointy (non-smooth) among the provided options. |
| sharp2 | Identify which of the pointy spots on the curve is the sharpest. |

| Key | Description |
| --- | --- |
| `sharp3-1` | Decide whether the curve is smooth (i.e. differentiable) at the indicated point. |
| `sharp3-2` | Decide whether the curve is pointy (i.e. not differentiable) at the indicated point. |
| `sharp4-smooth-1` | From the provided list, identify the segments of the curve that are smooth. |
| `sharp4-pointy-1` | From the provided list, identify the segments of the curve that are pointy. |
| `sharp4-smooth-2` | Select the labels corresponding to the smooth parts of the curve from the options given. |
| `sharp4-pointy-2` | Select the labels corresponding to the pointy parts of the curve from the options given. |
| **similarity** | |
| `similarity_1` | Determine which triangle among a set of given triangles is not similar to the others, while the remaining triangles share similarity. |
| `similarity_2` | Determine the pair of shapes that are similar to each other from a group of shapes, and provide their labels in lexicographical order. |
| `similarity_3` | Identify the shape that does not share similarity with the rest from a set of four labeled shapes. |
| `similarity_4` | Determine which one among five instances of a given shape is geometrically similar to the leftmost instance. |
| `similarity_5` | Determine which one among five instances of a given shape is not geometrically similar to the leftmost instance. |
| `similarity_6` | From three pairs of shapes, determine which pair exhibits a geometric similarity relationship. |
| **symbol** | |
| `symbol_1_a` | Determine which angle is represented by the angle symbol present in the image (often accompanying a triangle). |
| `symbol_1_b` | Determine which two lines form the angle indicated by the angle symbol in the image. |

| Key | Description |
| --- | --- |
| `symbol_1_c` | Determine, based on context, either the color of the angle symbol that marks a perpendicular angle or the two lines that form that perpendicular angle. |
| `symbol_2_a` | Determine which word among those displayed is marked as correct by a symbol (e.g., a green symbol) in the image. |
| `symbol_2_b` | Identify the word that is enclosed in a box of a specified color in the image. |
| `symbol3_a` | Determine the relationship between two lines as indicated by an angle symbol that denotes parallelism. |
| `symbol3_b` | Determine the relationship between two lines as indicated by a symbol that denotes that the lines have the same length. |
| `symbol4_a` | Determine which pair of lines (or sides) have equal length based on the information provided by the symbol. |
| `symbol4_b` | Determine which pair of sides in a polygon have equal length as indicated by the symbol. |
| `symbol4_c` | Interpret the compound symbol–often involving concatenated labels–to decide the equality or relationship between segments, as indicated by the symbol. |
| `symbol5` | Determine the meaning of a given symbol by choosing the correct option from multiple alternatives provided. |
| **tangency** | |
| `tangency1` | Determine whether the given line and the given curve in the figure touch at exactly one point of tangency. |
| `tangency2` | Determine whether the given line and the given curve in the figure do not meet tangentially. |
| `tangency3` | Determine whether the two curves, each drawn in a different color, meet at a common tangent point. |

| Key | Description |
|---|---|
| tangency4 | Determine whether the two differently colored curves in the figure do not form a tangential contact. |
| tangency5 | Determine whether the given <1> (e.g., a specific shape) and the adjacent line segment in the figure are tangent to each other. |
| tangency6 | Determine whether the given <1> and the line segment in the figure do not touch tangentially. |
| tangency7 | Identify, among the intersection points of the line and the curve, which ones are points of tangency. |
| tangency8 | Identify, among the intersection points of the two differently colored curves, which ones are tangent contacts. |
| tangency9 | Determine the color(s) of the curve(s) that are tangent to the given line at the specified point. |
| **texture** | |
| texture1 | Determine which of the given <1> lines has a different line style (or color) than the others. |
| texture2 | Determine which line among the given set has the same line style as line <1>. |
| texture3_a | Determine which one of the four shapes in the image has a texture that differs from the rest. |
| texture4 | Determine, among the four options, which shape (or side) has the same or different texture compared to the reference <2> <1>. |
| texture6 | Determine the line style of the given line <1> in the image. |
| texture7 | Determine the total number of segments into which the given line is divided by its texture pattern. |
| texture8 | Order the given <1> lines by the density of their texture (breaks per unit length) and identify the one with the most (or most sparse) texture. |
| texture9 | Count the number of distinct line styles present among the <1> lines in the image. |

| Key | Description |
| --- | --- |
| `texture10` | Determine the total number of $<1>$ lines in the image. |
| `texture11_one_different` | Determine which one of the $<1>$ shapes is drawn with a different line style (texture) compared to the others. |
| `texture11_all_different` | Identify the shape that is drawn with a specified $<1>$ line style among the given shapes. |
| `texture12` | Determine the shape (or the number of sides of the shape) that is formed by the $<1>$ lines in the image. |
| `texture13` | Determine the line style of the given $<1>$ based on its texture. |
| `texture14` | Identify, among the $<1>$ scattered lines, the $<2>$ line and state its $<3>$ (e.g., its label or color). |
| `texture15` | Determine the identifier (e.g., label) of the line that has a different line style (texture) from the rest among the given $<3>$ lines. |
| **width** | |
| `width1-color-widest` | Determine which road (or bar) with a specific color is the widest among the given options. |
| `width1-color-narrowest` | Determine which road (or bar) with a specific color is the narrowest among the given options. |
| `width1-label-widest` | Determine which road (or bar) with a given label is the widest among the provided options. |
| `width1-label-narrowest` | Determine which road (or bar) with a given label is the narrowest among the provided options. |
| `width2` | Determine whether the road is getting wider or narrower when moving in a specified direction. |
| `width3-label-widest` | Among several labeled roads (or bars), determine which one is the widest. |
| `width3-label-narrowest` | Among several labeled roads (or bars), determine which one is the narrowest. |

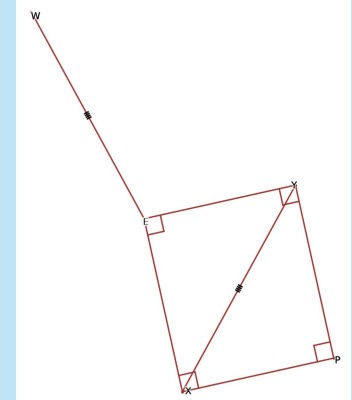

**Question**

Between (i) and (ii), which best describes the image? (i) A rectangle YPXE is present. (ii) Circles Y and W meet tangentially at N, while lines XP and PU also contact circle W at E and I, and point X belongs to circle Y.

| Ground Truth | Gemini 2.5 Flash |
|:---:|:---:|
| (i) | (i) |
| **OpenAI o3** | **Math-LLaVA** |
| (i) | (ii) |

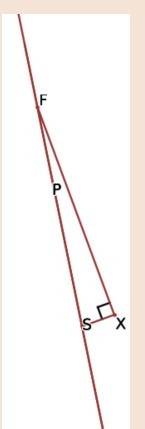

**Question**

True or False: The perpendicularity of XS and XF highlights their orthogonal relationship.

| Ground Truth | Gemini 2.5 Flash |
|:---:|:---:|
| True | False |
| **OpenAI o3** | **Math-LLaVA** |
| False | True |

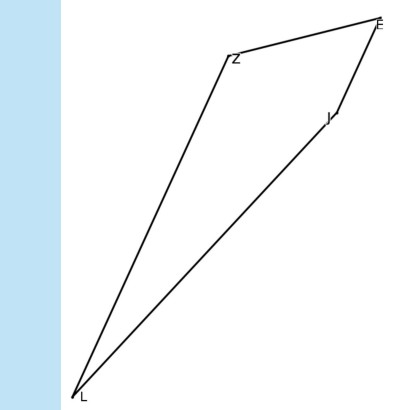

**Question**

True or False: Vertices J, E, Z, and L make up a convex four-sided polygon.

| Ground Truth | Gemini 2.5 Flash |
|:---:|:---:|
| True | True |
| **OpenAI o3** | **Math-LLaVA** |
| True | True |

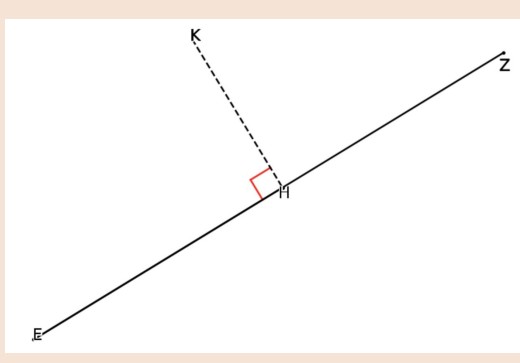

**Question**

Between (i) and (ii), which best describes the image? (i) The right-angle intersection from K to EZ is located at H. (ii) There exist a triangle KHZ that is right-angled and isosceles, with KH = KZ.

| Ground Truth | Gemini 2.5 Flash |
|:---:|:---:|
| (i) | (i) |
| **OpenAI o3** | **Math-LLaVa** |
| (i) | (i) |

Figure 14: Examples of $\nu$-geometry with fewer steps (1-3), and the responses from o3, Gemini 2.5 Flash, and Math-LLaVA.

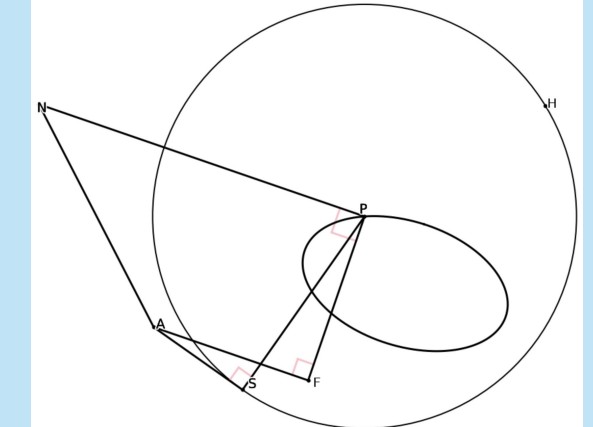

**Question**
True or False: Center P defines a circle and includes H as a boundary point.

| Ground Truth | Gemini 2.5 Flash |
|---|---|
| True | False |
| **OpenAI o3** | **Math-LLaVA** |
| False | True |

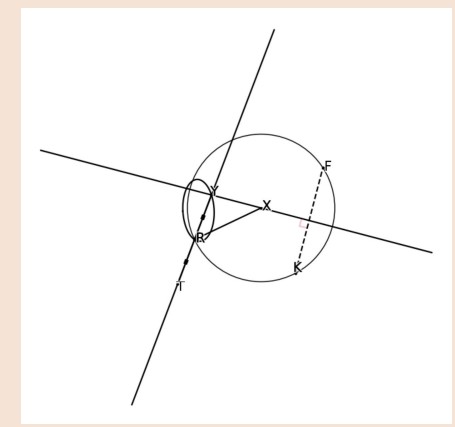

**Question**
Between (i) and (ii), which best describes the image? (i) Circle F and circle Y share two common points T, X. (ii) There exist a reflective setup with line XY as the mirror line, flipping K to F.

| Ground Truth | Gemini 2.5 Flash |
|---|---|
| (ii) | (ii) |
| **OpenAI o3** | **Math-LLaVA** |
| (ii) | (i) |

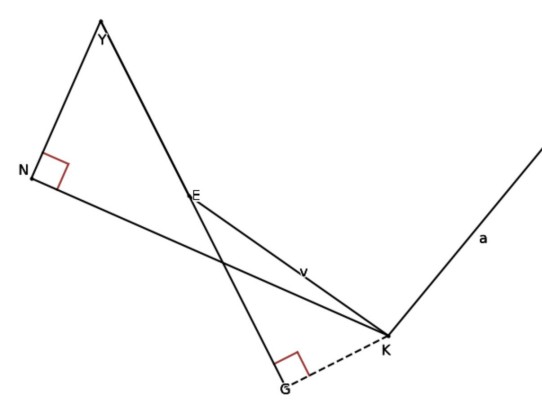

**Question**
Between (i) and (ii), which best describes the image? (i) Circle K lies within triangle GNY, tangent to each side. (ii) Line v shares the same endpoint K with line KN.

| Ground Truth | Gemini 2.5 Flash |
|---|---|
| (ii) | (ii) |
| **OpenAI o3** | **Math-LLaVA** |
| (ii) | (ii) |

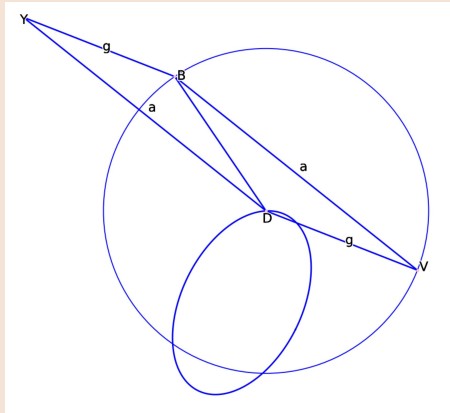

**Question**
Between (i) and (ii), which best describes the image? (i) There exist a center D and a circle that goes through V. (ii) There exist a common tangent IX for circles D and V in the plane.

| Ground Truth | Gemini 2.5 Flash |
|---|---|
| (i) | (i) |
| **OpenAI o3** | **Math-LLaVa** |
| (i) | (i) |

Figure 15: Examples of $\nu$-geometry with more steps (4-6), and the responses from o3, Gemini 2.5 Flash, and Math-LLaVA.

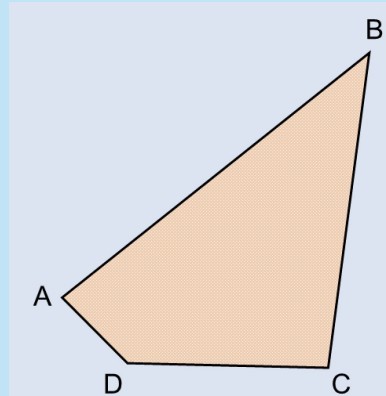

**Question**
Choose the words in parentheses that correctly describe the image. Rewrite the sentence with the chosen words. In the image, the longest line segment of the quadrilateral ABCD is line segment (AB/BC/CD/DA), and the shortest one is line segment (AB/BC/CD/DA).

| Ground Truth | GPT-4o |
|---|---|
| AB, DA | BC, AD |
| **OpenAI o3** | **Gemini 2.5 Flash** |
| BC, DA | AB, CD |

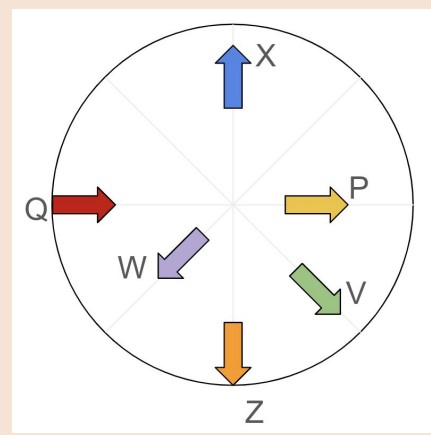

**Question**
In the image, inside a circle, there are arrows labeled as P, Q, X, V, W, and Z. Only one of them is pointing inward, while the others are pointing outward. Guess the arrow pointing inward.

| Ground Truth | GPT-4o |
|---|---|
| Q | Z |
| **OpenAI o3** | **Gemini 2.5 Flash** |
| Q | W |

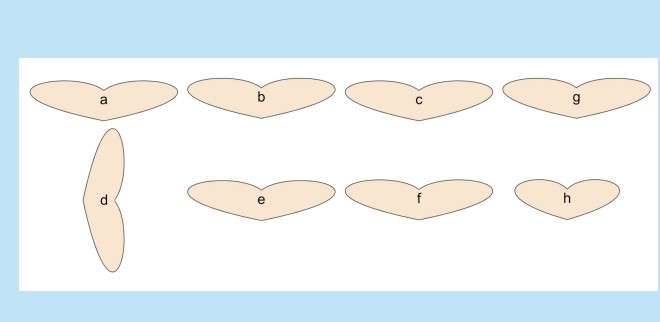

**Question**
In the image, all but one shape are congruent to each other, meaning they can be overlapped perfectly by moving, rotating, and flipping. Choose the one shape that looks different. Answer from: a, b, c, d, e, f, g, h.

| Ground Truth | GPT-4o |
|---|---|
| h | d |
| **OpenAI o3** | **Gemini 2.5 Flash** |
| h | d |

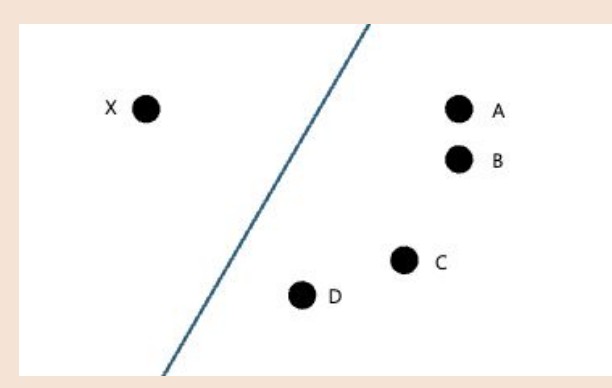

**Question**
Choose the word in parentheses that correctly describes the image. Rewrite the sentence with the chosen word. "Point X and point (A/B/C/D) are symmetric with respect to the blue line."

| Ground Truth | GPT-4o |
|---|---|
| C | D |
| **OpenAI o3** | **Gemini 2.5 Flash** |
| C | C |

Figure 16: Examples of AVSD-h, and the responses from o3, GPT-4o, and Gemini 2.5 Flash.

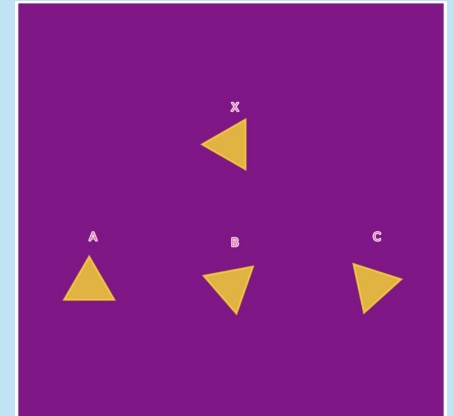

**Question**
In the image, out of triangles A, B, and C, which one is formed by rotating X by 90 degrees?

| Ground Truth | GPT-4o |
|---|---|
| A | B |
| **OpenAI o3** | **Gemini 2.5 Flash** |
| B | C |

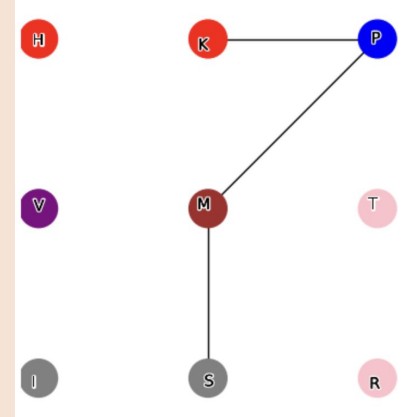

**Question**
Determine the count of connected components in the graph where edges link some of the 9 nodes.

| Ground Truth | GPT-4o |
|---|---|
| 6 | 6 |
| **OpenAI o3** | **Gemini 2.5 Flash** |
| 6 | 9 |

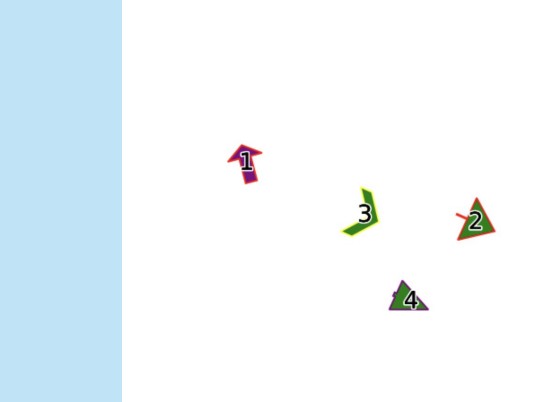

**Question**
Choose the arrow in the picture with distinct direction from the others.

| Ground Truth | GPT-4o |
|---|---|
| 1 | 1 |
| **OpenAI o3** | **Gemini 2.5 Flash** |
| 2 | 2 |

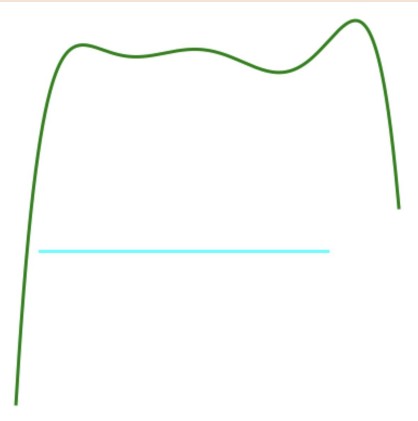

**Question**
Identify the total number of intersection points that occur when the line segment in cyan intersects with the curve in green.

| Ground Truth | GPT-4o |
|---|---|
| 0 | 4 |
| **OpenAI o3** | **Gemini 2.5 Flash** |
| 1 | 2 |

Figure 17: Examples of AVSD-s, and the responses from o3, GPT-4o, and Gemini 2.5 Flash.

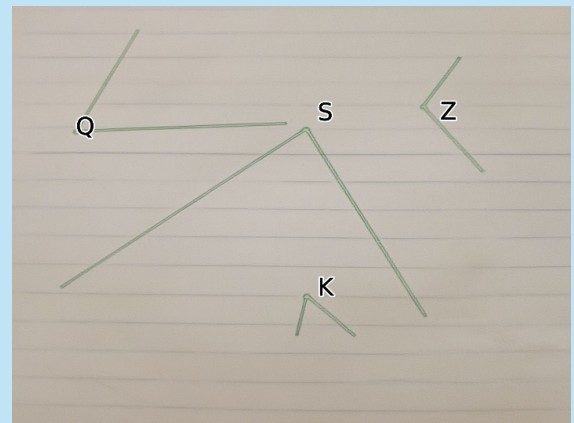

**Question**

Among the angles Z, K, Q, and S, which of them are obtuse?

| Ground Truth | GPT-4o |
|---|---|
| Z | Q, S |
| **OpenAI o3** | **Gemini 2.5 Flash** |
| Q, S | S |

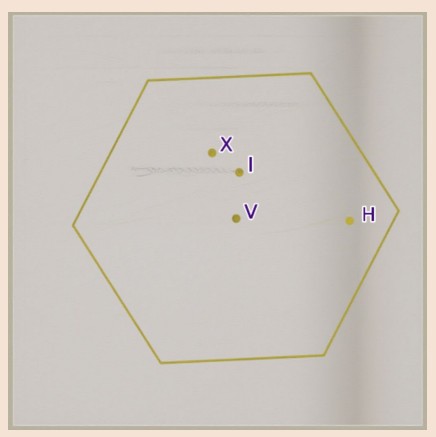

**Question**

A hexagon in the image demonstrates rotational symmetry around a point. Identify the center of this symmetry by its label (I, X, V, or H). For instance, if X is the center, respond with "X".

| Ground Truth | GPT-4o |
|---|---|
| V | I |
| **OpenAI o3** | **Gemini 2.5 Flash** |
| I | I |

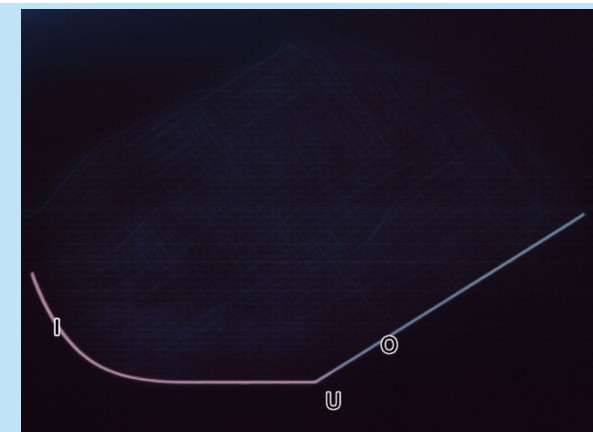

**Question**

Is the curve in the image smooth at U?

| Ground Truth | GPT-4o |
|---|---|
| No | No |
| **OpenAI o3** | **Gemini 2.5 Flash** |
| No | No |

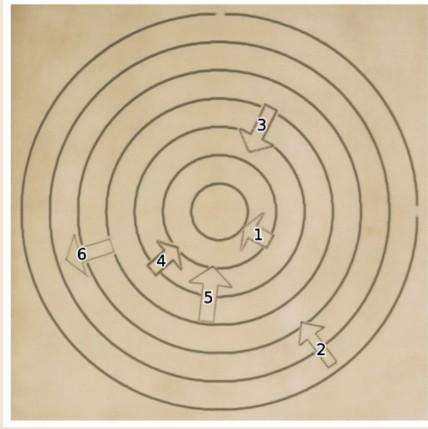

**Question**

Among the arrows in the diagram, only one arrow is pointing outwards from the center. What is the label of that arrow?

| Ground Truth | GPT-4o |
|---|---|
| 6 | 2 |
| **OpenAI o3** | **Gemini 2.5 Flash** |
| 6 | 2 |

Figure 18: Examples of AVSD-c, and the responses from o3, GPT-4o, and Gemini 2.5 Flash.

Table 17: Summary of Prompts used in the Generation of AVSD-c

| Category | Description |
| --- | --- |
| **Whiteboard Backgrounds** | |
| Canny Edge | Diagram on a whiteboard with Canny edge detection. |
| Clean Whiteboard | Diagram drawn with black marker on a clean whiteboard. |
| Classroom Whiteboard | Diagram drawn with red and blue markers on a classroom whiteboard. |
| Smudged Whiteboard | Diagram on a whiteboard with smudged dry-erase marker traces. |
| Glass Whiteboard | Diagram on a large office-style glass whiteboard with colorful markers. |
| Worn-Out Whiteboard | Diagram on a worn-out school whiteboard covered in faint eraser marks. |
| Equation Whiteboard | Diagram on a whiteboard with handwritten equations in different colors. |
| Clear Glass Whiteboard | Diagram on a clear glass whiteboard with reflections. |
| Smart Whiteboard | Diagram on a digital smart whiteboard screen. |
| Augmented Reality Whiteboard | Diagram on an augmented reality screen with floating digital lines. |
| **Chalkboard Backgrounds** | |
| Black Chalkboard | Diagram on a black chalkboard with hand-drawn chalk lines. |
| Smudged Chalkboard | Diagram on a black chalkboard with smudged chalk marks. |
| Colored Chalkboards | Diagram on blue, red, purple, brown, or gray chalkboards. |
| Dusty Blackboard | Diagram on a dusty, old-school blackboard. |
| Multicolored Chalkboard | Diagram on a multicolored chalkboard with blended pastel chalk. |
| **Paper Backgrounds** | |
| Lined Paper | Diagram on a lined page, drawn with black lines. |
| Grid Paper | Diagram on a grid page, drawn with black lines. |
| Narrow-Lined Paper | Diagram on a narrow-lined page. |
| Wide-Ruled Notebook | Diagram on a wide-ruled notebook page. |
| College-Ruled Notebook | Diagram on a college-ruled notebook page. |
| Dotted Notebook | Diagram on a dotted notebook page. |

| Category | Description |
| --- | --- |
| Graph Paper | Diagram on graph paper with blue grid lines. |
| Vintage Parchment | Diagram on vintage parchment paper with ink. |
| Torn Notebook Page | Diagram on a torn-out notebook page. |
| Crumpled Paper | Diagram on crumpled lined paper with faded ink. |
| Coffee-Stained Notebook | Diagram on a coffee-stained notebook page with ink smudges. |

### Special Surface Backgrounds

| Category | Description |
| --- | --- |
| Stone Wall | Diagram sketched on a stone wall with charcoal. |
| Wooden Surface | Diagram drawn on a wooden surface with white chalk. |
| Fabric | Diagram stitched on fabric with embroidery. |
| Ancient Manuscript | Diagram on an ancient manuscript with faded ink. |
| Engineering Blueprint | Diagram on an engineering blueprint. |
| Newspaper Style | Diagram on a newspaper-style background. |
| Neon Digital Display | Diagram on a neon-lit digital display. |
| Retro Computer Screen | Diagram on a retro computer screen in pixel art style. |
| Holographic Board | Diagram on a glowing holographic board. |
| Sci-Fi Display | Diagram on a sci-fi holographic display. |
| Transparent Digital Screen | Diagram on a futuristic transparent OLED screen. |
| Futuristic Touchscreen | Diagram on a futuristic touch-screen whiteboard. |

### Artistic and Abstract Backgrounds

| Category | Description |
| --- | --- |
| Textured Sketchpad | Diagram on a textured sketchpad with pencil marks. |
| Watercolor Background | Diagram on a watercolor-painted background. |
| Starry Sky | Diagram on a starry night sky, drawn with glowing lines. |
| Graffiti Wall | Diagram on a graffiti-covered wall with spray paint. |
| Foggy Mirror | Diagram on a foggy mirror, drawn with a fingertip. |

### Variations in Color and Line Styles

| Category | Description |
| --- | --- |
| Colored Lined Paper | Diagram on a lined page with black, blue, red, yellow, green, orange, purple, or golden lines. |
| Colored Grid Paper | Diagram on a grid page with black, blue, red, yellow, white, neon cyan, pink, or green lines. |
| College-Ruled Notebook Ink Variants | Diagram drawn on a college-ruled notebook page with black, blue, red, purple, green, or white ink. |
| Aged Parchment | Diagram on an aged parchment with dark brown ink sketches. |

| Category | Description |
|---|---|
| Old Manuscript | Diagram on an ancient scroll with sepia ink. |
| Newspaper Print | Diagram on a crumpled newspaper page with gray pencil strokes. |
| Whiteboard Marker Colors | Diagram drawn on a whiteboard with black, blue, red, green, orange, or metallic gold markers. |
| Chalk Colors | Diagram on a black or gray chalkboard drawn with white, yellow, pink, blue, neon green, orange, or mixed colored chalks. |
| Futuristic Neon | Diagram on a futuristic glass screen with glowing neon blue, cyan, or magenta lines. |
| Augmented Reality | Diagram on an augmented reality screen with floating yellow digital lines. |

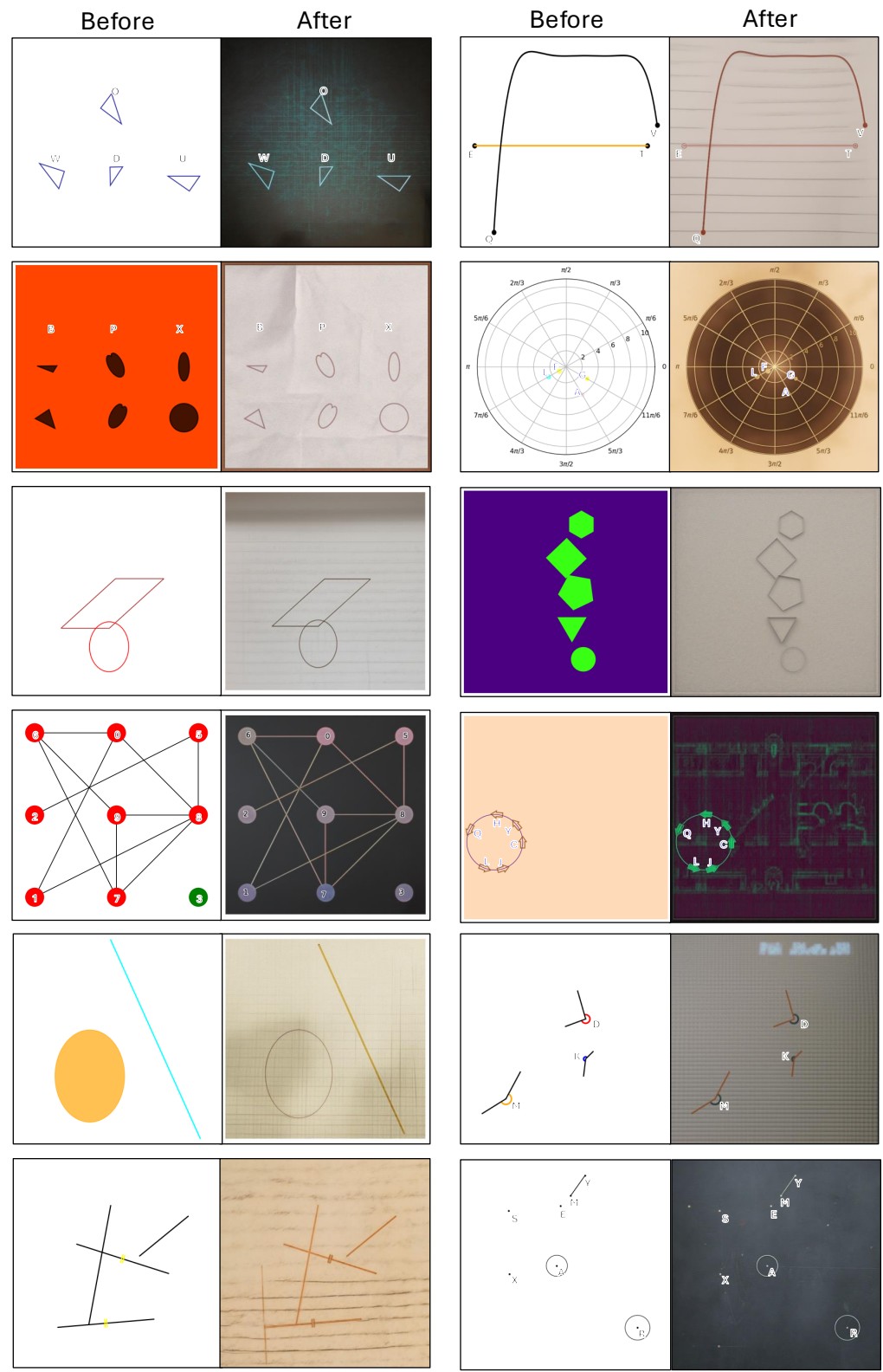

Figure 19: Examples of before and after the style transformation via ControlNet.

