# OpenReview forum: "Decomposing Complex Visual Comprehension into Atomic Visual Skills for Vision Language Models"
_DMLR — Accepted by DMLR_

### Review · Reviewer_wXLQ · 2025-11-12

**Recommendation:** 5
**Confidence:** 3

**Summary Of Contributions:**

The paper introduces a benchmark dataset to evaluate the performance of VLMs on "atomic geometric" tasks.
These tasks are so named as they are (i) intuitive to humans, (ii) indivisible (doing so would be "unnatural") and
(iii) they are complete as in no further such skills are required to deal with high school or lower geometric tasks.
Evaluating current models on this dataset, the paper finds that models across the board find similar tasks both easy
and hard, that CoT (chain-of-thought) is not helpful but "reasoning" models may be and that domain specific models
do not do better.

**Strengths:**

See above:
- Well written article with clear presentation of the results.
- Interesting results arising out of this novel benchmark dataset
- a new benchmark dataset that evaluates basic skills.

**Audience:**

Yes

**Broader Impact Concerns:**

No concerns here.

**Claims And Evidence:**

Apart form the highlighted bits above, the paper's claims are adequately supported by evidence.

**Datasets And Benchmarks:**

Sufficient detail, data (on huggingface) as well as the production code is available.

**Extended Submissions:**

Not applicable here.

**Limitations:**

Apart from the requested changes above to address the stated weaknesses, I do not think the paper has limitations. On the contrary.
The paper's observed results upon using its own benchmark are rather interesting and bring the scientific community forward.

**Requested Changes:**

- Please add ablation studies to the single network dependent parts of your work, i.e. to using something different to ControlNet to generate AVSD-c and using a different network to GPT-4o-mini in your evaluation.

**Strengths And Weaknesses:**

Strengths:
- Well written article with clear presentation of the results.
- Interesting results arising out of this novel benchmark dataset
- a new benchmark dataset that evaluates basic skills.

Weaknesses:
- The word "atomic" in "atomic visual skills" is a distraction. To me this word would assume that there is even a mathematical proof or at least no doubt that these elements are indivisible. The author's definition itself, in point (ii) needs to refer back to human intuition of further dividing these problems as being "unnatural". I think the authors would do well calling these skills "primitives" and maintain this human intuition definition rather than get into a futile argument over whether skill 35 could or should be decomposed further. The dataset and tasks are, imho, intuitive and pertinent and show a basic command of geometric skills a human would be expected to display.
- The third step in the evaluation protocol is dependent on one particular chosen model, here GTP-4o mini. Does the answer change if other models are used? Similarly, the style change model's (ControlNet) impact on the entire generation of AVSD-c is unknown and it would be important to do an ablation study whether results change significantly if another network was used.

---

### Review · Reviewer_F3wV · 2025-11-26

**Recommendation:** 4
**Confidence:** 3

**Summary Of Contributions:**

This paper proposes a novel evaluation paradigm for visual understanding, decomposing complex geometric diagram comprehension tasks into 36 "atomic visual skills." These skills are defined as fundamental perceptual abilities that are intuitive for adults, indivisible, and sufficient to support the understanding of geometric diagrams at the high school level and below. To systematically assess these abilities, the authors construct the Atomic Visual Skills Dataset (AVSD), comprising three subsets: AVSD-h (5,163 handcrafted problems), AVSD-s (5,400 procedurally generated problems based on AlphaGeometry rules), and AVSD-c (2,625 style-augmented problems generated using ControlNet), totaling 13,188 samples. The dataset is designed to achieve three key goals: diversity, skill isolation, and style robustness. Evaluation results on 11 mainstream VLMs show that even the strongest closed-source models (e.g., Gemini 2.5 Pro, o3) perform near random levels on multiple atomic skills (such as tangency, parallelism, and angle judgment). Chain-of-thought (CoT) prompting barely improves performance, indicating that the failure stems from low-level perception rather than high-level reasoning. Domain-specific models (e.g., Math-LLaVA, G-LLaVA) do not demonstrate advantages, and models are extremely sensitive to image style changes—performance generally drops by 5–10 percentage points on AVSD-c. Further experiments reveal that fine-tuning on composite geometric data (e.g., MathV360k) fails to enhance atomic skill performance, while direct training on AVSD-s-train (360,000 atomic problems) significantly improves generalization on AVSD-h. This indicates that atomic-level perception needs to be explicitly modeled during training and cannot be indirectly acquired through composing high-level tasks. This work uncovers the systematic deficiencies of current VLMs in basic visual perception and provides a "bottom-up" new path for future pre-training and evaluation.

**Strengths:**

1. The paper's primary contribution lies in its methodical decomposition of geometric perception into 36 atomic visual skills. The selection criteria—intuitive for humans, indivisible, and comprehensive for high school geometry—provide a principled foundation. Each skill is precisely defined in Appendix B with clear visual examples, demonstrating rigorous conceptual work. The coverage spans from basic skills (OCR, color, cardinal counting) to more nuanced geometric concepts (tangency, rotational symmetry, convexity), creating a valuable diagnostic toolkit.
2. The three-tiered dataset architecture addresses complementary evaluation needs. AVSD-h provides human-crafted diversity and eliminates data contamination concerns. AVSD-s enables scalable generation with 360,000 training examples, supporting reproducible research. AVSD-c's ControlNet-based style augmentation rigorously tests robustness, revealing a 5-15% performance drop across models (Table 3). The generation pipeline's quality control measures—Canny edge filtering, Bhattacharyya distance thresholding, and post-generation text scribing—ensure technical soundness.
3. Several results challenge conventional wisdom. The failure of Math-LLaVA and G-LLaVA (domain-specific models) suggests that current geometry-specialized training emphasizes reasoning over atomic perception. The neutral-to-negative effect of chain-of-thought prompting (Table 4) supports the authors' hypothesis that these tasks require perception rather than reasoning. The style sensitivity analysis quantifies a critical vulnerability in VLMs. The fine-tuning experiment demonstrates that atomic skill training yields clear improvements (0.33→0.45 on AVSD-h), while composite geometry training (MathV360k) does not, providing evidence for the pre-training hypothesis.
4. Using GPT-4o mini for answer extraction and scoring, validated against human annotation (99% agreement on 150 samples), balances scalability with accuracy. The two-stage evaluation process (extraction then judgment) with carefully designed prompts (Tables 10-11) minimizes scoring bias. The inclusion of ν-geometry as a composite perception baseline strengthens the motivation for atomic decomposition.

**Audience:**

Yes

**Broader Impact Concerns:**

The study uses only abstract geometric figures—no people, text, or cultural artefacts—so privacy or representational harms are negligible.
A positive consequence is that better geometric perception could improve AI tutors or computer-aided design tools; a negative risk is that automated graders built on such models might unfairly penalise students whose hand-drawn diagrams deviate from the clean synthetic style on which the benchmark is optimised.
Because the benchmark is publicly released under a permissive licence, future commercial systems could advertise “100 % AVSD accuracy” without revealing that the skills are still far from human-level robustness; the authors might consider adding a cautionary clause about out-of-distribution deployment.

**Claims And Evidence:**

The main empirical claims are supported by large-scale evaluation: VLMs perform far below ceiling on AVSD, domain-specific variants do not excel, chain-of-thought prompting helps little, and style shifts hurt performance.
Exact numbers, statistical spreads, and model versions are supplied, and code plus data are promised to be open-sourced, so the evidence is transparent and reproducible.
What remains thinner is causal explanation: the paper does not link failures to specific architectural components (e.g., patch size, vision encoder depth, or language bias) and the “pre-training hypothesis” is illustrated by only one fine-tuning contrast, not by a systematic ablation of data mixture or model scale.The submission is not described as an extended version of any prior publication; no overlap with previously published material is declared.

**Datasets And Benchmarks:**

The paper supplies a URL for the full dataset, generation scripts, and detailed card-style documentation (number of images, splits, licences, and maintenance plan).
Reproducibility is supported by open-sourcing the AlphaGeometry-based generator, ControlNet prompts, and evaluation prompts for GPT-4o-mini scoring.
Ethical considerations are minimal but explicitly discussed.

**Extended Submissions:**

The submission is not described as an extended version of any prior publication; no overlap with previously published material is declared.

**Limitations:**

The paper’s central assertion that the 36 skills are “atomic” is not accompanied by a formal decomposition argument or human-development evidence, so the reader is left to trust the authors’ intuition that skills such as “Angle”, “Symbol”, or “Tangency” cannot be split further.
Synthetic generation ensures variety but also guarantees clean, uncluttered diagrams; real textbooks contain overlapping labels, erased construction lines, and deliberate perspective distortions that are never tested, so the reported difficulty gap may be conservative.
Every score is produced by GPT-4o-mini parsing model answers; although the authors validate 150 cases, subtle mismatches between what the VLM outputs and what the scorer extracts are still possible, and no human re-grading of failure modes is offered.
Fine-tuning experiments compare only two data recipes (MathV360k versus AVSD-s-train) and keep architectural details short; without learning curves, frozen-component ablations, or cross-validation on held-out skills it is hard to tell whether the gains are robust or simply over-fitting to synthetic templates.

**Requested Changes:**

1. While the 36 skills are intuitively reasonable, the atomicity claim warrants deeper justification. Some skills appear compositionally reducible: for instance, "Angle" skill (understanding angle representation) seems to depend on "Line" and "Point" detection. "Symbol" skill (interpreting geometric notation) arguably requires OCR and spatial reasoning. Appendix B examples show that many problems implicitly combine multiple sub-skills. A formal graph-based dependency analysis or human factor study of skill acquisition order would strengthen the atomicity argument.
2. The paper identifies that models fail but provides minimal analysis of why. The visual encoders (CLIP-style) are known to prioritize semantic features over geometric precision; however, no attention visualization or feature attribution experiments probe failure modes. For example, do models confuse tangency with intersection due to limited resolution in patch-based encoding? Does style sensitivity stem from diffusion model training biases? Connecting perceptual failures to architectural limitations would elevate the work from diagnosis to explanation.
3. The reliance on GPT-4o mini for scoring, while validated, introduces a meta-model dependency that may obscure subtle errors. For instance, if a VLM outputs "yellow" instead of "orange" for a tangent circle, the scoring LLM might correctly extract "yellow" but miss that the error stems from color discrimination, not geometric perception. Human evaluation of a stratified error sample (e.g., 50 failures per model) with error categorization would provide richer diagnostic information.
4. The conclusion that "domain-specific models are not better" may be premature. Math-LLaVA and G-LLaVA are optimized for solving geometry problems (requiring theorem application and algebraic reasoning), not perceiving diagrams. Their training data likely emphasizes textual derivations over pixel-level geometric feature extraction. A more appropriate comparison would be models fine-tuned specifically on diagram parsing tasks (e.g., chart understanding models) or vision models pre-trained on technical drawings.
5. AVSD-s and AVSD-c, while procedurally diverse, may not capture the full complexity of real-world geometry diagrams. Textbook diagrams often contain contextual clutter (labels, theorems, auxiliary constructions) and intentional perceptual ambiguities that test human interpretation. The paper's synthetic diagrams are clean and unambiguous by design, potentially underestimating the difficulty of real geometric perception. A small validation set scraped from actual textbooks would ground the findings in practical applications.
5. The manuscript presents a compelling approach for reasoning tasks for visual inputs. To further elevate its scholarly impact, the related work section could engage with recent developments: IR3D-Bench: Evaluating Vision-Language Model Scene Understanding as Agentic Inverse Rendering,jarvisir: elevating autonomous driving perception with intelligent image restoration,JarvisArt: Liberating Human Artistic Creativity via an Intelligent Photo Retouching Agent.
6. The handcrafted dataset's easy/medium/hard classifications (Appendix C.1) are determined by authors' subjective time-to-solve metrics. This introduces potential bias and limits reproducibility. A crowdsourced difficulty rating from multiple human annotators with inter-annotator agreement metrics would strengthen this aspect.
7. The paper uses inconsistent abbreviations: "LVN" for LLaVA-NeXT in tables but "LN" in text (Section 5). The ν-geometry symbol (ν) appears inconsistently formatted. Table 1's column headers lack clarity on what "LN-13b" versus "LN-34b" specifically refer to. Standardizing notation throughout would improve readability.
8. The fine-tuning experiments (Section 5.2) lack crucial hyperparameters: learning rate, batch size, number of epochs, and whether vision encoder weights were frozen. These details are essential for reproducibility. The claim that training took "1–2 days on 8 H100 GPUs" is vague.

**Strengths And Weaknesses:**

**Strengths:**
1. The paper's primary contribution lies in its methodical decomposition of geometric perception into 36 atomic visual skills. The selection criteria—intuitive for humans, indivisible, and comprehensive for high school geometry—provide a principled foundation. Each skill is precisely defined in Appendix B with clear visual examples, demonstrating rigorous conceptual work. The coverage spans from basic skills (OCR, color, cardinal counting) to more nuanced geometric concepts (tangency, rotational symmetry, convexity), creating a valuable diagnostic toolkit.
2. The three-tiered dataset architecture addresses complementary evaluation needs. AVSD-h provides human-crafted diversity and eliminates data contamination concerns. AVSD-s enables scalable generation with 360,000 training examples, supporting reproducible research. AVSD-c's ControlNet-based style augmentation rigorously tests robustness, revealing a 5-15% performance drop across models (Table 3). The generation pipeline's quality control measures—Canny edge filtering, Bhattacharyya distance thresholding, and post-generation text scribing—ensure technical soundness.
3. Several results challenge conventional wisdom. The failure of Math-LLaVA and G-LLaVA (domain-specific models) suggests that current geometry-specialized training emphasizes reasoning over atomic perception. The neutral-to-negative effect of chain-of-thought prompting (Table 4) supports the authors' hypothesis that these tasks require perception rather than reasoning. The style sensitivity analysis quantifies a critical vulnerability in VLMs. The fine-tuning experiment demonstrates that atomic skill training yields clear improvements (0.33→0.45 on AVSD-h), while composite geometry training (MathV360k) does not, providing evidence for the pre-training hypothesis.
4. Using GPT-4o mini for answer extraction and scoring, validated against human annotation (99% agreement on 150 samples), balances scalability with accuracy. The two-stage evaluation process (extraction then judgment) with carefully designed prompts (Tables 10-11) minimizes scoring bias. The inclusion of ν-geometry as a composite perception baseline strengthens the motivation for atomic decomposition.

**Weakness:**
1. While the 36 skills are intuitively reasonable, the atomicity claim warrants deeper justification. Some skills appear compositionally reducible: for instance, "Angle" skill (understanding angle representation) seems to depend on "Line" and "Point" detection. "Symbol" skill (interpreting geometric notation) arguably requires OCR and spatial reasoning. Appendix B examples show that many problems implicitly combine multiple sub-skills. A formal graph-based dependency analysis or human factor study of skill acquisition order would strengthen the atomicity argument.
2. The paper identifies that models fail but provides minimal analysis of why. The visual encoders (CLIP-style) are known to prioritize semantic features over geometric precision; however, no attention visualization or feature attribution experiments probe failure modes. For example, do models confuse tangency with intersection due to limited resolution in patch-based encoding? Does style sensitivity stem from diffusion model training biases? Connecting perceptual failures to architectural limitations would elevate the work from diagnosis to explanation.
3. The reliance on GPT-4o mini for scoring, while validated, introduces a meta-model dependency that may obscure subtle errors. For instance, if a VLM outputs "yellow" instead of "orange" for a tangent circle, the scoring LLM might correctly extract "yellow" but miss that the error stems from color discrimination, not geometric perception. Human evaluation of a stratified error sample (e.g., 50 failures per model) with error categorization would provide richer diagnostic information.
4. The conclusion that "domain-specific models are not better" may be premature. Math-LLaVA and G-LLaVA are optimized for solving geometry problems (requiring theorem application and algebraic reasoning), not perceiving diagrams. Their training data likely emphasizes textual derivations over pixel-level geometric feature extraction. A more appropriate comparison would be models fine-tuned specifically on diagram parsing tasks (e.g., chart understanding models) or vision models pre-trained on technical drawings.
5. AVSD-s and AVSD-c, while procedurally diverse, may not capture the full complexity of real-world geometry diagrams. Textbook diagrams often contain contextual clutter (labels, theorems, auxiliary constructions) and intentional perceptual ambiguities that test human interpretation. The paper's synthetic diagrams are clean and unambiguous by design, potentially underestimating the difficulty of real geometric perception. A small validation set scraped from actual textbooks would ground the findings in practical applications.
6. The handcrafted dataset's easy/medium/hard classifications (Appendix C.1) are determined by authors' subjective time-to-solve metrics. This introduces potential bias and limits reproducibility. A crowdsourced difficulty rating from multiple human annotators with inter-annotator agreement metrics would strengthen this aspect.